# Infection length and host environment influence on *Plasmodium falciparum* dry season reservoir

Carolina M Andrade [1,11,14 ✉], Manuela Carrasquilla [2,14], Usama Dabbas[1,2], Jessica Briggs[3,4], Hannah van Dijk [1], Nikolay Sergeev [1], Awa Sissoko[5], Moussa Niangaly[5], Christina Ntalla[2], Emily LaVerriere [6,7], Jeff Skinner [8], Klara Golob[2], Jeremy Richter[2], Hamidou Cisse [5], Shanping Li [8], Jason A Hendry[2], Muhammad Asghar [9,12,13], Didier Doumtabe [5], Anna Farnert[9], Thomas Ruppert[10], Daniel E Neafsey[6,7], Kassoum Kayentao[5], Safiatou Doumbo[5], Aissata Ongoiba[5], Peter D Crompton[8], Boubacar Traore[5], Bryan Greenhouse[3,4] & Silvia Portugal [1,2 ✉]

## Abstract

Persistence of malaria parasites in asymptomatic hosts is crucial in areas of seasonally-interrupted transmission, where *P. falciparum* bridges wet seasons months apart. During the dry season, infected erythrocytes exhibit extended circulation with reduced cytoadherence, increasing the risk of splenic clearance of infected cells and hindering parasitaemia increase. However, what determines parasite persistence for long periods of time remains unknown. Here, we investigated whether seasonality affects plasma composition so that *P. falciparum* can detect and adjust to changing serological cues; or if alternatively, parasite infection length dictates clinical presentation and persistency. Data from Malian children exposed to alternating ~6-month wet and dry seasons show that plasma composition is unrelated to time of year in non-infected children, and that carrying *P. falciparum* only minimally affects plasma constitution in asymptomatic hosts. Parasites persisting in the blood of asymptomatic children from the dry into the ensuing wet season rarely if ever appeared to cause malaria in their hosts as seasons changed. In vitro culture in the presence of plasma collected in the dry or the wet seasons did not affect parasite development, replication or host-cell remodelling. The absence of a parasite-encoded sensing mechanism was further supported by the observation of similar features in *P. falciparum* persisting asymptomatically in the dry season and parasites in age- and sex-matched asymptomatic children in the wet season. Conversely, we show that *P. falciparum* clones transmitted early in the wet season had lower chance of surviving until the end of the following dry season, contrasting with a higher likelihood of survival of clones transmitted towards the end of the wet season, allowing for the re-initiation of transmission. We propose that the decreased virulence observed in persisting parasites during the dry season is not due to the parasites sensing ability, nor is it linked to a decreased capacity for parasite replication but rather a consequence decreased cytoadhesion associated with infection length.

**Keywords** Malaria; Dry Season; Sensing; Infection Length; Asymptomatic
**Subject Category** Microbiology, Virology & Host Pathogen Interaction

## Introduction

Progress against malaria remains globally uneven, and improvements achieved in the last decades have recently plateaued. In 2022, there were over 249 million cases of malaria caused by *P. falciparum* in 85 countries, and more than 600 thousand deaths, most of which were of African children under five (World Health Organization, 2023). *P. falciparum* asymptomatic infections constitute a reservoir of parasites (Barry et al, 2021; Bousema et al, 2014; Greenwood, 1987), contributing to local malaria prevalence and continuously challenging elimination efforts. In areas of seasonal transmission, the presence of asymptomatic infections during mosquito-free periods facilitates the restart of transmission when mosquitos return. In Mali, clinical malaria cases occur during the wet season, and asymptomatic infections serve as a reservoir during the 5- to 6-month dry season (Crompton et al, 2008; Portugal et al, 2017; Sondén et al, 2015). Recently, we reported that during the dry season, *P. falciparum* is maintained at low parasitaemia through increased circulation of infected red blood

[1]Center for Infectious Diseases, Parasitology, Heidelberg University Hospital, Heidelberg, Germany. [2]Max Planck Institute for Infection Biology, Berlin, Germany. [3]Department of Medicine, San Francisco General Hospital, University of California, San Francisco, CA, USA. [4]Chan-Zuckerberg Biohub, San Francisco, CA, USA. [5]Mali International Center of Excellence in Research, University of Sciences, Techniques and Technologies of Bamako, Bamako, Mali. [6]Department of Immunology and Infectious Diseases, Harvard TH Chan School of Public Health, Boston, MA, USA. [7]Broad Institute, Cambridge, MA, USA. [8]Laboratory of Immunogenetics, NIAID, National Institutes of Health, Rockville, MD, USA. [9]Division of Infectious Diseases, Department of Medicine Solna, Karolinska Institute, Stockholm Sweden and Department of Infectious Diseases, Karolinska University Hospital, Stockholm, Sweden. [10]ZMBH, Heidelberg University, Heidelberg, Germany. [11]Present address: Department of Medical Microbiology, Radboud University Medical Center, Nijmegen, the Netherlands. [12]Present address: Department of Biology, Lund University, Lund, Sweden. [13]Present address: Department of Healthcare Biotechnology, Atta-Ur-Rahman School of Applied Biosciences, National University of Sciences and Technology, Islamabad, Pakistan. [14]These authors contributed equally: Carolina M Andrade, Manuela Carrasquilla. ✉E-mail: carolina.andrade@radboudumc.nl; portugal@mpiib-berlin.mpg.de

cells (iRBCs) within each ~48 h asexual replication, promoting a higher risk of splenic clearance of iRBCs, and keeping parasite burdens below the immunological radar and preventing potentially fatal host pathology (Andrade et al, 2020). But what leads to the longer circulation time of iRBCs in the dry season is not defined. We hypothesise that either sensing the time of year, or accumulation of time since infection promotes or hinders *P. falciparum* silent persistence.

*Plasmodium* parasites require nutrients and metabolites from their host, and respond to dramatic changes between mosquito and human environments that grant the continuity of the life cycle. They respond to external factors like temperature, pH or xanthurenic acid (Billker et al, 1998), and contact with substrates of different elasticities (Ripp et al, 2021) in the mosquito; temperature and nutrient composition inform the transition from the mosquito into liver-stage forms (Hegge et al, 2010; Kappe et al, 2003); and limiting concentration of glucose, or the presence of detrimental amino acids affect liver-stage development (Meireles et al, 2020; Meireles, Mendes, et al, 2017; Meireles, Sales-Dias, et al, 2017). Molecular sensing mechanisms, however, have not been clearly defined in these studies. During the blood-stage of infection, nutrient availability can determine parasite replication and sexual commitment (Brancucci et al, 2017; Mancio-Silva et al, 2017; Marreiros et al, 2023). The kinase KIN was identified as a sensor activated by low nutritional conditions, altering gene expression and decreasing parasite replication (Mancio-Silva et al, 2017); and eIK1 and eIK2 were implicated in parasite development and replication in limiting amino acid conditions (Marreiros et al, 2023). Lysophosphatidylcholine (Lyso-PC) was shown to repress the pro-sexual commitment transcription factor AP2-G, and its low concentrations induce the positive transcriptional regulators GDV1 (Filarsky et al, 2018) and HDP1 (Campelo Morillo et al, 2022), increasing the proportion of sexual forms in vitro (Brancucci et al, 2017). Lastly, a response to febrile temperatures was recently reported and the transcription factor PfAP2-HS shown to regulate the protective heat-shock response in *P. falciparum* (Tintó-Font et al, 2021), but the sensing mechanism remains elusive. Nonetheless, the question of whether the parasite adapts its replication in response to seasonal variations in the host environment remains unexplored.

Prolonged survival of *P. falciparum* infections has been documented in humans. From several case reports reviewed by Ashley and White, the longest *P. falciparum* infection reported is thought to have lasted over 10 years in an asymptomatic individual departing from Senegal (Ashley and White, 2014). In malaria-naïve patients subjected to *P. falciparum*-induced fever therapy for neurosyphilis, infections often lasted ~200 days (Sama et al, 2006). Additionally, infections lasting more than 380 days have been detected in migrant populations (Wångdahl et al, 2023). In areas of seasonal transmission, the longevity of particular *P. falciparum* clones can reach one year (Babiker et al, 1998; Hamad et al, 2000); but often, infection length during the transmission season appears to be much shorter with the rapid acquisition of new clones replacing previous ones (Daubersies et al, 1996; Pinkevych et al, 2015). To survive long dry seasons, *P. falciparum* needs to establish persistent infections. These are often maintained within individuals from the beginning to the end of the dry season without the acquisition of new clones (Portugal et al, 2017), but whether the time of transmission in the preceding wet season favours

persistence until the end of the dry season is not known. In a high transmission area of Malawi, where transmission is perennial, the median duration of one *P. falciparum* clone was ~40 days, and adults and school-aged children had longer infections than children below five years (Buchwald et al, 2019). In a low transmission setting of Uganda, infection length was ~100 days (Briggs et al, 2020), and monoclonal infection, low parasite density, and female sex of participants were associated with faster clearance of asymptomatic infections.

Here, we aimed to elucidate whether sensing the time of year, or length of time since infection underlie aspects of *P. falciparum* infection associated with silent persistence. With plasma and parasites sampled longitudinally from children living in a sharply seasonal region of Mali, we interrogated potential serological differences in metabolome, lipidome and proteome; and we tested how *P. falciparum* replication and host-cell remodelling responded to cues present in the plasmas obtained during the dry or the wet seasons. Additionally, we used genomic DNA from dried-blood spots (DBSs) collected longitudinally from children to follow individual parasite clones over time, and related parasite genotype and duration of infection with the host clinical presentation. We found no evidence that a sensing mechanism of seasonal cues promotes the clear pattern of malaria cases observed during the wet season and the absence of clinical disease, despite parasite carriage in ~20% of children, in the dry season. Instead, our data indicates that time of transmission and length of infection may determine the reservoir composition, and that the parasites surviving the dry season are transmitted late in the preceding wet season.

## Results

During the wet season, *P. falciparum* infections in Malian children are frequently associated with high parasite densities and symptomatic disease. By contrast, dry season infections are frequently of lower density and asymptomatic. To test if these differences could be mediated by seasonal cues, we investigated (i) potential variations between plasma samples collected during the wet and the dry seasons, (ii) whether low and persisting parasitaemias at the end of the dry season could expand to high levels associated with clinical malaria cases once transmission resumed and (iii) the in vitro response of parasites from the dry and wet seasons when exposed to plasma collected at different timepoints in the year. Furthermore, we investigated whether the time of transmission and duration of infection of *P. falciparum* clones were associated with the parasites' ability to persist during the dry season, and we examined iv) the length of infection of clones acquired during the wet season, and the risk of persistence of *P. falciparum* clones transmitted early and late during the wet season (Fig. 1).

### Seasonality and asymptomatic infections promote minimal differences in plasma metabolites, proteins and lipids

First, we investigated seasonal properties of host serology by profiling the metabolome of plasma samples ($n = 38$) collected from six clinical malaria cases in the wet season, and from up to 11 asymptomatic children in the dry and the wet seasons.

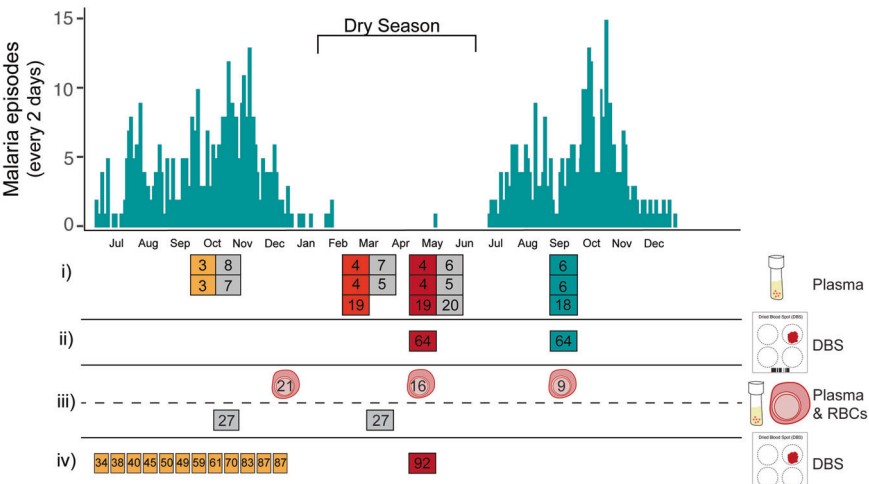

**Figure 1. Seasonal malaria and study design.**

Top panel shows a histogram of malaria cases in a cohort of ~600 individuals aged 3 months to 45 years, binned to 2-day intervals for 15 months in Kalifabougou, Mali. Malaria cases were defined by axillary temperature ≥37.5 °C, ≥2500 asexual parasites/μl of blood and no other apparent cause of fever. The bottom panel shows a cross-sectional sampling of plasma, DBSs, and fresh RBC pellets used to investigate (i) seasonal variations in plasma metabolites, lipids and proteins; (ii) the presence of clones persisting through the dry season in subsequent malaria episodes of the same individuals in the ensuing wet season; (iii) the in vitro response of parasites freshly collected from asymptomatic carriers in the dry season and clinical malaria cases in the wet season to supplementation of pooled plasmas from the dry and the wet seasons; and (iv) the approximate time of transmission, in the preceding wet season, of clones persisting until the end of the dry season. Negative samples for *P. falciparum* are shown in grey, *P. falciparum* positive samples of asymptomatic carriers are shown in yellow in October and during the wet season, light red in March and dark red in May during the dry season; and clinical cases obtained through passive surveillance are shown in aqua. The number of samples used for each analysis is specified within the coloured boxes or the representative RBC drawings. Source data are available online for this figure.

Asymptomatic individuals included donors that either did or did not carry *P. falciparum* parasites throughout the year (Fig. 1). We performed Ultra-high-performance liquid chromatography/triple quadrupole mass spectrometry (UHPLC-MS/MS) metabolomics on plasma from children who were *P. falciparum* positive or non-infected in the middle of the wet season in (October), and through the mid and end of the dry season (March and May), into the ensuing wet season, when many children presented at the clinic with malaria symptoms (MAL). Principal components analysis (PCA) of 456 metabolites in 35 samples that passed the filtering criteria (Dataset EV1) showed no clear segregation of samples between the different timepoints, infectious status, clinical presentation, or donor (Figs. 2A and EV1A,B). While we identified no significant differences (Pairwise student's *t*-test, adjusted *p* value <0.05) between asymptomatic carriers and non-infected individuals, we could identify two significantly different metabolites between mid-dry season (March) and mid-wet season (October), 11 significantly different metabolites between asymptomatic carriers and clinical cases, and 38 significantly different metabolites between non-infected individuals and clinical cases (Figs. 2B and EV1C). Of note, several Lyso-PC species and other glycerolphospholipids, a choline cofactor, and TAG-related metabolites were among the analytes decreased in clinical malaria samples; as were also several amino acids and amino acid-related metabolites (Figs. 2C and EV1D; Dataset EV2). Accordingly, a pathway enrichment analysis comparing clinical malaria versus asymptomatic carriers and negative individuals highlighted amino acid biosynthesis as the most affected pathway in clinical malaria samples (Fig. EV1E; Dataset EV3); and an overrepresentation analysis of all metabolite data using a hypergeometric test

(Pang et al, 2021) identified amino acid and glycophospholipid metabolism as the most impacted pathways (Dataset EV3). Additionally, we compared the concentration of 21 lipids, by Liquid chromatography–mass spectrometry (LC-MS), present in the plasma of children infected during the dry season (March and May, *n* = 19), versus those of age- and sex-matched negative individuals at the end of the dry season (May⁻, *n* = 20), and children showing clinical malaria symptoms in the wet season (MAL, *n* = 18). After filtering, we analysed data of 20 lipids from plasma samples of 19 individuals infected during the dry season in March and May, and 20 uninfected individuals in May, plus 15 malaria cases in the wet season. Once more, the principal component analysis did not reveal any separation of samples by either time of the year, infectious status or clinical manifestation (Fig. EV1E; Dataset EV4). These data confirmed the significantly lower levels of Lyso-PC 18.0 in malaria cases vs dry season infected or negative individuals, and it provided a more sensitive and comprehensive exploration of other potentially relevant lipid-related pathways. We found small but statistically significant differences in phosphatidylethanolamines and Lyso-PC 20.4, and one lysophosphatidic acid, which appeared increased in malaria cases, as well as one lipid species belonging to the lysophosphatidic acid class (Fig. 2D; Dataset EV4).

Furthermore, we analysed the proteome of plasma samples of 12 asymptomatic individuals that were infected or not during the dry and wet seasons, as well as six malaria cases during the wet season. After depleting the most abundant plasma proteins and implementing missingness filtration, we identified 146 proteins in 30 samples and this analysis did not reveal significantly enriched proteins among the compared groups (Dataset EV5). Unsupervised

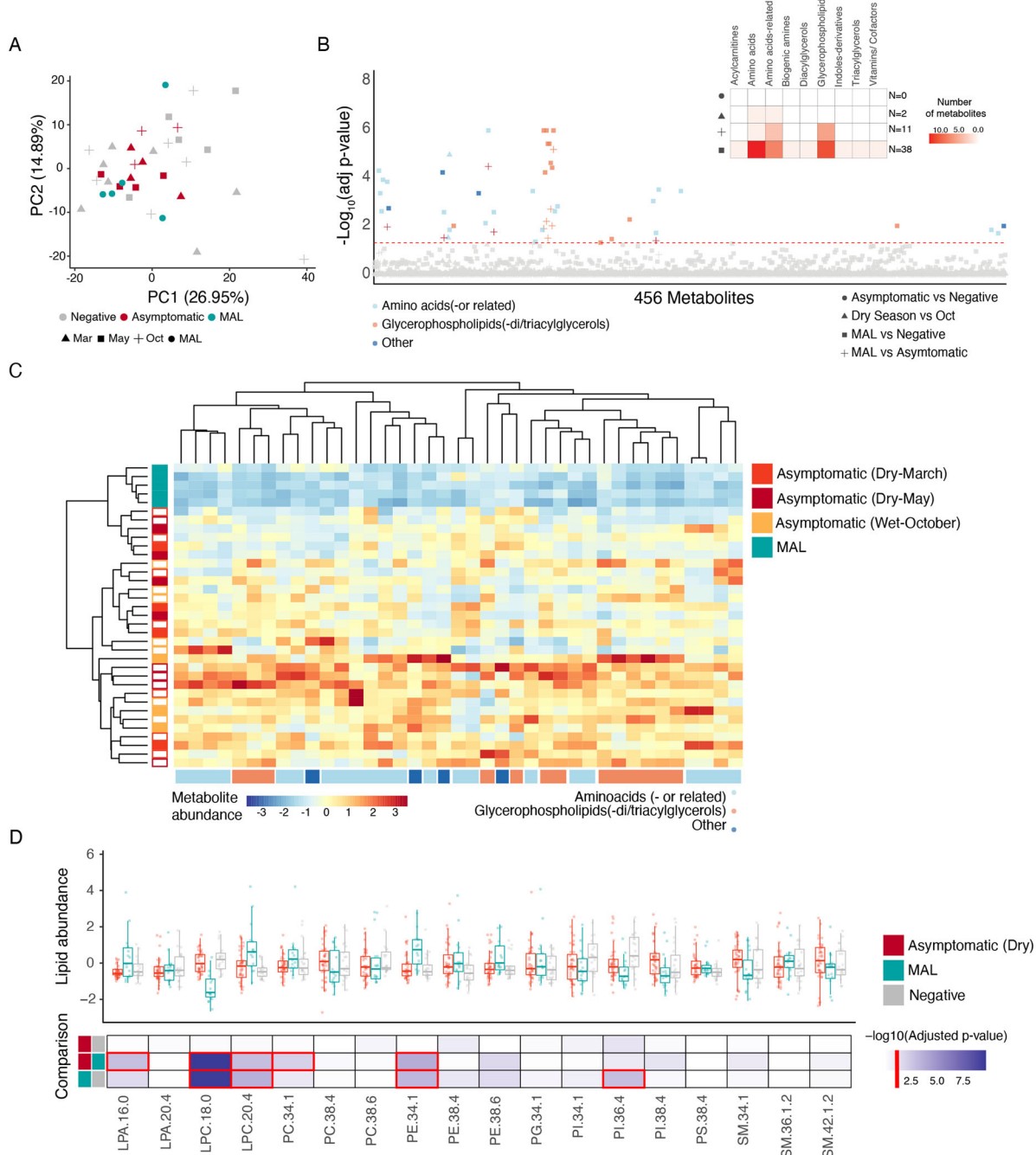

**Figure 2. Seasonality and asymptomatic infections promote minimal differences in plasma metabolites proteins and lipids.**

(A) Principal component analysis of 456 metabolites in the plasma of uninfected individuals (negative, $n = 19$) obtained in March ($n = 7$), May ($n = 5$) and October ($n = 7$), *P. falciparum* asymptomatic carriers ($n = 11$) obtained in March ($n = 4$), May ($n = 4$) and October ($n = 3$), and clinical cases during the wet season (MAL) ($n = 5$). (B) Pairwise comparison of plasma metabolites grouped into amino acids (red), glycerophospholipids (orange) and other metabolites (blue), between different infection status or time of the year. Adjusted $p$ (FDR), negative-log-transformed (y-axis) from $t$-test performed on each metabolite abundance and between sample groups (dot shape); x-axis sorted by group category and adjusted $p$ value. The panel box summarises the main metabolite type and the number of significantly different metabolites between the sample group comparison (Student's $t$-test). The Red dashed line shows the threshold of significance. (C) Hierarchical clustering of 39 significantly different metabolite abundances between-group comparisons. Hierarchical clustering applied to metabolites and samples is coloured by infection status or time of the year. Metabolites are grouped by categories as in (B). (D) Lipidomic analysis of 73 samples from 53 individuals, of whom 19 had paired samples collected in March ($n = 19$) and May ($n = 19$) in the dry season, 15 had clinical malaria ($n = 15$), and 20 were negative ($n = 20$) at the end of the dry season. Significantly different metabolites across all sample comparisons are grouped by lipid type class and timepoint or infectious status (Boxplots indicate median ± IQR with all individual values plotted, and pairwise Student's $t$-test, and adjusted $p$ values are shown). Heatmap of adjusted $p$ (FDR), negative-log-transformed from ANOVA and Tukey multiple comparison test performed on the abundance of each lipid species and between sample groups. Red boxes highlight significant metabolites found between the group comparisons, and the red line in the scale shows the threshold of significance.

hierarchical clustering analysis did not demonstrate any distinct segregation patterns among the samples (Fig. EV1F). Finally, we examined haemoglobin concentration throughout the year, in the blood of children older than 5 years, and observed no significant differences in infected or non-infected individuals, or due to clinical malaria (Fig. EV1G).

## Dry and wet season plasma have a similar effect on *P. falciparum* in vitro

Hypothesising the presence of a potential parasite-encoded sensing mechanism, we then investigated whether the minor differences detected in plasma composition between the dry and the wet seasons could impact *P. falciparum* growth or host-cell remodelling in vitro. We cultured *P. falciparum* 3D7 in RPMI 1640 medium (Trager and Jensen, 1976) with a physiological concentration of glucose (Desai, 2013), supplemented with increasing concentrations (20–98%) of uninfected Malian or German plasma for 56 h. In both conditions, we observed an inverse relationship between the percentage of plasma supplementation and the increase in parasitaemia after 56 h. We also observed a larger parasitaemia increase in the presence of German compared to Malian plasma, likely due to the presence of *P. falciparum*-specific antibodies in the Malian samples (Fig. 3A). We selected 25% plasma supplementation for further experiments and cultured three laboratory-adapted strains of *P. falciparum* (3D7, Dd2 and HB3) with paired plasma samples collected from both uninfected and infected Malian children aged 7–8 years at the end of the wet season (December) or at the end of the dry season (May) (Fig. EV2A). Flow cytometry analysis of the geometric mean fluorescence of nuclear staining (Sybr Green) revealed comparable parasite development in iRBCs following in vitro culture (Fig. EV2B), and the percentage of erythrocytes stained over time showed similar increases in parasitaemia after 60 h, regardless of the season of the plasma supplement (Fig. 3B). The vast majority of the Malian plasma samples supported parasite development and replication in vitro, suggesting the possibility of pooling plasma samples collected at the same timepoint to obtain larger volumes and allow antibody depletion. With this strategy, we used Pf2004, a lab-adapted strain of *P. falciparum* previously shown to respond to Lyso-PC 16:0, Lyso-PC 18.0 and Lyso-PC 18:1 levels in vitro (Brancucci et al, 2017), and we cultured it for 166 h, comprising three replicative cycles, and observed that parasitaemia increased similarly in the presence of the end of wet (Dec) or end of the dry season (May) pooled plasma from 8-year old uninfected Malians (Fig. EV2C).

We next investigated if *P. falciparum* parasites freshly isolated from Malian individuals would be more sensible to seasonal cues than lab-adapted strains. We asked whether freshly-collected *P. falciparum* parasites from rapid diagnostic test positive (RDT)+ individuals during the dry season, and from age- and sex-matched clinical malaria cases in the wet season, would be affected by 48-h in vitro culture with 25% plasma pooled from 27 uninfected donors, aged 9–16 years old at the end of the dry season (Dry) or at the peak of the wet season (Wet). Parasite samples collected at the end of the wet season (January, $n = 21$), end of the dry season (May, $n = 16$), and from clinical malaria cases during the wet season (MAL, $n = 9$) were used to test if a potential sensing mechanism would be more clearly detected during part of the year, measuring parasitaemia at 0, 18, 24, 36 and/or 46 h in the different

supplementation conditions by flow cytometry. Our data showed similar increases in parasitaemia independently of the time of collection of the plasma used to supplement the medium, or the presence of the plasma replacement Albumax (Fig. 3C). We did observe, however, that parasites from asymptomatic children in the dry season (Jan and May) increased parasitaemia after a shorter time in culture than parasites collected from clinical malaria cases during the wet season (MAL), consistent with our previous report of extended circulation of iRBCs in the dry season compared to clinical malaria cases (Andrade et al, 2020). This observation was confirmed by the statistically significant difference in the average fold change of percentage of iRBCs at 24 and 36 h between parasites collected in dry season or malaria cases, regardless of the plasma supplementation condition (Fig. EV2D).

Additionally, we inquired whether host-cell remodelling of iRBCs was affected by the supplementation with dry or wet season plasma, since it could affect circulation time. After panning *P. falciparum* FCR3 iRBCs on primary human dermal microvascular endothelial cells to enrich for parasites capable of making knobs (Lekana Douki et al, 2002), we cultured these iRBCs for 48 h in a medium supplemented with 25% plasma pooled from 27 donors collected at the end of the dry season (Dry) or at the peak of the wet season (Wet). Micrographs obtained through transmission and scanning electron microscopy of trophozoites and schizonts revealed similar knob diameter (Fig. 3D) and density (Fig. 3E,F) in the two conditions of seasonal plasma supplementation. Altogether, in the conditions tested, our data does not support the presence of a sensing mechanism responding to serological cues and altering *P. falciparum* growth or host-cell remodelling in vitro.

## Clinical malaria episodes are caused by newly transmitted *P. falciparum* clones

We then investigated if *P. falciparum* parasites carried from the dry into the ensuing wet season within an individual could increase parasitaemia and potentially cause clinical malaria once the wet season resumed. With genomic DNA obtained from longitudinally collected DBSs on filter paper from asymptomatic children who were persistently infected during the dry season, and later presented clinical malaria in the wet season, we performed *ama1* amplicon sequencing (Miller et al, 2017) ($n = 64$), and evaluated data of capillary electrophoresis of *msp*2 genotyping (Liljander et al, 2009) ($n = 7$) performed in the context of a previous study (Portugal et al, 2017). With both genotyping methods, we observed that in more than 75% of the clinical malaria cases, alleles present did not match, in sequence nor size, those observed in the same child at the end of the preceding dry season (Figs. 4A and EV3). This indicates that clinical malaria cases were primarily caused by *P. falciparum* alleles transmitted in the ongoing wet season. The *ama1* locus analysis identified 41 unique haplotypes in 305 infections from 64 individuals. From the 41 unique haplotypes detected in the two timepoints (May and MAL), 13 were only present in the malaria cases, that were haplotypes with the lowest population allele frequency; and only 5 were only present at the end of the dry season in May, which were also amongst the least frequent alleles in the population. Of the 64 children, 15 displayed one or more matching haplotypes between the end of the dry season and the following clinical malaria episode (Fig. EV3). Importantly, out of these 15 individuals who potentially maintained

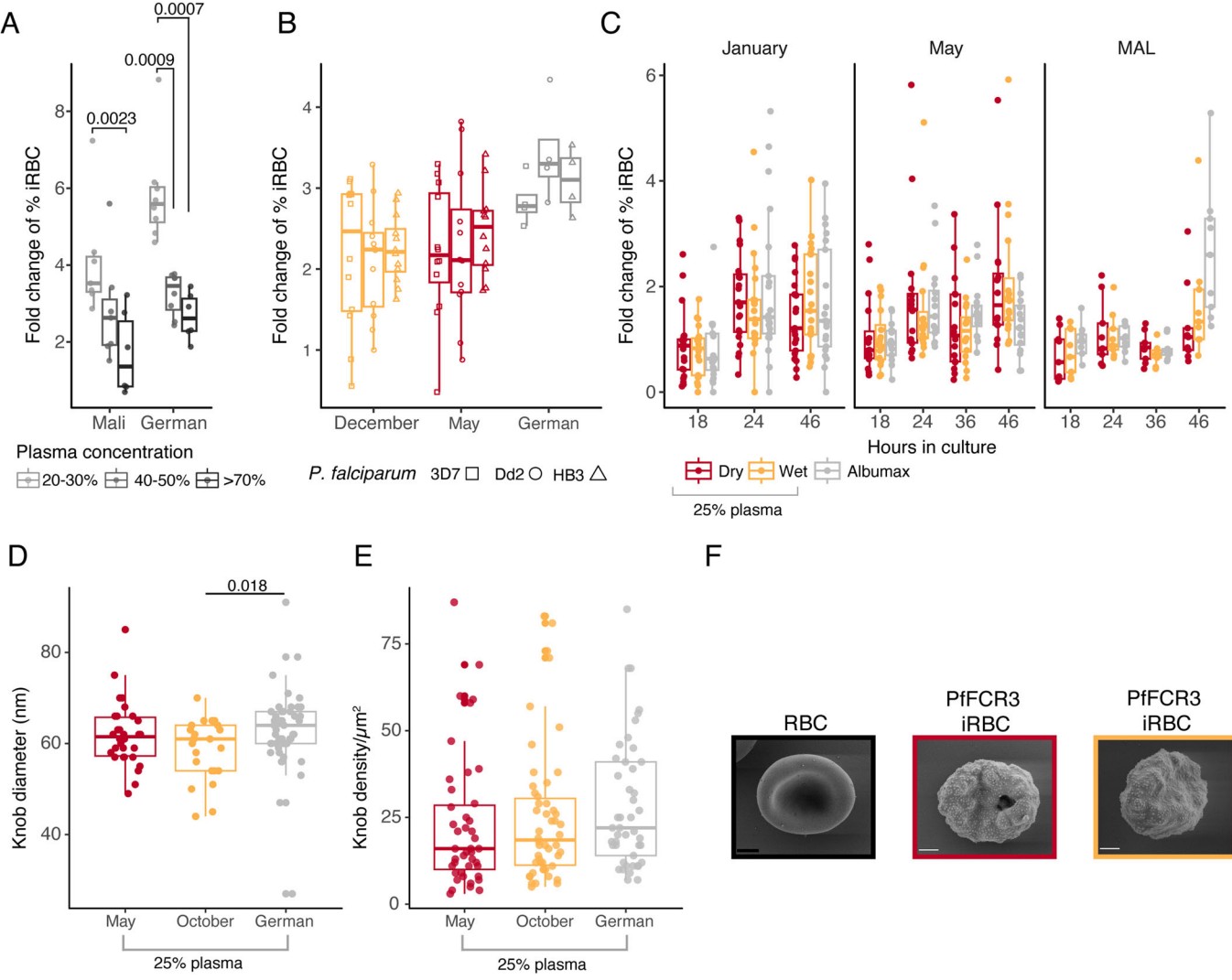

**Figure 3. Dry and wet season plasma have a similar effect on *P. falciparum* in vitro.**

(A) *P. falciparum* 3D7 parasitaemia increased after 48 h in vitro culture in low glucose (5.5 mM) RPMI supplemented with 20–30%, 40–50% or > 70% plasma from uninfected Malian (*n* = 7) or German (*n* = 8) donors. (B) Parasitaemia fold change of lab-adapted 3D7, Dd2 and HB3 *P. falciparum* strains, after 60 h in vitro culture supplemented with 25% plasma from Malian (*n* = 12) or German (*n* = 4) donors. (C) Parasitaemia fold change at 18, 24, 36, and/or 46 h after culture of parasites collected from children at different times during the dry season (Jan, *n* = 21), (May, *n* = 16) or at the first malaria case during the transmission season (MAL, *n* = 9), supplemented with dry or wet season antibody-depleted plasma. (D) Knob diameter measured by transmission electron microscopy and (E) Knob density per μm² of RBC measured by scanning electron microscopy of *P. falciparum* FCR3 parasites (May, *n* = 30; Oct, *n* = 24; German, *n* = 50) after 48 h in vitro culture supplemented with dry or wet season antibody-depleted plasma pooled from 27 Malian, or from four German donors. All boxplots indicate median ± IQR with all individual values plotted. Fold changes are defined as %iRBC t(*n*)/%iRBC t(*n* − 1). Kruskal–Wallis and Wilcoxon pairwise test. (F) Representative example of uninfected RBC (left), and *P. falciparum* FCR3 iRBCs after 48 h in vitro culture supplemented with dry (middle) or wet (right) season antibody-depleted plasma. Scale bar, 1 μm. Source data are available online for this figure.

a *P. falciparum* allele from the end of the dry season until their first malaria episode, 12 had also newly transmitted *P. falciparum* alleles, making it challenging to determine the one promoting the malaria symptoms. Of note, the *ama1* haplotypes found in the three individuals who only had alleles at their first malaria episode that were already present at the end of the previous dry season, were among the most frequently observed haplotypes in the cohort (Fig. 4B). In fact, in the 15 children with at least one shared allele between the end of the dry season and the first malaria episode in the wet season, there were 20 clonal infections, composed of 9

unique highly frequent *ama1* alleles (Fig. 4B). Also of note, although not statistically significant, clinical malaria cases occurring in the presence of a shared dry season allele often happened earlier in the transmission season than clinical malaria cases with only newly transmitted alleles (Fig. 4C).

To complement the single-locus analysis, we genotyped the paired samples from the end of the dry season and first malaria episode of the 15 individuals with shared *ama1* alleles between the two timepoints, using four highly polymorphic *P. falciparum* antigens *csp*, *ama1*, *sera2* and *trap* (4CAST) (LaVerriere et al,

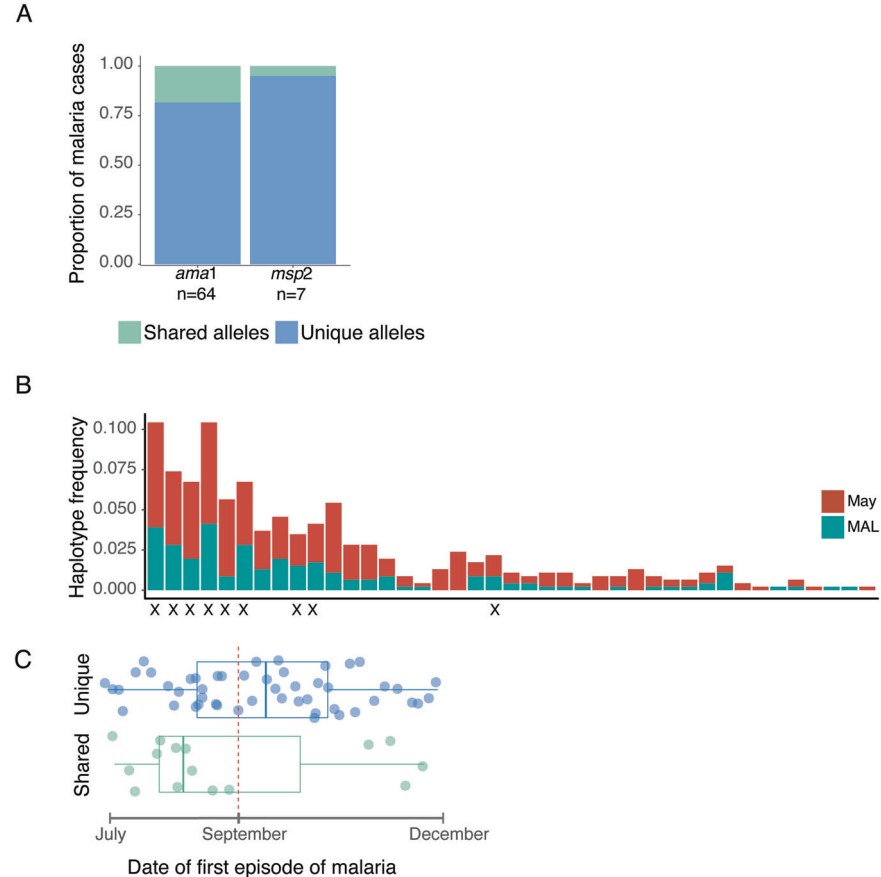

**Figure 4. Clinical malaria episodes are caused by newly transmitted *P. falciparum* parasites.**

(A) Proportion of clinical malaria cases presenting only newly transmitted *P. falciparum* (unique alleles, blue), or presenting at least one allele persisting since the dry season (shared alleles, green), of children that carried *P. falciparum* asymptomatically in the preceding wet season. Alleles were defined by *ama1* amplicon sequencing (*n* = 64) or *msp2* capillary electrophoresis (*n* = 7). (B) Frequency of 41 *ama1* alleles present at the end dry season (May), and during first clinical malaria (MAL) in paired individuals. Alleles shared between the two sampling times across the 64 children are marked with X. (C) Date of clinical malaria cases occurring in children presenting only newly transmitted *P. falciparum* alleles (blue, *n* = 49), or presenting at least one allele persisting since the dry season (green, *n* = 15). Chi-squared test, *p* = 0.08. Source data are available online for this figure.

2022). We obtained a lower number of reads per gene per sample compared to *ama1* alone, with a more uneven distribution of reads across samples (Appendix Fig. S1A), likely in part because we did not adjust PCR cycles for this method based on parasitaemia thresholds, whilst our method for *ama1* alone has been optimised to do so (LaVerriere et al, 2022). Among the 4CAST loci, *ama1* and *sera2* produced higher read numbers compared to *csp* or *trap*, and *ama1* was the most diverse locus (Appendix Fig. S1B). Of note, there was concordance between the most frequent *ama1* haplotypes obtained with both methods, with moderate and statistically significant correlations between the frequencies of the shared haplotypes for both 4CAST runs performed (Pearson correlation, *p* = 0.002 and *p* = 0.00021, Appendix Fig. S1C), and only low-frequency haplotypes were only detected with the 4CAST (Appendix Fig. S1D). The 4CAST analysis confirmed shared *ama1* haplotypes in nine of the 15 individuals, but only three individuals also shared *sera2* haplotypes, suggesting, within the limit of detection of the approaches used, that primarily newly transmitted infections lead to clinical malaria (Appendix Fig. S1E).

## iRBCs of asymptomatic carriers in the wet season share features with those of the dry season

Next, we asked how asymptomatic infections at the end of the dry season compared to asymptomatic infections in the wet season. We analysed *P. falciparum* parasites from asymptomatic RDT⁺ children in the wet season, in October (Oct), and compared it with our previously reported data of RDT⁺ asymptomatic children at the beginning (Jan), middle (Mar) and end (May) of the dry season, and of parasites from clinical malaria case in the wet season (MAL) (Andrade et al, 2020). Using flow cytometry, we determined parasitaemia in 67 asymptomatic RDT⁺ individuals in the wet season (Oct), and observed a broad range of parasitaemia, overlapping with values seen both in malaria cases and in dry season samples. Parasitaemias in asymptomatic wet season carriers (Oct) were significantly lower than those of clinical cases in the wet season, but significantly higher than parasitaemias seen in dry season infections (Fig. 5A,B). The percentage of circulating ring-stage parasites seen in asymptomatic carriers in the wet season (Oct) was significantly lower than in malaria cases (MAL), but

significantly higher than observed in dry season samples (Fig. 5C). As parasitaemia appears increasingly higher from asymptomatic infections in the dry season to asymptomatic infections in the wet season, also does the percentage of ring-stages observed, plateauing at nearly 100% ring-stages in malaria cases in the wet season (Fig. 5D), and significant positive correlations were found across all samples ($R = 0.56$, $p < 2.2e − 16$), and in the dry ($R = 0.47$, $p = 1.2e − 09$) and wet season ($R = 0.54$, $p = 2.8e − 06$) asymptomatic samples, while the narrow range of ~100% of ring-stages observed in malaria cases samples did not allow the detection of a significant correlation ($R = − 0.3$, $p = 0.065$) (Fig. 5D). Next, we determined how different proportions of ring-stages observed throughout the year affected replication in vitro. We cultured freshly isolated *P. falciparum* iRBCs from RDT[+] individuals in the wet season (Oct) for 48 h, and measured parasitaemia at 0, 16, 24, 30, 36 and 48 h by flow cytometry. Data showed that the highest increase in parasitaemia between any two timepoints of the short-term culture in October was 2.985-fold, 95% confidence interval (CI) (1.864, 4.119). This increase was comparable to what we observed in collections during the dry season (2.671 95% CI (2.197, 3.005)), and in malaria cases (2.229 95% CI (2.000,2.826)) (Andrade et al, 2020) (Fig. 5E). Similar to our previous findings in dry season samples ((Andrade et al, 2020) and Fig. 3C), we observed that wet season asymptomatic infections (Oct) could increase parasitaemia as early as 16 or 24 h of culture, while *P. falciparum* from malaria cases in the wet season required more than 40 h to do so (Fig. 5E,F). The time in culture at which *P. falciparum* of RDT[+] individuals in the wet season (Oct) showed the highest increase in parasitaemia was 29.91 h, 95% CI (26.99, 32.83), resembling values reported in the dry season (Jan, 26.80 h, 95% CI (23.48, 30.11); May, 25.62 h, 95% CI (23.23, 28.00)); and significantly lower from those observed during culture of parasites from malaria cases in the wet season (MAL, 44.08 h, 95% CI (41.09, 47.07)) (Andrade et al, 2020) (Fig. 5F). This observation was confirmed by the mean fold change and Loess regression model over equally spaced time intervals of the different sample groups (Appendix Fig. S2).

The time of highest increase was maximum when more than 92% of circulating parasites were in early stages, and almost the full 48 h of the intra-erythrocytic cycle were required to increase parasitaemia. On the other hand, when less than 90% of the parasites were ring-stages, shorter times were needed to significantly increase parasitaemia (Fig. 5G). In the dry season, the vast majority of asymptomatic carriers showed less than 90% rings, and many individuals presented less than 75% of early developmental stages, while children presenting with clinical malaria mostly had over 95% of the ring-stages (Fig. 5G), and asymptomatic carriers in the wet season (Oct) were in between with some individuals presenting over 95% of rings and others below 80%. Confirming these observations, images of thick blood smears obtained at the time of the blood draw in October, revealed larger ring- and trophozoite-stages of *P. falciparum* in asymptomatic infections (Fig. 5H). Altogether, these findings highlight that although not as prominent as in the dry season, extended circulation of iRBCs is also observed in asymptomatic infections during the transmission season, further supporting that a sensing mechanism may not be involved in inducing the persistent phenotype seen in asymptomatic infections during the dry season.

## Timing and duration of infection associated with dry season persistence

The broad range of parasitaemia, and differences of ring-stage proportion observed in asymptomatic individuals during the wet season (Oct) compared to the dry season, and the decrease of parasitaemia observed during the dry season (Jan to May) suggest that within an infection, not all parasites will reach the persistent state required to bridge two wet seasons. This, together with the lack of evidence of a clear sensing mechanism to define when infections may be more or less likely to cause disease, supports the hypothesis that the duration of infection or the number of replication cycles of a *P. falciparum* clone, starting from its transmission from mosquitoes, underlies the seasonality of clinical symptoms; where all parasites have the potential to cause malaria shortly after transmission but progressively transition to a persistent, low parasitaemia state. To test this hypothesis, we first investigated the duration of infection in asymptomatic individuals at the end of the dry season (May) who remained *P. falciparum*[+] without showing signs of disease or naturally clearing the infection. We identified 92 children aged 6–13 years old who were PCR[+] at the end of the dry season in May, and we analysed retrospectively filter paper samples collected biweekly during the preceding wet season at 12 timepoints between December and the preceding July, as well as at the end of the previous dry season, or until each individual was found PCR[−] in four consecutive timepoints, or received malaria treatment (Fig. 6A). Data showed a median duration of infection of 254 days in asymptomatic carriers at the end of the dry season, with a minimum of 144 and a maximum of 364 days consecutively *P. falciparum*[+] PCR, independent of participant sex or haemoglobin type (Fig. EV4A,B). Next, we quantified parasitaemia of all PCR[+] samples by qPCR of the *var* gene acidic terminal sequence (*var*ATS) (Hofmann et al, 2015) and performed *ama1* amplicon sequencing (Miller et al, 2017) on all *var*ATS qPCR[+] samples to determine the *ama1* sequence and longevity of individual *P. falciparum* clones present in each individual over time. Among the 92 children, allowing for up to two base-pairs differences within a cluster, we identified 47 unique clones in 2677 infections across all individuals and timepoints. The complexity of infection (COI), measured as the number of different *P. falciparum* clones carried by an individual, remained stable throughout the year, with most individuals carrying between 3 and 7 different clones, and a median COI of 3 (95% CI (2,3)) (Fig. EV4C). To determine the longevity of each clone within an individual, we established that within a series of timepoints with a particular clone present, one or two consecutive timepoints could be skipped due to parasite sequestration or detection limit (Nkhoma et al, 2018). Allowing the two skips increased the median length of infection by 26.42 days (95% CI (18.53,24.62)), but did not significantly differ when three or more skips were allowed (median difference 6.2 days, 95% CI (−2.29,14.84)) (Fig. EV4D). Among the 92 children analysed, we identified 844 *P. falciparum* infections, 269 of which persisted until the end of the dry season (Fig. 6B; Dataset EV6). At the population level, we observed that most of the 47 *ama1* haplotypes were found amongst clones that persisted until the end of the dry season, and also amongst clones that were cleared prior to the end of the wet season, indicating that no particular *ama1* haplotype is more prone to persist than others (Venn diagram, Fig. 6B). The median length of clonal infections

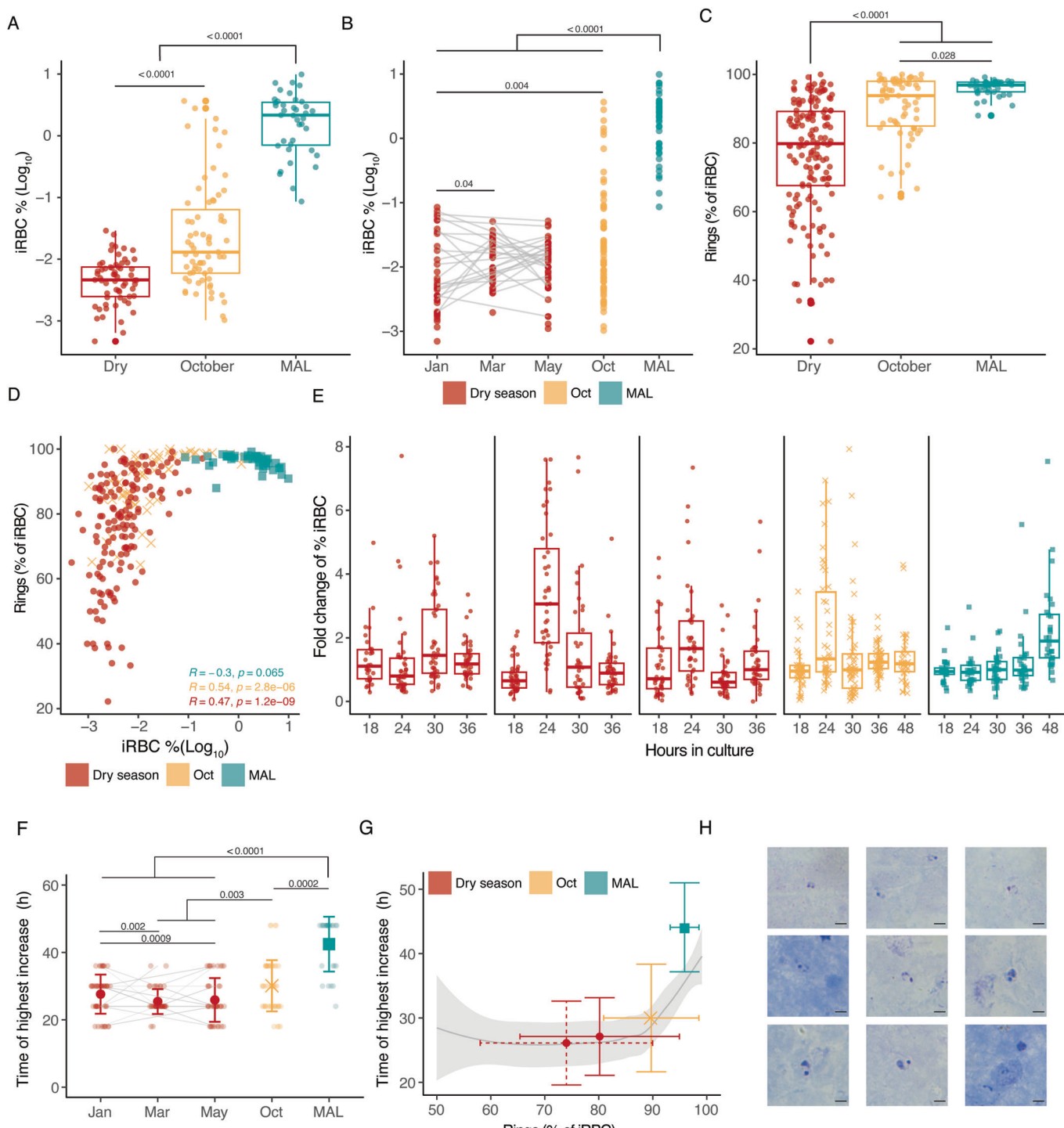

was 49 days, 95% CI (48, 56). Infections lasting fewer days than the median length of all infections in the cohort were defined as short (<49 d), and infections longer than the dry season (129 d) were defined as long (>129 d). Short infections were more frequently seen than long ones (407 short vs 277 long). Infections persisting until the end of the dry season had a median length of 178 days, 95% C.I (174, 188), and the longest infection detected at the end of the dry season had persisted for at least 313 days

(Figs. 6B and EV4E,F). Infection length varied according to the children's age; younger children (6–9 years) had longer infections than older ones (10–13 years) (Fig. 6C), and were likely to have more *ama1* clones persisting until the end of the dry season than the older age group (χ², *p* = 0.0056) (Fig. 6D). Furthermore, we performed a Kaplan–Meier survival analysis and found a statistically significant association between the probability of a clone surviving until the end of the dry season in the younger age

**Figure 5. iRBCs of asymptomatic carriers in the wet season share features with those of the dry season.**

(A) Parasitaemia was detected by flow cytometry in asymptomatic individuals in the dry season (Dry, $n = 108$), asymptomatic individuals in transmission season (Oct, $n = 67$), and individuals presenting their first clinical malaria episode (MAL, $n = 39$) in the wet season. (Kruskal–Wallis and Wilcoxon pairwise test, $p = 1.21e{-}9$ between October and dry season). (B) Parasitaemia was measured by flow cytometry at three different timepoints during the dry season with lines indicating individuals with at least two paired timepoints ($n = 80$), with unpaired October ($n = 67$) and MAL ($n = 39$) (Kruskal–Wallis and Wilcoxon pairwise test). (C) Percentage of circulating ring-stage iRBCs detected by flow cytometry in asymptomatic individuals carrying *P. falciparum* in the dry (dry season, $n = 108$), or the wet (Oct, $n = 67$) season, and during first clinical malaria and in the wet season (MAL, $n = 39$). (Kruskal–Wallis and Wilcoxon pairwise test, $p = 8.28e{-}11$ between October and dry season). (D) Relation between percentage of circulating ring-stage and parasitaemia determined by flow cytometry in asymptomatic individuals carrying *P. falciparum* in the dry (dry season, $n = 108$), or the wet (Oct, $n = 67$) season, and during first clinical malaria in the wet season (MAL, $n = 39$). Pearson correlation ($p$ value for Dry, October and MAL are 1.2e$^{-9}$, 2.8e$^{-6}$, 0.065, respectively). (E) Parasitaemia fold change at 16, 24, 30, 36 and 48 h after culture of parasites collected from children during the dry (Jan, $n = 53$), (Mar, $n = 42$) and (May, $n = 41$), and the wet (Oct, $n = 54$) season, and at the first clinical malaria episode in the wet season (MAL, $n = 26$). Fold changes are defined as % iRBC t($n$)/%iRBC t($n - 1$). All boxplots indicate median ± IQR with all individual values plotted. (F) Time of highest increase in parasitaemia detected during in vitro culture of *P. falciparum* parasites from children with at least two paired asymptomatic samples throughout the dry season ($n = 38$), and parasites from children carrying asymptomatic (Oct, $n = 54$) and symptomatic (MAL, $n = 26$) *P. falciparum* in the wet season. Dot corresponds to the average time of increase ± SD (Pairwise Fisher exact test between months and highest time of increase above or below 24 h, $p$ values and significant levels are shown). (G) Relation between the average time of highest increase in vitro and proportions of ring-stage parasites collected from asymptomatic individuals in the beginning (Jan, red solid SD bars) and end (May, red dashed SD bar) of the dry season, during asymptomatic infections the wet season (Oct, blue SD bars), and during the first clinical malaria episode (MAL, aqua SD bars). The shaded curve indicates the Loess regression model with a 95% confidence interval. (H) Giemsa-stained thick blood films of *P. falciparum* parasites were collected directly from the arms of nine asymptomatic individuals carrying *P. falciparum* during the wet season (Oct). Young and late rings are visible in the top panels, and late rings to trophozoite stages are visible in the mid to bottom panels. Scale bar, 5 µm. Source data are available online for this figure.

group (7–9 years) (Fig. EV4G). Lastly, clonal infection length measured by *msp2* capillary electrophoresis from the beginning of the wet season onwards in 51 children that did or did not keep asymptomatic infections into the ensuing dry season, showed that most clonal infections were indeed short (Fig. EV5A; Dataset EV7); and highlighted a trend of longer clonal infections occurring in older rather than younger children (Fig. EV5B).

With *ama1* data from all children remaining PCR⁺ for eight or more timepoints ($n = 52$), we questioned whether survival of a clonal infection until the end of the dry season was affected by the timing of the beginning of the infection in the preceding wet season (Fig. 6E). We determined the likelihood of parasites transmitted early (>84 days) or late (<84 days) in the wet season becoming persistent until the end of the dry season, and observed that clones transmitted after the middle of the transmission season were more likely to survive until the end of the dry season (Poisson generalised linear mixed effects model, odds ratio = 3.32 and $p$ value = 5.27e–9, Fig. EV4H).

Indeed, time of first appearance of a *P. falciparum* clone seems to affect its ability to survive the dry season and reach the ensuing transmission season; and while early transmission allows for very long infections, we observe that later appearances also become progressively longer as they more often bridge two wet seasons (Fig. EV5C).

## Discussion

Malaria parasites rely on a reservoir of persistent infections to bridge two wet seasons that may be several months apart. Older children and adolescents, rather than very young children or adults, carry most of this infectious reservoir (Andrade et al, 2020; Portugal et al, 2017; Tiedje et al, 2017), but the factors determining parasite survival through the dry months remain poorly understood. In this study, we investigated whether a sensing or timing mechanism could support silent parasite carriage in ~20% of children and the absence of disease during the dry season (Andrade et al, 2020; Crompton et al, 2008; Males et al, 2008; Portugal et al,

2017), while clinical malaria cases appear restricted to the wet season. Previous research has highlighted the ability of malaria parasites to detect cues from their host or in vitro surroundings and adjust growth, replication or sexual conversion accordingly (Billker et al, 1998; Brancucci et al, 2017; Campelo Morillo et al, 2022; Filarsky et al, 2018; Hegge et al, 2010; Kappe et al, 2003; Mancio-Silva et al, 2017; Marreiros et al, 2023; Meireles et al, 2020; Meireles, Mendes, et al, 2017; Meireles, Sales-Dias, et al, 2017; Ripp et al, 2021; Tintó-Font et al, 2021). Seasonal variations in host factors such as blood haemodynamic, glucose levels, enzymes, proteins (Radke and Izzo, 2010; Röcker et al, 1980), and hormones (Bjørnerem et al, 2006; Land et al, 2005) have also been described. We tested a possible mechanism by which the parasite could sense seasonality and found decreased levels of several amino acids and phospholipids during clinical malaria cases, but detected only minimal differences in children's plasma composition between the dry and wet season, regardless of asymptomatic *P. falciparum* infection (Fig. 2). Additionally, we observed no differences in replication or host-cell remodelling when supplementing in vitro cultures with plasma from different seasons (Fig. 3). Persisting clones present at end of the dry season were unlikely to drive disease in their hosts despite the resuming of the wet season, with most clinical malaria episodes being associated with newly transmitted parasites in the wet season (Fig. 4). In this study, we did not quantify gametocyte carriage, but we acknowledge the critical importance of sexual stages in resuming transmission as the rainy season returns. Altering transmission investment could be a mechanism to slow parasite growth. However, gametocytes have been shown not to survive beyond 20 days (Smalley et al, 1977; Gebru et al, 2017), hence maintaining asexual replication during the dry season while continuously promoting the differentiation of a small proportion of intra-erythrocytic asexual parasites into sexual development (Kafsak et al, 2014) may be the best way to ensure the presence of gametocytes when mosquitoes return. Accordingly, data from *Plasmodium* species infecting humans and birds have shown increased gametocytes at higher asexual parasitaemias (Bruce et al, 1990; Cornet et al, 2014; Salgado et al, 2021).

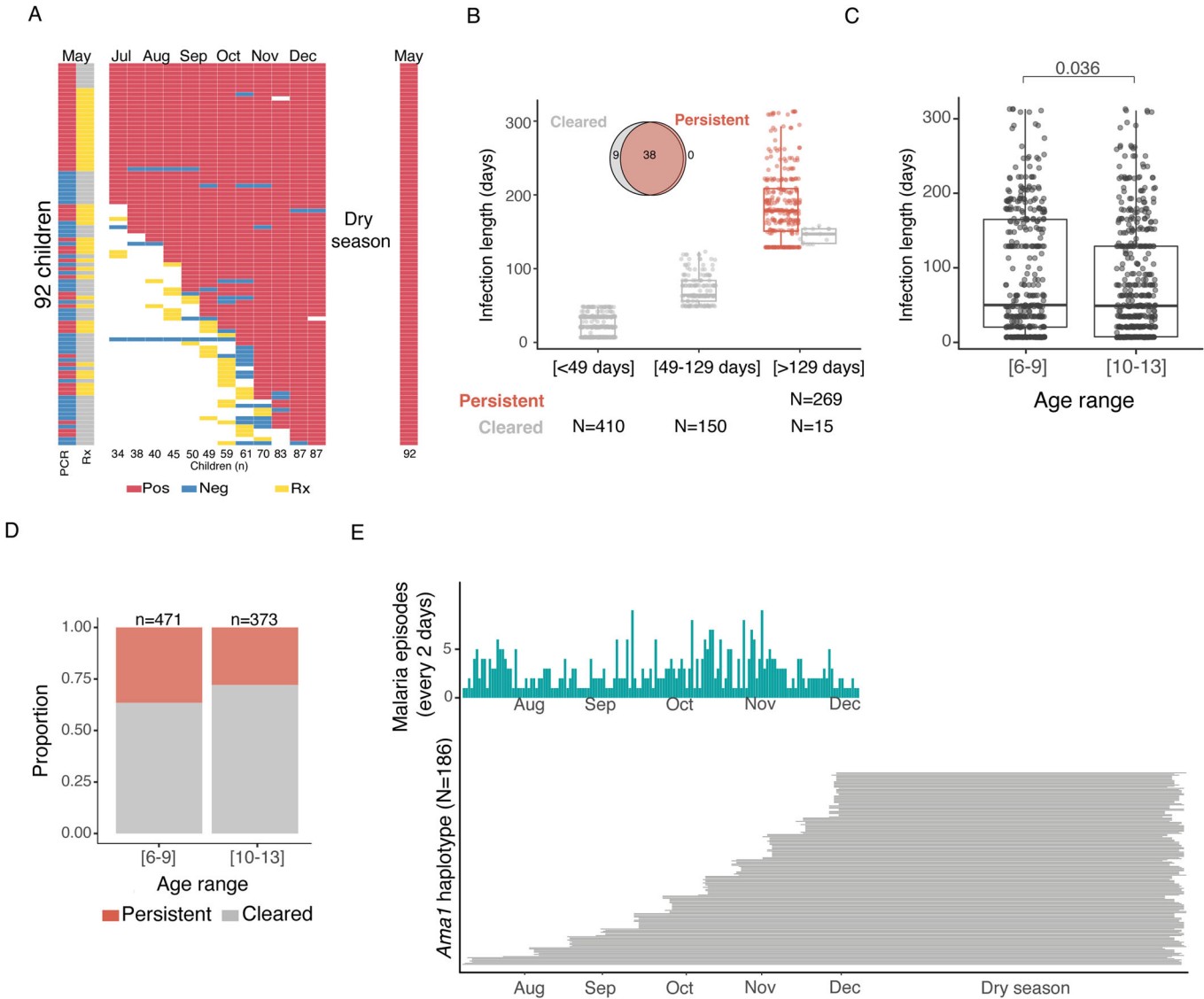

**Figure 6. Timing and duration of infection associated with dry season persistence.**

(A) Retrospective *P. falciparum* PCR results determined every 2 weeks between January 2013 and July 2012 of 92 asymptomatic children found PCR positive (red) in May 2013 at the end of the dry season. PCR was performed until individuals became PCR negative (blue) for four consecutive timepoints, received malaria treatment (yellow), or until the end of the preceding dry season (May 2012). Each column represents a collection timepoint. The first column shows *P. falciparum* PCR result in May 2012, and the secnd whether treatment occurred (yellow) or not (grey), prior to the first timepoint in July 2012. (B) Clonal infection length in persistent (brick, n = 269) or cleared (grey, n = 575) *P. falciparum* clones, grouped into short (<49 d), intermediate (49–129 d) or long (>129 d), allowing up to two consecutively skipped timepoints within a series of positive for a particular clone. Venn diagram of *ama1* sequences found in persisting and/or cleared clones by the end of the dry season. All boxplots indicate median ± IQR with all individual values plotted. (C) Infection length of clones from younger (6–9, n = 471) or older (10–13, n = 373) children. Wilcoxon rank-sum test (p = 0.036). (D) Proportion of persistent (brick) and cleared (grey) clones found throughout the year in younger (6–9, n = 471) or older (10–13, n = 373) children who were PCR positive in May 2013 ($\chi^2$ test, p = 0.0056). (E) Length of individual *ama1* haplotypes (n = 186) present at the end of the dry season in May 2013 in children (n = 52) remaining PCR positive >8 timepoints (Poisson GLMM, p value = 5.27e-9). The histogram illustrates the frequency of clinical malaria in the full cohort of 595 individuals aged 1–45 during the 2012 wet season.

Our serological analyses may not have fully captured variation in low-abundant plasma analytes, as they often fall outside the filtering criteria; and a larger sample size could have allowed detecting further differences that we may not have been powered to do. Furthermore, the absence of in vitro evidence of a sensing mechanism may be attributed to the inability to fully reproduce the seasonal environment in vitro. Nevertheless, the observation that dry and wet season parasites in asymptomatic children share similarities (Fig. 5), supports that seasonal sensing is not necessary for the extended circulation of iRBCs in asymptomatic infections.

Our data confirmed that circulating parasites in malaria cases were largely young rings, requiring nearly 48 h to increase parasitaemia in vitro. In contrast, dry season *P. falciparum* exhibited earlier increases in parasitaemia (Fig. 3C), due to the circulation of further developed rings and trophozoites (Andrade et al, 2020). Furthermore, we now demonstrated that compared to

persistent infections during the dry season, and clinical malaria cases in the wet season, asymptomatic infections in the wet season have intermediate levels of both parasitaemia and proportion of ring-stages, often requiring a similarly low number of hours as dry season infections to complete the intra-erythrocytic cycle and increase parasitaemia (Fig. 5E). This suggests that wet season asymptomatic carriers may harbour a mixture of *P. falciparum* parasites, some of which may or may not survive through the dry season, including clones yet to be cleared or transition to the dry season state. These data support that circulation time and cytoadhesion dynamics of iRBCs are a major forces determining the clinical prognosis of malaria, as highlighted in our previous study in different transmission settings and clinical presentations (Thomson-Luque et al, 2021).

Alternatively, or in addition, the variability seen in the proportion of ring stages in vivo may also be attributed to a faster population expansion of parasites in malaria cases compared to asymptomatic infections, leading to a bias towards earlier developmental stages in clinical malaria cases (Khoury et al 2014; Greischar et al, 2019).

Another possibility would be the requirement of fever to synchronise parasites (Kwiatkowski, 1989) and loss of synchrony in its absence. However, periodic fevers themselves have been proposed to result from synchronous schizogony, and previous work has suggested that febrile temperatures induce cytoadherence of ring-stage *P. falciparum*-iRBCs (Udomsangpetch et al, 2002). Furthermore, although studies are scarce, asymptomatic *P. falciparum* infections also reveal a degree of inherently synchronous subpopulations of parasites, despite the absence of fluctuating temperature (Farnert et al, 1997). Even if there were complete synchrony in symptomatic infections and no synchrony at all in asymptomatic infections, the act of sampling individuals at different infection stages, based on clinical presentation rather than the timing of their parasitaemias, would result in a well-mixed distribution of parasite ages across different samples, truncated by cytoadhesion, as we suggest.

Following *P. falciparum* clones from the wet into the dry season evidenced that malaria transmission in this area of Mali is very high, and most clones result in short-lived infections. In fact, less than 30% of clones transmitted to children persist and contribute to the dry season reservoir observed (Fig. 6B,C). This aligns with modelling data from Uganda and Ghana, estimating that many asymptomatic infections are rapidly cleared regardless of age (Andolina et al, 2023; Bretscher et al, 2015). Our data showed that parasites surviving the dry season were largely transmitted late in the preceding wet season (Fig. 6), suggesting that the duration of infection plays a key role in determining clinical presentation and the composition of the dry season reservoir for transmission. Furthermore, we show that older children (10–13 years) have shorter infections compared to younger children (6–9 years), and that the proportion of clones that persist throughout the entire dry season is higher in the younger age group, indicating that individuals with more developed immunity against malaria more efficiently clear some clones (Dent et al, 2008).

Similar to previous reports (Andrade et al, 2020; Babiker et al, 2000; Sondén et al, 2015), we observed, at the parasite population level, that the vast majority of the *ama1* haplotypes found persisting until the end of the dry season in some individuals, were found to clear prior to the end of the wet season in other individuals

(Figs. 4C and 6B). This supports, albeit the limited number of loci analysed, that no particular parasite genotype is inherently more capable of surviving the dry season, but rather that all parasites have the potential to cause malaria shortly after transmission and progressively transition to the persistent, low parasitaemia state. Such a mechanism would avoid the yearly winnowing-down of the faster-growing genetic variants in infections over time, leaving at the end of the dry season only the variants with poor ability to sequester or cause clinical disease, which would require de novo generation of parasite variants causing clinical malaria every year as the transmission season restarts.

The selective pressure regulating circulation time of iRBCs leading to acute infections when mosquitoes are present and to asymptomatic carriage during the dry season is not known. We hypothesise that it may be related to acquired immunity, even if asymptomatic infections do not seem to elicit detectable inflammation, affect immune cell function, or prevent antibody loss during the dry season compared to negative individuals (Andrade et al, 2020). Antibodies targeting *P. falciparum* erythrocyte membrane protein 1 (PfEMP1) and other variant surface antigens (VSAs) that promote cytoadhesion of iRBCs may be critical to parasite silent persistence (Chan et al, 2012). The *var* gene family encodes PfEMP1, which displays a monoallelic expression of ~60 variants (Chan et al, 2012). PfEMP1s have been associated with severe malaria (Jespersen et al, 2016; Kaestli et al, 2006; Lavstsen et al, 2012; Storm et al, 2019; Storm and Craig, 2014; Turner et al, 2013), and the acquisition of antibodies that bind to and hinder PfEMP1-mediated cytoadhesion has been shown to contribute to malaria immunity (David et al, 1983). Sequential acquisition of anti-PfEMP1 antibodies may begin targeting more pathogenic variants (Cham et al, 2010; Obeng-Adjei et al, 2020), aligned with the observation of quick acquisition of protection from severe malaria after a few clinical episodes (Gupta et al, 1999), and progress to extend infection length (Klein et al, 2014).

Longer exposure and more acute episodes leading to both strain-specific and cross-strain responses will eventually lead to predominantly avirulent infections in clinically-immune adolescents and adults (Marsh and Kinyanjui, 2006; Miller et al, 1994; Tran et al, 2013; Weiss et al, 2010), in line with the observed proportions of persistent vs cleared infections in different age groups (Fig. 6D). Accordingly, antibodies targeting PfEMP1s of pathogenic variants are acquired before those targeting less efficient growers (Miller et al, 1994), resulting in older individuals more often harbouring parasites with poor endothelial binding capacity and low ability to increase parasitaemia or induce malaria symptoms (Andrade et al, 2020; Thomson-Luque et al, 2021). These may express PfEMP1s that cytoadhere poorly and are poor immunogens, or that may simply express lower amounts of PfEMP1 and other VSAs and fail to efficiently bind or be cleared by immunity. Further studies are required to investigate how the progressive acquisition of antibodies against monoclonal *P. falciparum* infections impacts *var* gene expression, circulation time and cytoadhesive properties of iRBCs. In a recent study using the intestinal parasite *Giardia lamblia* model, it was demonstrated that a low concentration of antibodies targeting the expressed variant surface antigen trigger the switching and emergence of different and seemingly random variant surface antigens (Tenaglia et al, 2023). It is possible that during the wet season, PfEMP1-specific antibodies induced by increasing *P. falciparum* parasitaemia stimulate *var* gene switching,

while low parasite levels in the dry season may be insufficient to do so. In fact, *P. falciparum*-specific antibody levels have been shown to decrease during the dry season (Akpogheneta et al, 2008; Andrade et al, 2020; Portugal et al, 2017), possibly avoiding exhaustion of the *var* gene repertoire and resolving the apparent incompatibility between high switch rates and long-lasting infections (Childs and Buckee, 2015; Nyarko and Claessens, 2021). Interestingly, *var* gene switching was recently shown to respond to disruptions in *S*-adenosylmethionine (SAM) metabolism (Schneider et al, 2023). While we observed no seasonal pattern of methionine or SAM-related metabolites, we cannot exclude that such a mechanism might affect switching of PfEMP1 in asymptomatic infections.

Understanding the seasonal patterns affecting the survival length of *P. falciparum* infection and clonal persistence will be essential in guiding elimination efforts targeting the silent reservoirs of malaria. Our data provides evidence of a relation between time since parasite transmission and potential of dry season persistence, and highlights the need to study the complex interactions between the parasite, host and environment.

# Methods

## Study individuals and ethical approval

From 2011 to 2020, ~600 individuals aged 3 months to 45 years were followed in a longitudinal cohort in the village of Kalifabougou, Mali, as previously described (Portugal et al, 2017; Tran et al, 2013). Individuals were age-stratified, and randomly selected (computer-generated randomisation) from the previously collected census data who meet the eligibility criteria and have provided written informed consent or assent (including parental consent) were enroled in the study. The study was approved by the Ethics Committee of Heidelberg University Hospital; the Faculty of Medicine, Pharmacy and Odontostomatology at the University of Bamako; and the National Institute of Allergy and Infectious Diseases of the National Institutes of Health Institutional Review Board, and is registered at ClinicalTrials.gov (identifier NCT01322581). Written informed consent was obtained from all study participants or their parents/guardians to acquire samples and clinical data. Exclusion criteria at enrolment were: haemoglobin <7 g/dl, axillary temperature ≥37.5 °C, acute systemic illness or use of antimalarial or immunosuppressive medications in the preceding 30 days. Clinical malaria cases were defined by axillary temperature ≥37.5 °C, ≥2500 asexual parasites per µl of blood and no other apparent cause of fever. Clinical malaria cases were treated with a 3-day course of artemether and lumefantrine, according to national guidelines. All experiments conformed to the principles set out in the WMA Declaration of Helsinki and the Department of Health and Human Services Belmont Report.

## Sample collection

Samples used in this work were acquired during cross-sectional visits and through passive surveillance of clinical malaria episodes. During cross-sectional scheduled visits (January, March, May and October) and at their first malaria episode in the transmission season, DBSs on filter paper (Protein Saver 903, Whatman), thick blood smear and venous blood (4 or 8 ml depending on whether donor age was below or above 4 years) were collected. Venous blood samples were drawn by venepuncture and collected in sodium citrate-containing cell preparation tubes (Vacutainer CPT Tubes, BD) and transported to the laboratory. Plasma and RBC pellets were separated by centrifugation and stored at −80 °C or used immediately (*P. falciparum* short-term culture). Plasma used for metabolomic and lipidomic analysis experiments were immediately frozen in liquid $N_2$ after separation by centrifugation. Additionally, during the transmission season of 2012, DBSs on filter paper were collected at the end of the dry season in May 2012, every two weeks from July, 2012 until December 2012, again at the end of the following dry season in May 2013, and at the first clinical malaria of the transmission season of 2013.

## *P. falciparum* detection and quantification

To determine if study participants carried *P. falciparum* infection, *P. falciparum* parasites were detected and counted in Giemsa-stained thick blood smears by light microscopy. Parasites were counted against 300 leucocytes, with an average count of 7500 leucocytes per µl. All smears were evaluated independently by two microscopists, and by a third one in case of any discrepancies. During cross-sectional time-points, *P. falciparum* infections were detected by rapid diagnostic tests (RDT; SD BIOLINE Malaria Ag P.f test of histidine-rich protein II), which has a sensitivity of ~100 parasites per µl (Ratsimbasoa et al, 2008). Nested PCR was performed retrospectively in all samples collected during the cross-sectional time-points from filter papers or frozen RBC pellets. The 18S ribosomal RNA gene of *P. falciparum* was amplified through nested PCR following a previously described protocol with a sensitivity of 0.5– 1 parasite per µl (Tran et al, 2013). Two mm circular punch of a DBS or 2 µl of frozen RBC pellet was used to extract DNA and direct PCR using Phusion Blood Direct PCR Kit (Thermo Fisher Scientific), with the following cycling conditions 98 °C, 5 min, 30 cycles of amplification (98 °C, 1 s, 61 °C, 5 s, and 72 °C, 30 s) and 72 °C, 1 min. A second PCR was performed with 1 µl of product from the first PCR, with the following cycling conditions: 95 °C, 2 min, 30 cycles of amplification (95 °C, 1 min, 58 °C, 1 min, 72 °C, 1 min) and 72 °C, 5 min. *P. falciparum* infection was determined by capillary electrophoresis using LabChip GX Touch HT Nucleic Acid Analyzer (Perkin Elmer) as per manufacturer instructions.

Samples analysed through amplicon sequencing had their DNA extracted by Chelex-Tween from 12 mm punches (Harris Uni-Core 3 mm Whatman) of DBS as described earlier (Teyssier et al, 2021). Parasite quantification was achieved by qRT-PCR targeting the var gene acidic terminal sequence (varATS) following a previously described protocol (Hofmann et al, 2015) with TaqMan Multiplex Master Mix (Applied Biosystems), varATS primers (ATS-fwd: 5′-CCCATACACAACCAAYTGGA-3′, ATS-rev: 5′-TTCGCACA TATCTCTATGTCTATCT-3′) and varATS TaqMan probe (6-FAM-TRTTCCATAAATGGT-NFQ-MGB) and 5 µl of genomic DNA; PCR data was acquired with qTower3 (Analytik Jena) with the following cycling conditions: 2 min at 50 °C, 10 min at 95 °C, 15 s at 95 °C and 1 min at 55 °C for 45 cycles. varATS qPCR further confirmed *P.falciparum* infection status in donor samples including and spanning dry seasons. A standard curve was included in each plate generated from cultured ring-stages of a laboratory-adapted Malian

parasite strain, with a serial dilution from $10^6$ to $10^{-3}$ parasites per µl in which parasite density was calculated based on the RBC count in a haemocytometer. Blood was blotted on a filter paper at 50% haematocrit and extracted following the Chelex-Tween method.

## Metabolomics of human plasma

Plasma samples frozen in liquid $N_2$ were thawed and profiled by Quantitative Targeted Analyses with Ultra-high-performance liquid chromatography/triple quadrupole mass spectrometry (UHPLC-MS/MS) using the MxP Quant 500 kit (Biocrates) following the manufacturer's instructions. Briefly, 10 µl of human plasma blinded samples were added to internal standards and dried under a nitrogen stream with a positive pressure manifold (Waters). About 50 µl of a 5% phenyl isothiocyanate (PITC) solution was added to derivatize amino acids and biogenic amines at RT and dried after 1 h. To extract the metabolites, 300 µl 5 mM ammonium acetate in methanol was added to each filter and incubated for 30 min. The extract was eluted using positive pressure. For further LC-MS/MS analyses 150 µl of the extract was diluted 1:2 in water. For FIA-MS/MS analyses 10 µl extract was diluted 1:50 in FIA solvent (Biocrates) and LC-MS/MS and FIA-MS/MS measurements were performed. For chromatographic separation, an UPLC I-class PLUS (Waters) system was used coupled to a SCIEX QTRAP 6500+ mass spectrometry system in electrospray ionisation (ESI) mode. Data were generated using the Analyst (Sciex) software suite and transferred to the MetIDQ software (Biocrates) for further data processing and analysis. Metabolites were identified by isotopically labelled internal standards and multiple reaction monitoring (MRM) using optimised MS conditions. For quantification, a seven-point calibration curve or one-point calibration was used. From the 531 metabolites detected in 38 individuals, a filtering threshold was applied, and individuals with less than 80% of metabolites detected were excluded. Additionally, metabolites absent in <90% of individuals were also excluded. This resulted in a table of 35 individuals and 456 metabolites. Metabolite abundance was normalised to reduce the impact of outliers by using the Python Sklearn scaler algorithm according to the interquantile ranges (IQR). The final table, after applying the filtering criteria, was used for principal component analysis (PCA). The two first principal components and the variance explained were visualised with a scatter plot. Significant differences between normalised metabolite abundances were determined for each metabolite for each group of samples by using a Student's $t$-test and $p$ values were adjusted using FDR. Metabolites with significantly different concentrations were used as input when choosing the KEGG Homo sapiens as a database and hypergeometric test as the enrichment methods used to identify biological functions that may be over-represented in a group of metabolites more than would be expected by chance and ranks these functions by relevance. Pathway impact reflects centrality of metabolites within a pathway with a significant threshold, the significant pathways were classified as having an adjusted $p$ value (in $-\log 10$ scale) of 1,3 or above with pathway impact >0. Pathway enrichment analysis was performed with R package Metaboanalyst (Xia et al, 2009).

## Phospholipid analysis

Plasma were analysed by Liquid chromatography–mass spectrometry (LC-MS). Samples were blinded, extracted, processed, and analysed at the Lipidomics Unit at the Institute of Physiological Chemistry in Mainz, using the liquid-liquid extraction protocol as described (Lerner et al, 2017). All values were normalised to the analysed plasma volume.

## Proteomics of human plasma samples

Plasma samples were first depleted of the 14 most abundant proteins in human plasma using Seppro IgY 14 (Sigma-Aldrich), following the manufacturer's instructions and blinded for the rest of the analysis. Plasma was diluted in Seppro Dilution buffer and filtered using a Costar Spin-X filtering tube (Corning). After depletion, protein concentration was determined by Pierce BCA Protein Assay kit (Thermo Scientific). About 100 µg of protein was precipitated using methanol and chloroform as previously described (Wessel and Flügge, 1984) and resuspended in Urea buffer (6 M urea, 100 mM NaCl in 50 mM in TEAB buffer). Protein samples were mixed with 10 mM (Tris(2-carboxyethyl)phosphine)) solution (TCEP) and 40 mM Chloroacetamide (CAA) followed by 30 min incubation at RT. ~8.5 pH was confirmed for all samples, and proteinase Lys-C was added to the samples and incubated for 4 h at 37 °C. Urea concentration was adjusted to 1.5 M by the addition of 50 mM Triethylammonium bicarbonate (TEAB). Samples were trypsin digested overnight at 37 °C. Digestion was stopped by adding 0.4% (vol/vol) trifluoroacetic acid (TFA) solution. Below 2 pH was confirmed, and samples were centrifuged (2500×$g$, 10 min) and peptides-containing supernatants were then loaded on SepPak tC18 cartridges at 1 drop/second. The cartridges were placed in new tubes, and peptides were eluted with Acetonitrile (CAN) and TFA mixed solution as 1 drop/second and the final elute was concentrated using a Speedvac centrifuge to get the final dried peptides. Dried peptides were resuspended in the loading buffer. Mass spectrometry data were acquired using Ultimate 3000 UPLC coupled to a Q-Exactive HF-X mass spectrometer (Thermo Fisher Scientific). Raw data files were processed with MaxQuant software (MPI of Biochemistry) for protein identification and quantification. MS/MS spectra were searched against the Homo sapiens or P. falciparum databases. Quantitative normalised ratios were calculated by MaxQuant and used for further data analysis. Perseus software Version 1.6.13 (Tyanova et al, 2016) was used to add Homo sapiens and P. falciparum annotations to the detected protein groups. All statistical analyses were performed with R, where hits of label-free quantitation intensity (LFQ-intensity) were only considered when present in at least three samples in one categorical group. From the 275 proteins detected in 35 individuals, we applied filtering based on missingness for each protein (60%) and for each individual (90%), resulting in 146 proteins and 30 individuals. Data scaling for each protein, as well as hierarchical clustering, were performed using custom R scripts. Principal component analysis (PCA) and tests for significance were performed as described in the metabolomics section.

## P. falciparum culture of laboratory-adapted strains

P. falciparum parasites were cultured in fresh human $O^{Rh+}$ erythrocytes at 5% haematocrit in RPMI 1640 (Gibco) complete medium (with L-glutamine and HEPES), 7.4% sodium bicarbonate (Gibco), 100 µM hypoxanthine (C.C.Pro) and 25 mg ml$^{-1}$ gentamycin (Gibco) added with 0.25% Albumax II (Gibco), at 37 °C either in the

presence of a gas mixture containing 5% $O_2$, 5% $CO_2$ and 90% $N_2$. *P. falciparum* parasites were routinely synchronised to ring-stage parasites with sorbitol following a previously described protocol (Radfar et al, 2009). For tight synchronisation of a 3–4 h window post-invasion, *P. falciparum* parasites were synchronized by magnetic activated cell sorting (MACS) at a late-schizont stage where multinucleated schizonts were purified, allowed to invade for 3–4 h, and selected for recently invaded rings with a second MACS purification. Following synchronisation, young rings were put in culture in fresh human $O^{Rh+}$ erythrocytes at 2% haematocrit supplemented with Albumax II or non-heat inactivated human plasma in 96-well round-bottom plates. Antibodies were depleted from human plasma using Pierce Protein G and Protein L Plus Agarose beads (Thermo Fisher Scientific). Briefly, plasmas were incubated with the same volume of both Protein G and Protein L plus agarose beads for 30 min; antibody-depleted plasmas were collected, and the antibodies were eluted from the beads and discarded. Plasma was incubated with beads for a total of three times to achieve successful depletion. IgG and IgM quantification in the plasma before and after depletion with Ready-SET-Go! ELISA Kit (eBioscience) confirmed the successful depletion of antibodies. ELISA results were acquired on Cytation3 (BioTek) plate reader at 450 nm.

## *P. falciparum* ex vivo culture

RBC pellets collected from CPT tubes, either from RDT+ samples from asymptomatic individuals in cross-sectional time-points (January and May 2019) or malaria cases samples in the ensuing transmission season, were cultured at 2% haematocrit in RPMI 1640 with adjusted 5.5 mM glucose supplemented with 0.25% Albumax II or 25% antibody-depleted (described above) pooled plasma collected in May 2018 (dry season) or October 2017 (transmission season), using the candle jar system method previously described (Trager and Jensen, 1976). *P. falciparum* parasites were grown at 37 °C in a candle jar for 36 to 48 h, until parasitaemias significantly increased. Due to high parasitaemia found in clinical malaria samples, RBCs of malaria case samples were diluted 1:100 in fresh human $O^{Rh+}$ erythrocytes to allow maximum parasite growth.

## *P. falciparum* analysis by flow cytometry

Parasitaemia from freshly collected samples and short-term culture at different time-points was determined by flow cytometry. Parasites were stained with 5x SYBR Green I and 0.6 µM MitoTracker for 30 min at 37 °C. Cells were acquired using FACS Canto II or LSR II (BD) and analysed using FlowJo software 10.2 or higher versions. The gating strategy used to determine %iRBCs is available in Appendix Fig. S4. SYBR Green positive RBCs were counted as iRBCs and MitoTracker was used to assess the viability of trophozoite and schizont-staged parasites (Amaratunga et al, 2014). Parasite development in infected red blood cells was further determined by the geometric mean fluorescence of SYBR Green nuclear staining. Fold change of parasitaemia was determined for each sample as the ratio of %iRBCs (by SYBR Green) at each timepoint over its preceding one. The time of highest increase of parasitaemia was the timepoint at which the fold change of parasitaemia was the highest for each sample. To address technical noise caused by low parasitaemia, and potential measurement artifacts from flow cytometry, we excluded fold changes exceeding 10, which was four times the median absolute deviation across all measurements of the highest increase.

## Electron microscopy

FCR3 *P. falciparum* parasites cultured to the schizont stage were incubated with gelafundin for 15 min at 37 °C to enrich for knob-positive iRBC, and then were allowed to invade fresh RBCs for 3–4 h, after which newly invaded ring-stages were synchronized with sorbitol (Radfar et al, 2009) to eliminate remaining schizonts in culture. After 30–38 h post invasion, iRBCs were cultured with 25% antibody-depleted human plasma in fresh human $O^{Rh+}$ erythrocytes at 2% haematocrit in RPMI 1640 medium without glucose, supplemented with L-glutamine, 25 mM HEPES, 7.4% sodium bicarbonate, 25 mg/ml gentamycin (all Gibco) and 5.5 mM D-glucose (Sigma-Aldrich) at 37 °C in the presence of a gas mixture containing 5% $O_2$, 5% $CO_2$ and 90% $N_2$. Forty-eight hours later, the iRBC pellet was washed and fixed overnight in 2% PFA and 0.016% Glutaraldehyde in 0.1 M Cacodylate buffer, at 4 °C.

For Transmission electron microscopy, iRBCs were pelleted and then embedded in 3% low-melt agarose in 0.1 M Cacodylate buffer. Briefly, after dissolving in agarose solution at 80 °C, the pellet was infiltrated for 1 min in agarose solution at 37 °C, hardened on ice for 15 min, and then placed again in agarose solution, infiltrated for 1 min at 37 °C, hardened on ice for 15 min, and cut into cubes with 1–2 mm edges. The iRBC pellet was then washed in 0.1 M Cacodylate buffer and stained with 1% $OsO_4$ in 0.1 M Cacodylate buffer for 1 h at RT. iRBC pellet was washed in 0.1 M Cacodylate buffer and then in water, and then stained with 1% Uranyl Acetate overnight at 4 °C. The pellet was then washed in water and dehydrated by incubation with increasing concentrations of acetone (30, 50, 70 and 90%) for 10 min at RT each, and then incubated with 100% acetone overnight at 4 °C. After, the pellet was incubated with different concentrations of Spurrs resin in acetone (25, 50 and 75%), 45 min each, followed by incubation with 100% Spurrs resin overnight at 4 °C. The resin was then polymerised by incubation at 60 °C overnight, and the pellet-containing portion was sectioned into 70 nm slices using a diamond knife (Diatome ultra 35°) on Ultramicrotome EM UC6 (Leica), and placed on a copper grid coated with pioloform film. Samples were acquired using JEM-1400 (Jeol Ltd.) transmission electron microscope with EM- Menu (TVIPS GmbH) software. Images were analysed using ImageJ (NIH) software.

For scanning electron microscopy, 12 mm coverslips were cleaned by sonication (2x) in Milli Q water, incubated in 0.1 N HCl for 1 h, washed with water, incubated for 1 h in 96% ethanol, and stored in 96% EtOH until further use. Coverslips were coated with 0.01% poly-L-lysine (Merck Millipore) for 20 min at RT. The iRBCs were gently added to the coverslips, allowed to settle for 15 min at RT, and gently washed with 0.1 M Cacodylate buffer. The samples were stained with 1% $OsO_4$ solution for 1 h at RT, washed with 0.1 M cacodylate buffer and then with water. The samples were then dehydrated by incubation with different concentrations of acetone (30, 50, 70 and 90%) once, and in 100% acetone twice, 10 min each at RT. Critical point drying was performed with Leica EM CPD300. Next, coverslips were mounted on pins with silver glue and sputtered with a 10 nm film of palladium and gold mixture using Leica EM ACE200. Zeiss Leo 1530 (Zeiss) scanning electron

microscope was used for sample acquisition at 2 kV accelerating voltage, 4–6 mm working distance, and 12,000 x magnification, with the SE2 detector at 3000 × 2000 image setting in scanning speed 3, using SmartSEM (Zeiss) software. Images were analysed using ImageJ (NIH) software.

## *msp*2 genotyping

DNA was extracted using the DNAeasy Blood & Tissue Kit (Qiagen) from two DBS, each with 3 mm punches (Harris Uni-Core). Nested PCR of *msp*2 was performed following a previously described protocol (Liljander et al, 2009). Briefly, the outer *msp*2 domain (F: 5′-ATGAAGGTAATTAAAACATTGTCTATTATA -3′; R: 5′-CTTGTACACGGTACATTCTT-3′), was amplified with the following first PCR reaction: 95 °C for 5 min, 58 °C for 1 min, 72 °C for 1 min, 94 °C for 30 s, for 23 cycles, 58 °C for 1 min and 5 min at 72 °C. IC/3D7 and FC27 *msp*2 allelic families were identified using fluorescently labelled primers (IC/3D7 F: 5′-AGAA GATGCAGAAAGAAKCCTYCTACT-3′ and R: 5′-GATGTAATC GGGGGATCAGTTTGTTCG-3′ VIC; (FC27F: 5′ AATACAAGAG GTGGGCRATGCTCCA-3′; R: 5′-TTTTATTTGGTGCATTGCCA GAACTTGAAC-3′ 6-FAM) with the following cycling conditions: 5 min at 95 °C, 1 min at 61 °C, 1 min at 72 °C, 1 min at 94 °C, for 22 cycles, 30 s at 61 °C and 5 min at 72 °C. About 1 µl of nested PCR product was added to 9 µl of Hi-Di formamide (Applied Biosystems) and 0.5 µl size standard (GSTM-LIZ 1200, Applied Biosystems), containing 73 single-stranded DNA fragments ranging in size from 20 to 1200 bp. Fragment analysis was performed by capillary electrophoresis using a 3730 DNA sequencer (Applied Biosystems) and analysed using GeneMapper 5 software (Thermo Fisher Scientific). The cut-off value was set at 300 relative fluorescent units. Fragments were considered to be the same allele within each allelic type if they differed in size by less than three base pairs.

## *ama1* amplicon sequencing and analysis

DNA was extracted from DBS by Chelex-Tween as described above and hemi-nested PCR of the 236 base-pair segments *ama*1 gene of *P. falciparum* following previously published protocols (Briggs et al, 2020; Miller et al, 2017; Teyssier et al, 2021). PCR was performed with 5 µl of DNA, *ama*1 primers (*ama1*-OF: 5′-GCTGAAG-TAGCTGGAACTCAA-3′, *ama1*-R: 5′-TTTCCTGCATGTCTT-GAACA-3′) and Fast Start High Fidelity PCR kit (Roche) with the following cycling conditions: 2 min at 95 °C, 30 s at 95 °C, 30 s at 55 °C,1 min at 72 °C for 20 cycles, and then 10 min at 72 °C, followed by a second nested PCR using as template PCR products from the first PCR where a 5′ linker sequences were added (F: 5′-ACACTCTTTCCCTACACGACGCTCTTCCGATCT-3′, R: GTGA CTGGAGTTCAGACGTGTGCTCTTCCGATCT-3′) for Illumina sequencing. The same cycling conditions were used, and the number of cycles was adjusted based on parasite density (10, 20 or 25 cycles for samples with $10^3$–$10^4$, 10–$10^3$, $1^{-1}$–10 parasites per µl, respectively). Finally, an indexing PCR was performed using the PCR product from the second PCR and NEBNext Master Mix (New England BioLabs) with primers binding to the linker sequences and carrying Tru-Seq Illumina sequence adaptors. Samples were then combined per concentration threshold for fragment size analysis

using Agilent Bioanalyzer and pooled equimolarly before sequencing using an Illumina Miseq platform with a V2 500 cycles for 250 bp paired-end. Data were analysed using SeekDeep software (Miller et al, 2017). as described in Briggs et al, 2020. Haplotypes were included if they were present in both PCR duplicates. Haplotypes were further clustered using the cd-hit clustering algorithm (cd-hit) with a similarity threshold of 0.989 based on an elbow-method previously described (Fu et al, 2012).

Continuous clonal infections were defined as identical haplotypes consecutively present within an individual, allowing for haplotype absence in up to two consecutive timepoints (~1 month), given the high transmission intensity occurring at this cohort site that would allow for a fast clone turnover (Dicko et al 2004). To calculate the start of a clonal infection, we determined the calendar day in the middle of the timepoint of appearance and the previous collection timepoint. Given the absence of transmission during the dry season (Portugal et al, 2017; Andrade et al, 2020). Infections that were only detected at the end of the dry season (May 2013) were assumed to have been transmitted 14 days after the last timepoint during the transmission season (December 2012).

## Multiplexed amplicon sequencing 4CAST

After genomic DNA extraction from DBS as described above, 3 µl of DNA was used as a template for multiplexed amplicon sequencing, as previously described (LaVerriere et al, 2022), targeting *csp/ama1/sera2/trap* with the following primers, CSP-F: 5-TTAAGGAACAAGAAGGATAATACCA-3, CSP-R: 5-AAATGA CCCAAACCGAAATG-3; AMA1-F: 5-CCATCAGGGAAATGTC CAGT-3, AMA-R: 5-TTTCCTGCATGTCTTGAACA-3; SERA2-F: 5-TACTTTCCCTTGCCCTTGTG-3, SERA2-R: 5-CACTACAGA TGAATCTGCTACAGGA-3; TRAP-F: 5-TCCAGCACATGC GAGTAAAG-3, TRAP-R: 5-AAACCCGAAAATAAGCACGA-3, and linker sequences for Nextera adaptors, F: 5-TCGTCGGC AGCGTCAGATGTGTATAAGAGACAG-3, R: 5-GTCTCGTG GGCTCGGAGATGTGTATAAGAGACAG-3 in a first PCR using KAPA HiFi 2X (Roche), followed by an indexing PCR using unique dual indexes. The indexed samples were pooled and cleaned with Ampure XP beads with a final elution volume of 25 µl (Tris-HCl pH 8.0). Library quantification was performed with a Qubit High-Sensitivity kit (Thermo Fisher) and library fragment sizes were confirmed with an Agilent Bioanalyzer 2100. Libraries were then sequenced on an Illumina MiSeq using flow cell kit Nano v2 (500 cycles, paired-end), following manufacturers' instructions with 10% PhiX. Raw FASTQ files obtained for each sample and replicate were run through a previously described denoising pipeline based on a Divisive Amplicons Denoising Algorithm (DADA2) (Callahan et al, 2016; LaVerriere et al, 2022), and haplotypes were clustered using cd-hit-est algorithm and similarity threshold as described for *ama1* amplicon sequencing.

## Statistical analysis

Significance was assessed through different statistical methods and categorical variables in the different datasets. Continuous variables for metabolomics and proteomics were analysed using a Student's *t*-test, with *p* values adjusted using Bonferroni and FDR <0.05.

## The paper explained

### Problem

Malaria parasites are responsible for over 200 million malaria cases and 500,000 malaria-related deaths of African children yearly. Asymptomatic reservoirs of the infection continuously challenge elimination efforts, and, in seasonal areas, can persist for several months to restart transmission every year after the dry season. What determines parasite persistence for long periods of time is not known, and we speculate that seasonality may affect plasma composition so that *P. falciparum* can detect and adjust to changing serological cues; or alternatively that parasite infection length dictates clinical presentation and persistency.

### Results

Here we show, with data from more than a hundred children living in the malaria endemic area of Mali, how maintaining malaria parasites during the dry season is linked to the time when the individual parasite clones are transmitted, and not seasonality, or particular plasma profiles of the children keeping the reservoir.

We observed no, or minimal, differences in plasma composition across multiple times of the year, and we demosntrate that parasites collected from children, do not respond differently to plasma from the dry or the wet seasons in vitro. Our data also shows that parasites kept asymptomatically during the dry season do not reactivate to cause clinical malaria in the same individual, in response to the beginning of the wet season; and we see that parasites of asymptomatic children during the wet season are alike to dry season parasites in critical features associated with persistent infections. All together, a sensing mechanism appers not to be in place nor be necessary to promote dry season persitence.

On the other hand, we do show that parasite clones suviving the dry season are more likely to be transmistted late in the preceding transmisison season than those that are transmisted early on.

### Impact

Our results shed light on the intricate interactions between parasites, the human host, and the surrounding environment, and expand our understanding of how the dry season reservoir is established and allows the yearly restart of malaria. These data obtained with infected individuals naturally exposed to malaria may inform elimination strategies targeting the silent reservoirs of malaria.

We conducted a power analysis using the sample sizes for each of our experimental groups (asymptomatic, $n = 11$; negative, $n = 21$; dry season, $n = 32$; October, $n = 11$; MAL, $n = 6$) and with the statistical test used to detect differences in metabolite abundances (a two-sided $t$-test). We computed a conservative lower bound on the power using a Bonferroni (BF) adjusted $t$-test, which controls family-wise error rate and is more stringent than a Benjamini–Hochberg correction. For comparison, we used a power curve without adjustment for multiple correction as a maximum upper bound. Effect sizes are represented with Cohen's $d$ statistic, which is calculated as the difference in means of the groups divided by a pooled estimate of the standard deviation. (Appendix Fig. S3). Pathway enrichment analysis for metabolomics data using a hypergeometric test was also performed. For non-normally distributed continuous variables with more than two independent groups, a Kruskal–Wallis and Wilcoxon multiple comparison test were used. When comparing the median of multiple groups in normally distributed variables, the analysis of variance (ANOVA)

test, followed by the Tukey test for multiple comparisons was used. To test for independence between two or more categorical variables, either a chi-square test or Fisher's exact test were used, depending on the sample size and expected frequencies in the contingency table. We defined statistical significance as a two-tailed $p$ value of $\leq 0.05$. All statistical analyses were conducted using base R, and all R packages enclosed in the Tidyverse. Specific statistical tests employed are described in the corresponding figure legends. Multivariate analysis and generalised linear regression models were performed using R packages lme4 for the GLMM Poisson and GLMM Negative Binomial models and glmmADMB for the GLMM Zero-inflated Negative Binomial. Kaplan–Meier was performed using the Survival R package, with statistical significance between groups assessed using the log-rank test with significance.

## Data availability

The datasets and computer code produced in this study are available in the following databases: Amplicon sequencing data can be accessed through the NIH Sequence Read Archive (SRA) BioProject accession number PRJNA1054477, and sample accession numbers are listed in Dataset EV8. Proteomics data were uploaded to the Proteins Identifications Database (PRIDE) under project accession number PXD043980. The project name is "Characterisation of plasma from children living in a malaria endemic area of seasonal transmission". Infection duration of *P. falciparum* clones using ama1 amplicon sequencing: Github (https://github.com/macarrasq/falciparum_infection_length). All tables with metabolomics, proteomics and sequencing raw values used for analysis and data visualisation are listed in the supplementary material. All additional code used for analysis will be made available upon request. Flow cytometry files (Fig. 5) are available under https://doi.org/10.17617/3.V6KJN5.

The source data of this paper are collected in the following database record: biostudies:S-SCDT-10_1038-S44321-024-00127-w.

## Peer review information

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

## Acknowledgements

We thank the residents of Kalifabougou, Mali, for participating in this study. We acknowledge the support of the Immunology Core Lab of the UCRC in Bamako, Mali, and the COS Metabolomics Core Technology Platform at the Ruprecht-Karls-University Heidelberg. Pf2004 was kindly provided by Matthias Marti. Other Pf parasite lines were obtained through BEI Resources, NIAID, and NIH as part of the Human Microbiome Project. Manijeh Vafa Homann supported *msp*2 work. We thank Tuan M. Tran for reading and discussing the manuscript. This work was supported by the European Research Council under the European Union Horizon 2020 Research and Innovation Programme (grant agreement 759534), the German Center for Infection Research (DZIF TTU 03.701), the Lisa Meitner Excellence Programme of the Max Planck Society, the National Institutes of Health R01 AI141544, NIH/NIAID K24 AI144048 and the Division of Intramural Research, National Institute of Allergy and Infectious Diseases, of the National Institutes of Health.

## Author contributions

**Carolina, M Andrade**: Conceptualisation; Data curation; Formal analysis; Investigation; Methodology; Writing—original draft; Writing—review and editing. **Manuela Carrasquilla**: Data curation; Formal analysis; Investigation; Visualisation; Methodology; Writing—original draft; Writing—review and editing. **Usama Dabbas**: Investigation; Methodology; Writing—review and editing. **Jessica Briggs**: Data curation; Investigation; Methodology; Writing—review and editing. **Hannah van Dijk**: Investigation; Methodology; Writing—review and editing. **Nikolay Sergeev**: Investigation; Writing—review and editing. **Awa Sissoko**: Investigation; Writing—review and editing. **Moussa Niangaly**: Resources; Investigation; Writing—review and editing. **Christina Ntalla**: Investigation; Writing—review and editing. **Emily LaVerriere**: Methodology; Writing—review and editing. **Jeff Skinner**: Data curation; Writing—review and editing. **Klara Golob**: Investigation; Writing—review and editing. **Jeremy Richter**: Investigation; Writing—review and editing. **Hamidou Cisse**: Investigation; Writing—review and editing. **Shanping Li**: Methodology; Writing—review and editing. **Jason, A Hendry**: Data curation; Writing—review and editing. **Muhammad Asghar**: Methodology; Writing—review and editing. **Didier Doumtabe**: Resources; Writing—review and editing. **Anna Färnert**: Resources; Writing—review and editing. **Thomas Ruppert**: Methodology; Writing—review and editing. **Daniel, E Neafsey**: Funding acquisition; Investigation; Methodology; Writing—review and editing. **Kassoum Kayentao**: Resources; Writing—review and editing. **Safiatou Doumbo**: Resources; Writing—review and editing. **Aissata Ongoiba**: Resources; Writing—review and editing. **Peter Crompton**: Resources; Funding acquisition; Writing—review and editing. **Boubacar Traore**: Resources; Supervision; Writing—review and editing. **Bryan Greenhouse**: Resources; Supervision; Funding acquisition; Investigation; Methodology; Writing—review and editing. **Silvia Portugal**: Conceptualisation; Resources; Data curation; Formal analysis; Funding acquisition; Investigation; Visualisation; Methodology; Writing—original draft; Writing—review and editing.

Source data underlying figure panels in this paper may have individual authorship assigned. Where available, figure panel/source data authorship is listed in the following database record: biostudies:S-SCDT-10_1038-S44321-024-00127-w.

## Funding

## Disclosure and competing interests statement

The authors declare no competing interests.

# Expanded View Figures

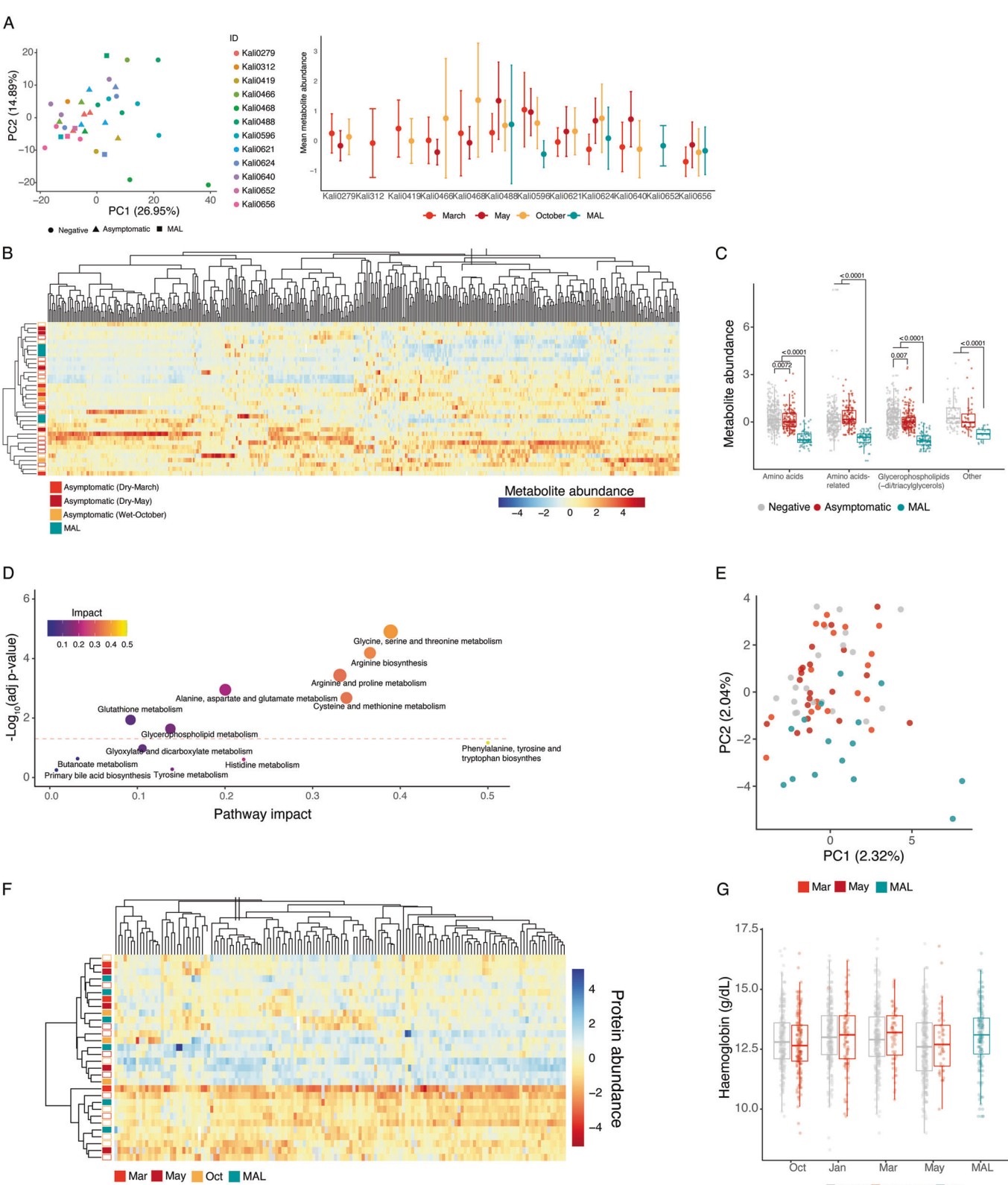

**Figure EV1. Plasma metabolites, lipids and proteins across seasons and clinical presentations.**

(A) Principal component analysis (PCA) of 456 metabolites in the plasma of uninfected (Negative shown in circles) individuals ($n = 19$), obtained in March ($n = 7$), May ($n = 5$) and October ($n = 7$); *P. falciparum* asymptomatic carriers (Asymptomatic shown in triangles) ($n = 11$), obtained in March ($n = 4$), May ($n = 4$) and October ($n = 3$); and clinical cases during the wet season (MAL shown in squares) ($n = 5$) presenting the variance explained by the two first principal components between individuals based on their metabolite profiles and coloured by individual (left panel). Mean and standard deviation of the normalised metabolite abundance for each individual ($n = 11$ individuals and 35 samples) at the different timepoints included in the analysis (right panel). (B) A heatmap with hierarchical clustering showing the normalised and log2-transformed abundance of 456 metabolites (columns) of 35 plasma samples (rows) collected along the year, infection and clinical presentation status. Metabolites are listed in Dataset EV1. (C) Significantly different metabolites across all sample comparisons ($n = 39$) grouped by metabolite class and infectious status, including uninfected individuals (Negative), *P. falciparum* asymptomatic carriers (Asymptomatic) and clinical malaria cases (MAL). Boxplots indicate median ± IQR with all individual values plotted, pairwise Student's *t*-test and adjusted *p* values are shown. (D) Pathway enrichment analysis of the significant metabolites between clinical cases during the wet season and uninfected and *P. falciparum* asymptomatic carriers. Circles represent pathways with the colour scale and size adjusted to the pathway impact (x-axis) and number of hit metabolites, respectively. Statistical significance was determined with hypergeometric test for overrepresentation analysis, and impact calculated based on the relative centrality from pathway topology analysis with multiple testing correction. Significant pathways have an adjusted *p* > 1.3 (y-axis) and pathway impact >0 and are listed in Dataset EV3. (E) Principal component analysis (PCA) was performed on data from 73 samples and 20 lipid species, showing the variance explained by the two first principal components between individuals based on their lipid profiles. (F) A heatmap with hierarchical clustering showing protein abundance for each of the 146 proteins (columns) across 30 plasma samples (rows) collected along the year, infection and clinical presentation status. Proteins are listed in Dataset EV4.

(G) Haemoglobin levels in the blood of study participants above 5 years old collected along the year, infection and clinical presentation status ($n = 391$ study participants). Boxplots indicate median ± IQR with all individual values plotted.

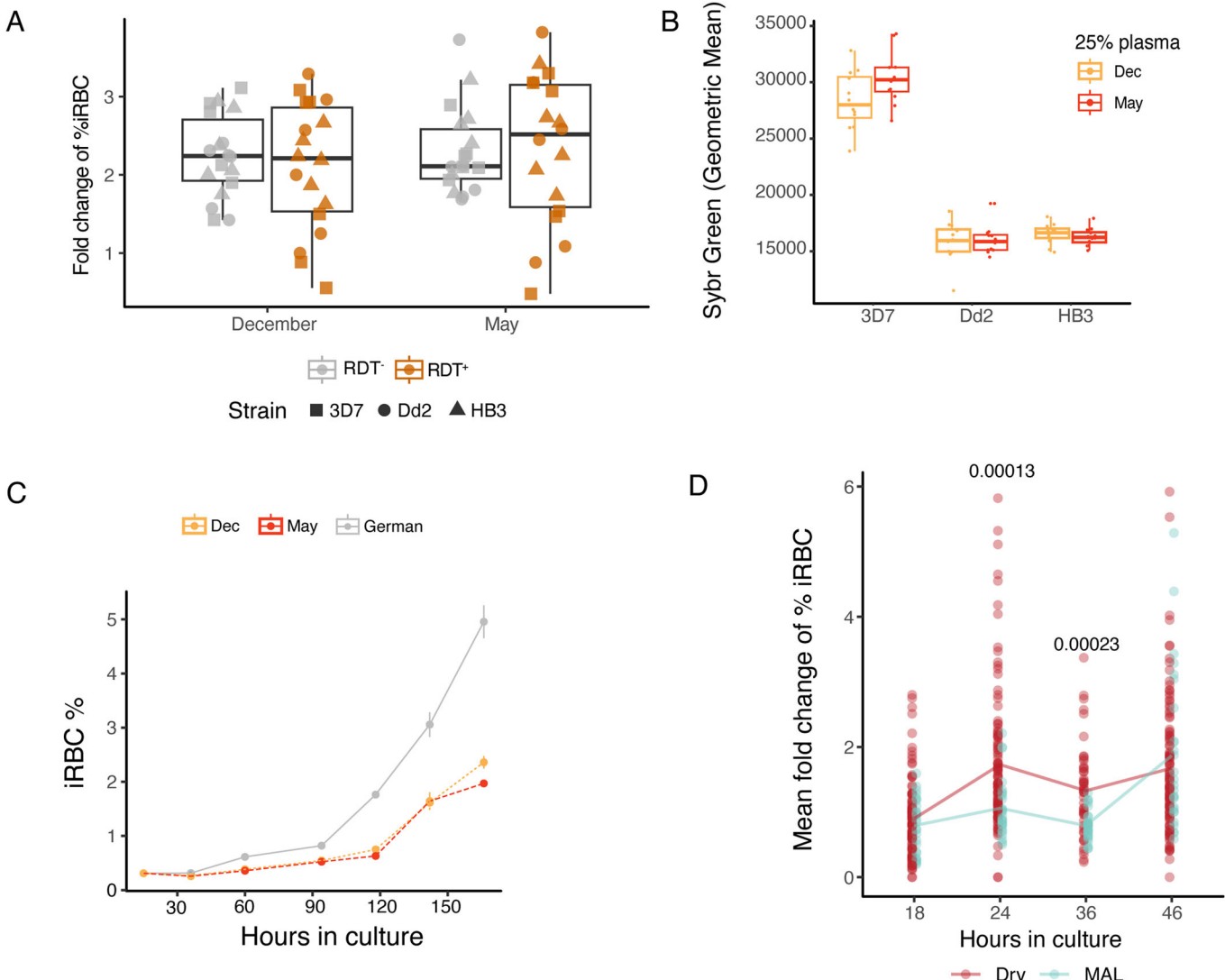

**Figure EV2. Plasma from the dry and wet seasons have a similar effect on *P. falciparum* lab-adapted strains in vitro.**

(A) Parasitaemia fold change of lab-adapted 3D7, Dd2 and HB3 *P. falciparum* strains after 60 h in vitro culture supplemented with 25% plasma from Malian donors, split by infection status by rapid diagnostic test of plasma donors (RDT⁻, *n* = 6, RDT⁺, *n* = 6). (B) Parasite development is measured as Sybr Green geometric mean in different *P. falciparum* lab-adapted strains cultured in vitro supplemented with December or May plasma from Malian donors (*n* = 12), measured at 36 h in culture (3D7) or 30 h in culture (Dd2 and HB3). (C) *P. falciparum* 2004 strain growth for over three 48 h cycles in vitro supplemented with 25% plasmas pooled from 12 Malian children in the beginning (December, *n* = 3) or end (May, *n* = 3) of the dry season, or from four German adults (*n* = 3). (D) Average fold change of iRBCs at 18, 24, 36 and 46 h after culture and across all plasma conditions for parasites obtained from the dry season (*n* = 37) or from clinical malaria cases (*n* = 9). Kruskal–Wallis and Wilcoxon pairwise test. Fold changes are defined as %iRBC t(*n*)/%iRBC t(*n* − 1). All boxplots indicate median ± IQR with all individual values plotted.

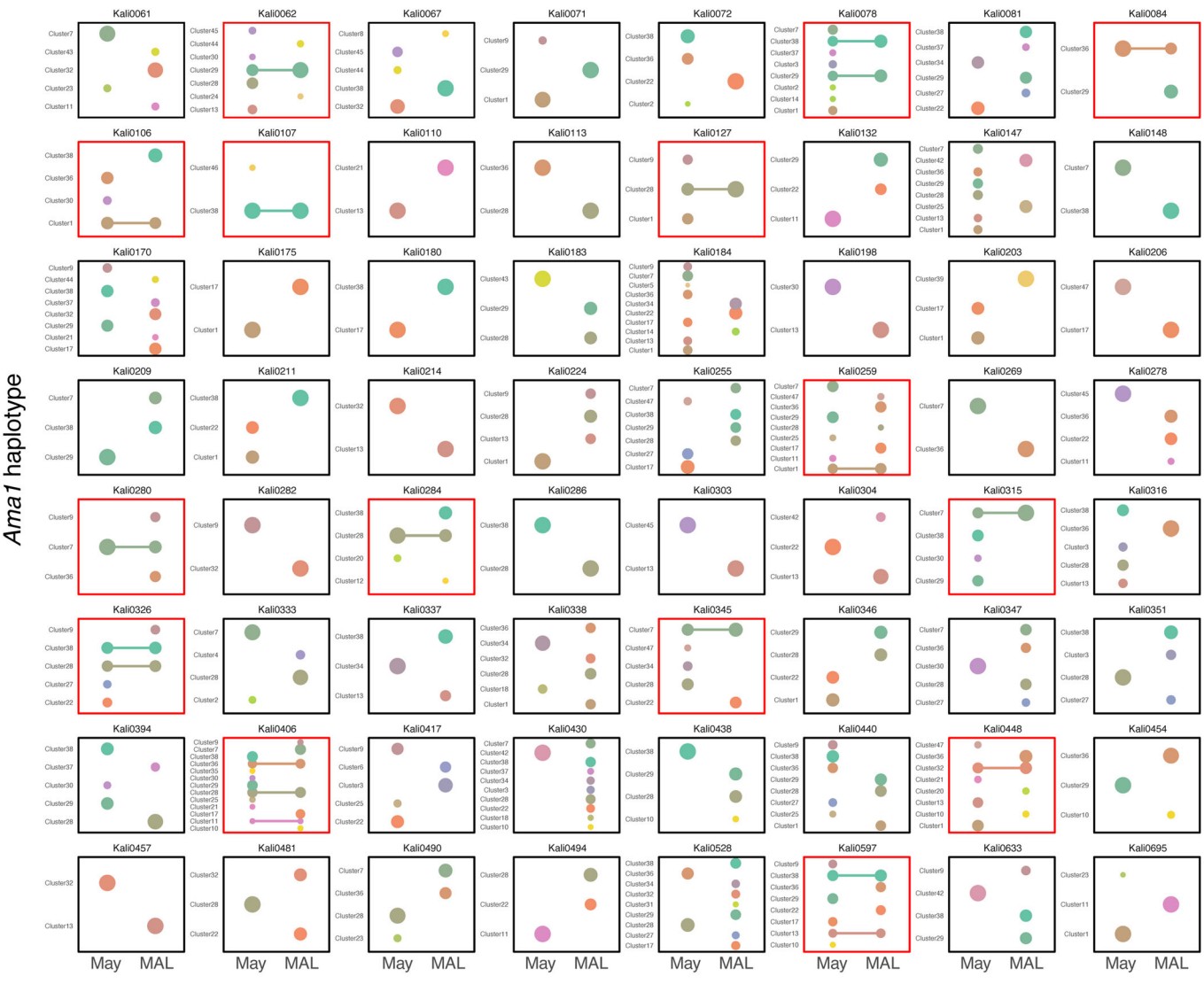

**Figure EV3.  Clinical malaria episodes are mainly caused by newly transmitted *P. falciparum* parasites.**

*ama1* haplotypes in paired samples from 64 children collected at the end of the dry season (May) and during their first clinical malaria case in the ensuing wet season (MAL). Red squares highlight children with shared haplotypes in May and MAL. Circle size represents haplotype proportion within each timepoint.

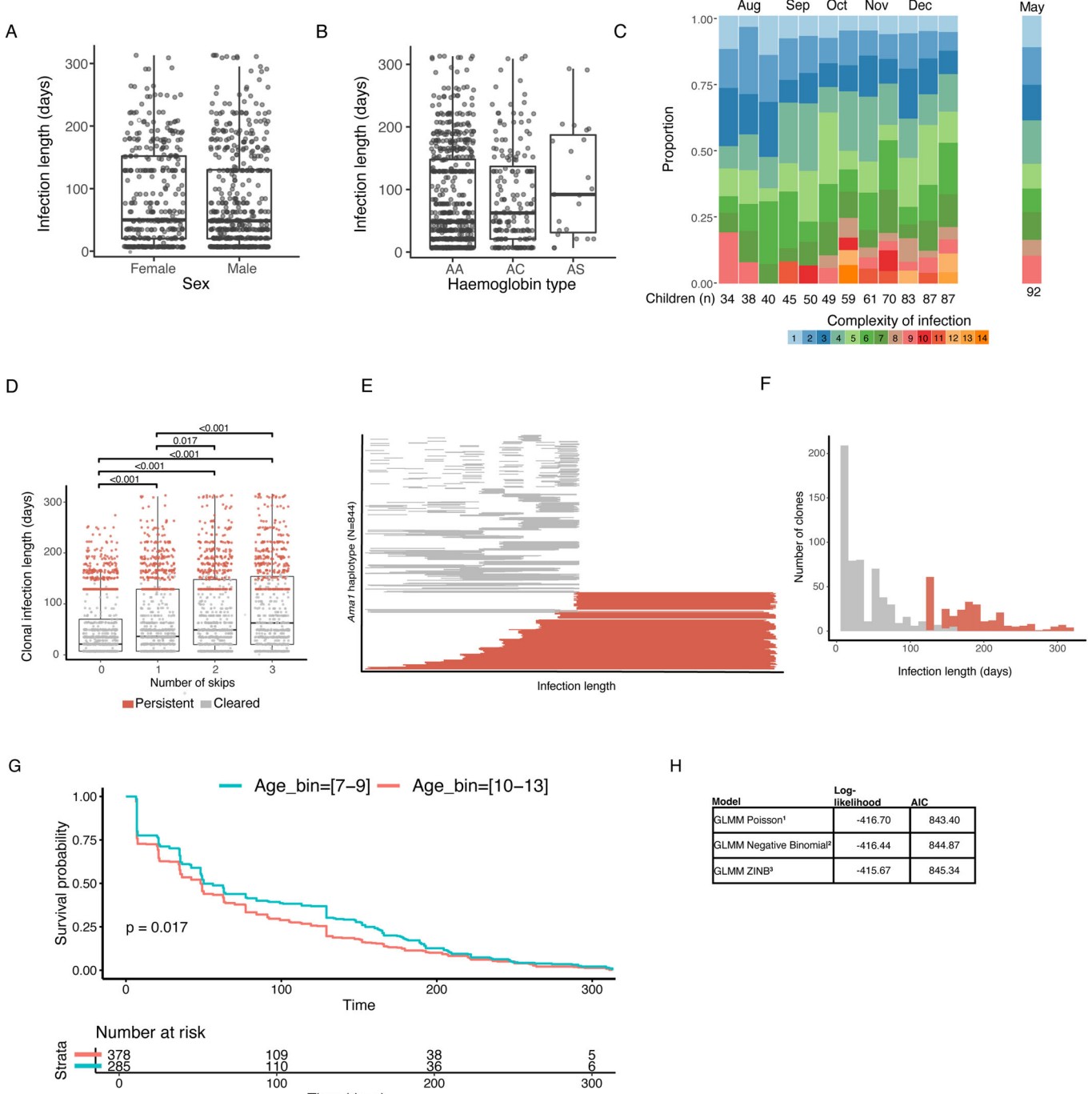

**Figure EV4. Determinants of infection length and complexity of infection in dry-season persisting infections.**

Infection length of all *P. falciparum* clones (allowing for two skips) found in 92 Malian children separated by (**A**). Sex and (**B**). haemoglobin type; (AA *n* = 71, AC *n* = 12, AS *n* = 9). Boxplots indicate median ± IQR with all values plotted corresponding to the individuals and timepoints. (**C**) Complexity of infection (COI) of 92 asymptomatic children found PCR In May 2013 at the end of the dry season, analysed retrospectively over the 12 timepoints in the preceding wet season from December 2012 until July 2012, and at the end of the preceding dry season (May 2012). (**D**) Infection length of *P. falciparum* clones allowing 0, 1, 2 or 3 skips (negative timepoints of a particular *ama*1 clone within a series of positive ones). Brick dots show persistent infections, and grey dots represent clones cleared before the end of the dry season (ANOVA and Tukey multiple comparison test). Boxplots indicate median ± IQR. (**E**) Haplotypes are ordered by clonal infection length. (**F**) Distribution of clones and their infection length. Brick lines show persistent infections, and grey lines represent clones cleared before the end of the dry season. (**G**) Kaplan–Meier analysis showing the probability of survival of *P. falciparum* clones (y-axis) over time, from the start of the wet season (July 2012) to the end of the dry season (May 2013) (x-axis) in individuals of 7–9 (aqua) and 10–13 (coral) years old, with the log-rank test used to determine statistical differences. (**H**) Summary of three generalised mixed-effect models used showing the Akaike Information Criteria and the log-likelihood for the selected model, GLMM Poisson (1), GLMM negative binomial model (2) and GLMM ZINB (3).

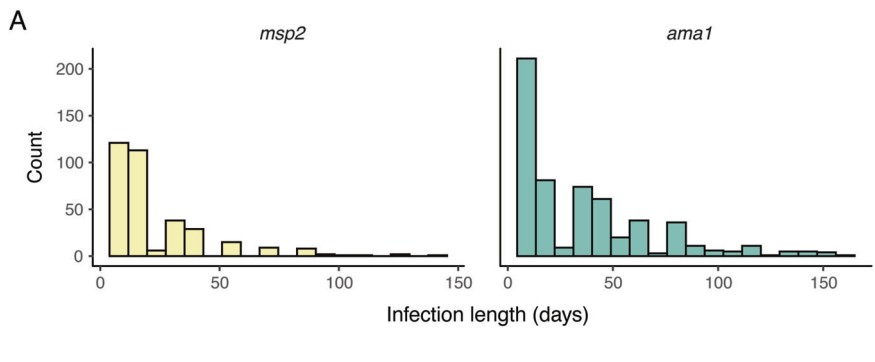

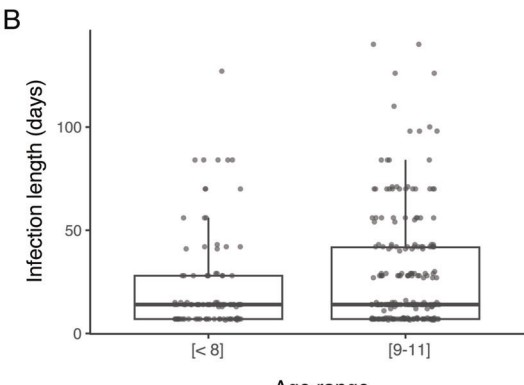

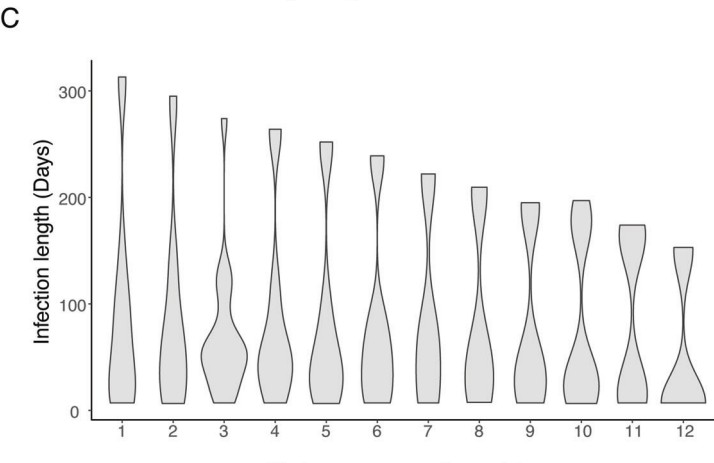

**Figure EV5.** Distribution of infection length of *P. falciparum* clones using two genotyping approaches.

(**A**) Distribution of infection length in days of *P. falciparum* clones detected by *msp2* genotyping (yellow) and *ama1* genotyping methods (green), showing length with two skips of 346 and 582 clonal infections, respectively (x-axis). The Y-axis corresponds to the number of clones per genotyping method. (**B**) Infection length determined by *msp2* genotyping of younger (5–8) or older (9–11) children (**C**) Infection length distribution using *ama1* amplicon sequencing and 2 skips of individual clones across all individuals for the 12 timepoints spanning the transmission season (July to December, 2012).

