## [Peer Review File · EMBO Molecular Medicine]

Infection length and host environment influence on *Plasmodium falciparum* dry season reservoir

Carolina Andrade, Manuela Carrasquilla, Usama Dabbas, Jessica Briggs, Hannah van Dijk, Nikolay Sergeev, Awa Sissoko, Moussa Niangaly, Christina Ntalla, Emily LaVerriere, Jeff Skinner, Klara Golob, Jeremy Richter, Hamidou Cisse, Shanping Li, Jason Hendry, Muhammad Asghar, Didier Doumtabe, Anna Färnert, Thomas Ruppert, Daniel Neafsey, Kassoum Kayentao, Safiatou Doumbo, Aissata Ongoiba, Peter Crompton, Boubacar Traore, Bryan Greenhouse, and Silvia Portugal

Corresponding authors: *Silvia Portugal* (portugal@mpiib-berlin.mpg.de) , *Carolina Andrade* (carolina.andrade@radboudumc.nl)

Review Timeline:

Submission Date:	27th Jul 23
Editorial Decision:	3rd Aug 23
Appeal:	8th Aug 23
Editorial Decision:	14th Sep 23
Revision Received:	12th Feb 24
Editorial Decision:	9th Apr 24
Revision Received:	19th Jun 24
Editorial Decision:	8th Jul 24
Revision Received:	5th Aug 24
Editorial Decision:	7th Aug 24
Revision Received:	7th Aug 24
Accepted:	9th Aug 24

Editor: Poonam Bheda

Transaction Report:

1st Aug 2023

Decision on your manuscript EMM-2023-18321

Dear Dr. Portugal,

Thank you for submitting your manuscript "Infection length and host environment influence on Plasmodium falciparum dry season reservoir" to EMBO Molecular Medicine. I have now carefully read your manuscript and discussed it with my colleagues. I regret to say that we all agree that the manuscript not well suited for publication in EMBO Molecular Medicine and therefore have decided not to proceed with peer review.

In this study, you compare the metabolomic and proteomic profiles of plasma samples collected at different times of year and with different clinical presentations of malaria from a Malian longitudinal cohort. We appreciate the hypothesis that *P. falciparum* may have no specific sensing mechanism of the dry season based on the fact that there were limited differences in the proteome and metabolome of Malian plasma collected at different times of the year with no significant effects on parasitemia, but that rather the duration of infection and/or the number of replication cycles underlies the seasonality of clinical symptoms. Unfortunately we are not persuaded that your findings provide the sort of decisive clinical insight we would expect in an EMBO Molecular Medicine article.

That being said, I appreciate the findings and have discussed the manuscript with my colleague Achim Breiling at our sister journal EMBO Reports. If you are interested in transferring your manuscript to EMBO Reports, he would discuss your manuscript with an expert advisor on sending out the study for full peer review. If you are interested in this option, please use the transfer link below. Please note that no reformatting of the manuscript is necessary for a transfer to EMBO Reports. If you have any further questions, don't hesitate to contact Achim Breiling (a.breiling@emboreports.org) for further information.

I am sorry I could not bring better news regarding the publication of your study in EMBO Molecular Medicine and I hope that you will view the possibility of a transfer to EMBO Reports favorably.

Yours sincerely,

Poonam Bheda

Poonam Bheda, PhD
Scientific Editor
EMBO Molecular Medicine

=====

As a service to authors, EMBO provides authors with the possibility to transfer a manuscript that one journal cannot offer to publish to another EMBO publication. The full manuscript and if applicable, reviewers reports are automatically sent to the receiving journal to allow for fast handling and a prompt decision on your manuscript. For more details of this service, and to transfer your manuscript to another EMBO title please click on Link Not Available

Dear Dr Bheda,

Thank you so much for the very quick assessment of our manuscript.

Your reply was indeed not the outcome we expected, and we remain confident that our data will attract the interest of a broad readership of clinical infectious diseases, pathogen persistence or immune evasion communities, as have our previous work.

I am a recurrent reviewer for EMBO Molecular Medicine, and can confidently say that many articles are often published in your journal with far less clinical insight than what we now submit and would have liked to see fairly reviewed.

We show, with data from more than a hundred children living in a malaria endemic area how maintaining *P. falciparum* parasites during the dry season is linked to the time when the individual parasite clones are transmitted, and not seasonality or particular metabolome/proteome profile of the children keeping the reservoir.

These data shed light on how the the dry season reservoir is established and allows the yearly start of transmission of one of the deadliest diseases in the world, and our data was obtained with infected individuals naturally exposed to malaria, which no experimental setting would be even close to successfully attempt.

In my view you overemphasise the metabolomic and proteomic data as the only piece to sustain the lack of evidence of a parasite encoded sensing mechanism, but in fact we show across multiple figures:

- 1) no or minimal differences in metabolome and proteome of plasma from children across multiple time point of the year
- 2) parasites collected from children at different time of the year do not respond differently to plasma from the dry and the wet season in parasite development, replication or host cell remodelling in vitro
- 3) parasites kept asymptotically during the dry season do not reactivate to cause malaria in same individual in response to the beginning of the wet season
- 4) parasites of asymptomatic children during the wet season are alike to dry season parasites in critical features promoting extended circulation which is the main feature of a dry season persistent infection.

Then, we questioned and show a relation between the time of first appearance of a parasite clone in the wet season and its chance of survival to compose the dry season reservoir and bridge to the ensuing wet season, which again supports the lack of a sensing mechanism.

Thank you for re-considering the potential value of our manuscript to EMM given the reasons above-mentioned and I look forward to hearing back from you.

Silvia

14th Sep 2023

Dear Dr. Portugal,

Thank you for the submission of your manuscript to EMBO Molecular Medicine. We have now received feedback from the three reviewers who agreed to evaluate your manuscript. As you will see from the reports below, the referees acknowledge the interest of the study and are overall supporting publication of your work pending appropriate revisions.

Addressing the reviewers' concerns in full will be necessary for further considering the manuscript in our journal, and acceptance of the manuscript will entail a second round of review. EMBO Molecular Medicine encourages a single round of revision only and therefore, acceptance or rejection of the manuscript will depend on the completeness of your responses included in the next, final version of the manuscript. For this reason, and to save you from any frustrations in the end, I would strongly advise against returning an incomplete revision.

We are expecting your revised manuscript within three months, if you anticipate any delay, please contact us.

We require:

4) A .docx formatted letter INCLUDING the reviewers' reports and your detailed point-by-point responses to their comments. As part of the EMBO Press transparent editorial process, the point-by-point response is part of the Review Process File (RPF), which will be published alongside your paper.

5) A complete author checklist, which you can download from our author guidelines (<https://www.embopress.org/page/journal/17574684/authorguide#submissionofrevisions>). Please insert information in the checklist that is also reflected in the manuscript. The completed author checklist will also be part of the RPF.

6) Please note that all corresponding authors are required to supply an ORCID ID for their name upon submission of a revised manuscript.

7) It is mandatory to include a 'Data Availability' section after the Materials and Methods. Before submitting your revision, primary datasets produced in this study need to be deposited in an appropriate public database, and the accession numbers and database listed under 'Data Availability'. Please remember to provide a reviewer password if the datasets are not yet public (see <https://www.embopress.org/page/journal/17574684/authorguide#dataavailability>).

In case you have no data that requires deposition in a public database, please state so in this section. Note that the Data Availability Section is restricted to new primary data that are part of this study. This study includes no data deposited in external repositories.

8) For data quantification: please specify the name of the statistical test used to generate error bars and P values, the number (n) of independent experiments (specify technical or biological replicates) underlying each data point and the test used to calculate p-values in each figure legend. The figure legends should contain a basic description of n, P and the test applied. Graphs must include a description of the bars and the error bars (s.d., s.e.m.). Please provide exact p values.

9) Our journal encourages inclusion of *data citations in the reference list* to directly cite datasets that were re-used and obtained from public databases. Data citations in the article text are distinct from normal bibliographical citations and should directly link to the database records from which the data can be accessed. In the main text, data citations are formatted as

follows: "Data ref: Smith et al, 2001" or "Data ref: NCBI Sequence Read Archive PRJNA342805, 2017". In the Reference list, data citations must be labeled with "[DATASET]". A data reference must provide the database name, accession number/identifiers and a resolvable link to the landing page from which the data can be accessed at the end of the reference. Further instructions are available at .

13) Author contributions: CRediT has replaced the traditional author contributions section because it offers a systematic machine readable author contributions format that allows for more effective research assessment. Please remove the Authors Contributions from the manuscript and use the free text boxes beneath each contributing author's name in our system to add specific details on the author's contribution. More information is available in our guide to authors.

Please also suggest a striking image or visual abstract to illustrate your article as a PNG file 550 px wide x 300-600 px high. Share synopsis text and image, as well as eTOC:

Please note that these would be the final versions and changes during proofing are usually not allowed

16) As part of the EMBO Publications transparent editorial process initiative (see our Editorial at <http://embomolmed.embopress.org/content/2/9/329>), EMBO Molecular Medicine will publish online a Review Process File (RPF) to accompany accepted manuscripts.

In the event of acceptance, this file will be published in conjunction with your paper and will include the anonymous referee reports, your point-by-point response and all pertinent correspondence relating to the manuscript. Let us know whether you agree with the publication of the RPF and as here, if you want to remove or not any figures from it prior to publication.

I look forward to receiving your revised manuscript.

Yours sincerely,

Poonam Bheda

Poonam Bheda, PhD
Scientific Editor
EMBO Molecular Medicine

We require:

- 1) A .docx formatted version of the manuscript text (including legends for main figures, EV figures and tables). Please make sure that the changes are highlighted to be clearly visible.
 - 2) Individual production quality figure files as .eps, .tif, .jpg (one file per figure). For guidance, download the 'Figure Guide PDF': (<https://www.embopress.org/page/journal/17574684/authorguide#figureformat>).
 - 3) A .docx formatted letter INCLUDING the reviewers' reports and your detailed point-by-point responses to their comments. As part of the EMBO Press transparent editorial process, the point-by-point response is part of the Review Process File (RPF), which will be published alongside your paper.
 - 4) A complete author checklist, which you can download from our author guidelines (<https://www.embopress.org/page/journal/17574684/authorguide#submissionofrevisions>). Please insert information in the checklist that is also reflected in the manuscript. The completed author checklist will also be part of the RPF.
 - 5) Please note that all corresponding authors are required to supply an ORCID ID for their name upon submission of a revised manuscript.
 - 6) It is mandatory to include a 'Data Availability' section after the Materials and Methods. Before submitting your revision, primary datasets produced in this study need to be deposited in an appropriate public database, and the accession numbers and database listed under 'Data Availability'. Please remember to provide a reviewer password if the datasets are not yet public (see <https://www.embopress.org/page/journal/17574684/authorguide#dataavailability>).
- In case you have no data that requires deposition in a public database, please state so in this section. Note that the Data Availability Section is restricted to new primary data that are part of this study.
- 7) For data quantification: please specify the name of the statistical test used to generate error bars and P values, the number (n) of independent experiments (specify technical or biological replicates) underlying each data point and the test used to calculate p-values in each figure legend. The figure legends should contain a basic description of n, P and the test applied. Graphs must include a description of the bars and the error bars (s.d., s.e.m.). See also 'Figure Legend' guidelines: <https://www.embopress.org/page/journal/17574684/authorguide#figureformat>
 - 8) At EMBO Press we ask authors to provide source data for the main manuscript figures. Our source data coordinator will contact you to discuss which figure panels we would need source data for and will also provide you with helpful tips on how to upload and organize the files.
 - 9) Our journal encourages inclusion of *data citations in the reference list* to directly cite datasets that were re-used and obtained from public databases. Data citations in the article text are distinct from normal bibliographical citations and should directly link to the database records from which the data can be accessed. In the main text, data citations are formatted as follows: "Data ref: Smith et al, 2001" or "Data ref: NCBI Sequence Read Archive PRJNA342805, 2017". In the Reference list, data citations must be labeled with "[DATASET]". A data reference must provide the database name, accession number/identifiers and a resolvable link to the landing page from which the data can be accessed at the end of the reference. Further instructions are available at .
 - 10) We replaced Supplementary Information with Expanded View (EV) Figures and Tables that are collapsible/expandable

online. A maximum of 5 EV Figures can be typeset. EV Figures should be cited as 'Figure EV1, Figure EV2' etc... in the text and their respective legends should be included in the main text after the legends of regular figures.

.

13) Author contributions: You will be asked to provide CRediT (Contributor Role Taxonomy) terms in the submission system. These replace a narrative author contribution section in the manuscript.

14) A Conflict of Interest statement should be provided in the main text.

Please note: When submitting your revision you will be prompted to enter your funding and payment information. This will allow Wiley to send you a quote for the article processing charge (APC) in case of acceptance. This quote takes into account any reduction or fee waivers that you may be eligible for. Authors do not need to pay any fees before their manuscript is accepted and transferred to the publisher.

EMBO Press participates in many Publish and Read agreements that allow authors to publish Open Access with reduced/no publication charges. Check your eligibility: <https://authorservices.wiley.com/author-resources/Journal-Authors/open-access/affiliation-policies-payments/index.html>

***** Reviewer's comments *****

Referee #1 (Remarks for Author):

The manuscript by Andrade et al. describes analyses of sera and *P. falciparum* parasites from blood samples collected during the dry and wet season in Mali. The authors investigate a number of hypotheses that have been proposed to explain parasite persistence during the dry season. They used a comprehensive and well-designed series of experiments to test these hypotheses and, while the results are mostly negative, shed new lights on the mechanisms underlying parasite persistence. I really appreciated the comprehensiveness of the analyses, the multi-pronged approaches to study both serum and parasites, and the objectivity in presenting the results. Overall, I find the study interesting and definitely worth publishing. I have a few minor comments that I describe below.

- the description of the serum selection for the first analysis is hard to follow (lines 151-154). The authors might want to clarify this section and maybe add a supplemental figure. The colors on Figure 1 are also very difficult to distinguish.
- the significance threshold used for the metabolite analysis (lines 161- 164) needs to be included in the text. Figure 2B also seems to show two distinct distributions (for the pvalues under the cutoff). Why is that?
- the analyses of plasma supplementation on parasite growth in vitro (lines 207+) are really exciting but the authors need to clarify when these plasma samples were collected and whether the individuals were infected at the time of collection (or shortly before).
- it looks like *ama1* was genotyped twice independently (once by amplicon sequencing and once by 4CAST). What was the concordance between the two approaches? Also, the statement that "newly transmitted infections lead to clinical malaria" (line 296) could be moderated (or reinforced) by a discussion of the limit of detection of these approaches.
- the numbering of the multi-panel figures (Fig 3 and 4) is really annoying (C before B)
- the analyses presented in Fig 5 are fascinating. However, given the extensive variations observed in %Rings on Figure 5D, I would have liked to see an analysis of the growth in vitro vs. %Rings performed at the individual parasite level rather than only by agglomerating the data by group (i.e., for the dry season samples, how is the correlation between %Rings and fold-change of iRBCs?)

Overall, a very comprehensive and interesting study presented very objectively.

David Serre

Referee #2 (Comments on Novelty/Model System for Author):

There are some issues with justification of methods, and the MS would benefit from alternative explanations for the patterns observed. These patterns are novel and likely to be highly cited, and the study has considerable epidemiological relevance for malaria. The data was collected from humans on the most virulent malaria parasite species (*P. falciparum*) so is an entirely relevant system for human health.

Referee #2 (Remarks for Author):

In this study, the authors seek to answer how *P. falciparum* persists throughout the dry seasons in asymptomatic infections when transmission isn't possible - specifically, by investigating whether parasites respond to seasonality through cues within plasma composition. Samples from infected individuals through wet and dry seasons. The authors found that i) there is no significant difference in the plasma on the basis of seasonality or presentation (asymptomatic vs symptomatic), ii) likewise, plasma acquired during wet versus dry seasons had similar effects on artificially cultured parasites, iii) through the use of sequencing to track clones, authors found that clinical malaria was caused by clones more recently transmitted during the wet season, and iv) clones transmitted later in the wet season were more likely to survive longer and thus make it to the ensuing dry season. The authors found little evidence that parasites respond to serological cues in the plasma of their hosts. Instead, the authors propose that changes in the dynamics of circulating iRBCs, specifically the timing with which iRBCs sequester, could lead to clinical differences in wet versus dry season infections.

The paper does well in introducing and motivating the questions and differentiating between wet vs dry season and asymptomatic vs clinical infections. The patterns the authors describe are extremely interesting and likely to spur a lot of useful research in this space, but a more in-depth discussion of alternative explanations for these patterns and justification of the methods would greatly enhance the study (see major comments below). In particular, the discussion of possible explanations for the dynamics of circulating iRBCs and developmental stages of infections requires more elaboration.

Major comments

A fuller discussion of possible explanations for the observed patterns would be useful. First, the introduction suggests that the authors are considering the possibility of a parasite adaptation in the form of sensing pathways to enable infections to persist through the dry season, but it is not clear how their primary explanation for observed patterns-parasites extending circulation times and being killed at higher rates in the spleen-would help parasites persist. Second, the positive association between ring stage parasites and iRBC burden is exactly what would be expected according to population dynamics. An expanding parasite population (i.e., one that generates a large percentage of iRBCs) would show a bias towards earlier developmental ages (a higher percentage of rings). This pattern has been known for populations of any organism for decades and noted for malaria parasites specifically (e.g., Khoury et al. 2014 *Infect Immun*; Greischar et al. 2019 *Trends Parasitol*). The authors need to

provide more explanation and justification for why they favor the hypothesis of increased circulation times over the alternative, that some later stages circulate in both wet and dry season infections but that those stages are harder to find in wet season infections because those infections are expanding. Third, highlighting other possibilities in the discussion would make this study more impactful, including the possibility that dry season infections show more later stages because they persist longer and have had longer to desynchronize. Wet season infections will be in the early portion of infection, since these infections do not last long, and may not have had time to desynchronize. Changing investment into transmission stage production would also alter rates of replication in the blood and may explain the dynamics of chronic infection in other hosts (e.g., canaries, Cornet et al. 2014 PLoS Path). Do the authors have reason to believe altered transmission investment strategies could not explain the patterns they see? If the authors have on transmission stages, it could bolster their conclusions. In lieu of such data, it would be helpful for the authors to discuss the potential role of transmission investment strategies in persistent infections and transmission during wet seasons. Is the pattern necessarily driven by parasites or could host differences in the dry season (e.g., immune exhaustion/depletion following intense wet season transmission) be responsible? Discussing these key points would help guide future work in this space.

More context and in some cases justification of the methods is needed to bolster the results. Some rationale and further citations would be useful on possible cues in plasma that could trigger changes in parasite strategies (paragraph starting at line 102). Counts of iRBCs were made in comparison to 300 leukocytes (lines 601-602), but are the authors confident that there were no systematic differences in leukocyte counts between hosts in the wet and dry seasons (and if so, why)? Studies suggest other methods (e.g., counting to a particular number of fields) are more accurate than counting to a particular number of leukocytes (Kilian et al. 2000 Trop Med Hyg Int Health, O'Meara et al. 2005 Am J Trop Med Hyg), and that could be especially true if there are systematic differences in mean leukocyte counts across season. The authors' previous study (Andrade et al. 2020) seems to show no systematic differences in leukocyte composition but a case needs to be made that overall leukocyte abundance is also similar across seasons.

Minor

- Lines 58-60. This finding needs more explanation so that readers can get the key point without going back to read the previous paper. The phrasing leaves the time scale open to misinterpretation-circulation could refer to persistence in the blood rather than the very specific meaning of circulating as opposed to sequestered iRBCs. Adding a sentence to orient readers to malaria biology and terminology would be helpful.
- Lines 66-67: The phrase "rigidity of substrate" is a bit mysterious in this context-rephrase or explain?
- Line 218: Comparable in what way? Supplementary fig. 2 shows similar SYBR Green uptake (for which a brief explanation of the biological significance of that measure would be very helpful) and similar change in percent iRBCs through time, but the phrasing in the main text could mean comparable intraerythrocytic developmental duration, or comparable multiplication rates.
- Line 223: Could the authors add a few more words to explain the meaning of 'demonstrated sensing ability'?
- Line 397: 'individual was found PCR' negative or positive presumably?
- Line 477: late in the preceding wet season?
- In figure 1, the colored human shapes are a bit confusing as to what they represent - a brief explanation in the caption or text explaining what each person is (a set of participants, how many, etc.) should suffice.
- Fig 2: Pathway analysis, and what pathway impact entails, requires more explanation.
- The tracking of clones throughout the seasons and their survival time is a supporting pillar to one of the paper's conclusions (lines 434, 500, 513), and a supplemental graph that actually shows the timelines of the different clones - from when they were acquired (including wet and dry season) to how long they likely survived - would be useful in illustrating the finding that parasites acquired earlier in the wet season had shorter survival/those transmitted later in the wet season were more likely to last the dry season.
- The authors offer to make their code available upon request, but the study would be more likely to be used (and cited) if the code were made publicly available (e.g., on Github).

Referee #3 (Comments on Novelty/Model System for Author):

See main review below for full details.

Statistical analysis is poor - recommend engaging with a statistician.

Novelty and medical impact are low, mainly due to failure to find significant correlations.

The paper does not include work with a model organism - the actual parasite causing human malaria is used, and the samples are from naturally infected children.

Ethical issues are covered appropriately.

Referee #3 (Remarks for Author):

In this study the authors have investigated factors contributing to the persistence of human malaria parasites (*P. falciparum*) in the human body during the long dry season where there is no mosquito transmission. They focus on potential cues in the blood that the parasite could respond to in order to change growth characteristics, by analysing the metabolome and proteome within

plasma at different time periods throughout the year. Alternatively they hypothesise that parasite persistence is related to the length of an infection.

Their results are largely negative - they failed to find any significant differences in the metabolites and proteins present in the plasma taken in the wet and dry seasons, nor in their impact on parasite growth in vitro. They then conclude that this means that there cannot be a sensing mechanism in parasites to adjust their growth/ behaviour seasonally. This conclusion is not convincing, since not all components of plasma have been analysed - significant omissions are lipids and immune effectors, and they did not investigate sensing mechanisms, but only potential cues. "Absence of Evidence is not Evidence of Absence" [Sagan]. A second analysis concludes that parasites transmitted later in the wet season had a higher chance of surviving the long dry season, and that persistence is linked to the length of infection. The statistical analyses to support this conclusion are not wholly convincing.

Parasite persistence over long periods in the absence of reinfection from mosquitoes is poorly understood, and this paper adds a little to what is known already, but generally reinforces previous work from the same group and from others. There is relatively little of novelty presented, largely because the study failed to find differences between wet and dry season host plasma. The paper is unlikely to be of strong interest to more than a specialised audience.

The main shortcomings of the manuscript are the small number of individuals that have been included in the metabolome and proteome study, the omission of analyses of lipid content and immune effectors in the plasma, and the weakness of the statistical approaches. Failure to find a difference does not mean that there is not a difference if the study is underpowered. It is possible that with more sophisticated statistical approaches, for example multivariate modelling approaches, stronger conclusions could be drawn, and I would recommend that the authors engage with a statistician/modeller if they have not done so already. Another weakness is the lack of detail provided in the methodology at various points (given below) - while it is possible that some of the methods have been published previously, this information is not given either, and so a reader wishing to understand at a deeper level what has been done is out of luck.

Major Points

1) Sample size: The number of individuals in the metabolomics study was surprisingly small - they appear to have included only 5 infected, and 6 parasite-free individuals during the dry season, with 5 age- and sex-matched individuals in the wet season (presenting with clinical malaria). What is the statistical power given these low numbers to see any significant difference in the presence, absence or levels of any of the 456 metabolites examined? The absence of a significant difference between groups, given these low numbers, does not indicate that a difference does not exist, but could simply be a lack of statistical power to observe a difference.

2) Control samples: There appear to be no non-infected individuals included in the analysis from the wet season. This seems a major omission given that dietary differences exist in wet and dry season, which could influence some metabolites.

3) Statistical analyses: There is a lack of detail on the statistical analyses generally - for example, PCA is not explained.

a) The statistical testing is rather simplistic, being mainly single variate analysis (t-tests, chi-sq), whereas a multivariate approach (such as GLMM) could be really beneficial, and may even reveal significant findings, for example a generalised linear modelling approach would allow multiple components to be analysed instead of multiple pairwise comparisons, and could also allow for repeated measures (if done correctly). The data shown in supplementary figure 6C show an apparent bimodal distribution, which an analysis of medians would miss.

b) In some cases, statistical testing is missing, but conclusions are drawn that a finding is significant, e.g. line 231-236 "parasites from asymptomatic children in the dry season...increased parasitaemia after a shorter time in culture than wet season cases" - where is the statistical analysis and P value? This analysis need to consider the parasite density in the samples since this could affect parasite growth, and not necessarily in a linear fashion - for example, nutrients can be limiting for optimum growth at very high parasitaemia. Presumably the parasitaemia in the clinical samples was much higher than in the asymptomatic cases - or were the samples matched for parasite density? Also fig 5D: "positive correlation between parasitaemia and percentage rings" - there is no correlation analysis and no P value to support this. The distribution of % rings is very broad/variable at lower parasitaemia and it is not obvious there is any correlation (at least linear).

c) Non-independent samples (repeated samples from the same individual): The metabolomic analyses included repeated samples taken at different time points from the same individual. This can be an issue for interpretation of the results as these repeated samples cannot be treated as independent samples (a requirement for many statistical tests) - it seems intuitive that an individual will have genetically or environmentally-determined variation in the levels of some metabolites. Similarities between samples from the same individual should be picked up by the PCA (as a principal component), but only if the information on individual ID was included as a variable in the analyses. It is unclear from the description of the PCA if this was done, or what PC1 and PC2 (Figure 2A) represent. The problem of repeated samples from the same individual affects, for example, the interpretation of Figure 2B - are the differences between the 5 individual asymptomatic carriers and the 5 clinical cases due to repeated measurements of the same individual skewing the results - one individual could easily do this given the very small sample size, if a particular metabolite/class of metabolite was especially low or high in that individual. Has the correlation of metabolite presence/absence between repeated samples from the same individual been analysed, to assess if there is correlation over time? Statistical tests such as Student's t-test assume independence so the use of this test would be

inappropriate e.g. Fig 2D

d) Survival analysis (Fig 6E) - a Kaplan Meier survival analysis would be appropriate here - it is unclear how this analysis has been done? An OR is reported but the Figure legend describes a Fisher's Exact test (which returns only a P value). The conclusion that the time of first appearance of a "clone" (allele) is inversely related to its survival length (lines 438-441) has no statistical support (Supplementary Fig 6C) - what is the P value/stat for the correlation?

4) Genotyping/ COI: It is not clear why different approaches have been used e.g. ama1 amplicon sequencing, msp2 genotyping, and why these have been analysed separately rather than using all the information available to draw a conclusion about the number of genotypes present. The use of a single gene to define infection multiplicity and similarity of genotypes - although ama1 diversity seems reasonable within the study population - is risky when there are some alleles at higher frequency. Multiple genes are preferred for this purpose. Please also re-consider the use of the word "clone" - you describe an ama1 allele, and the parasites carrying that allele may differ at other loci, and so may not be the same genotype (or clone).

5) Analysis of plasma: there are limitations in metabolomics, as partially acknowledged in the discussion, but another limitation is the absence of analysis of lipids - which standard metabolomics does poorly. Lipidomics is possible (but has not been applied here) - this should also be noted in the discussion as an omission, and is important given the reliance of parasites on lipids salvaged from the host for optimum growth, and their importance in signalling (for example for production of sexual stages). Another omission is seasonal differences in immune effectors in the plasma (e.g. cytokines, antibodies) which have not been considered at all. The conclusions drawn that there is no difference between dry and wet season plasma is therefore questionable - since lipids and immune effectors have not been examined?

6) Duration of infection: What is the statistical support to conclude that short infections were seen more frequently than long infections (line 419-)? In Figure C, were the age-grouped samples also matched for sex or how was this included in the analysis? A multivariate analysis here would allow both age and sex to be considered (and could also include COI, any haemoglobinopathy, etc).

Minor points

1. Were the plasma samples heat inactivated before use in the in vitro growth experiments (methods, lines 592-3, 698-700)?

2. It is unclear if the antibodies were depleted from all plasma samples before their use in the in vitro growth work. Lines 698-700 (Methods) say they were, but the text around line 212 suggests otherwise - please clarify.

3. For the data shown in Figure 4C, the p value given for the chi-sq test is $p=0.08$. This is non-significant at the 0.05 threshold, so the conclusion in the text (line 283-285) "clinical malaria cases...in presence of shared dry season clone...happened earlier in the transmission season than...newly transmitted clones" is not supported by the analysis - there is no significant difference between these two groups based on the P value.

4. Determination of parasitaemia (data, fig 5A): It is unclear how sybrgreen and mitotracker have been used to determine parasitaemia in FACS - the methodology does not explain this and there is no reference to a previous publication. For example, how are white blood cells (which would be positive for both mitotracker and Sybrgreen) distinguished from parasitised RBC? Were the parasitaemias calculated from the white blood cell counts or from the proportion of RBC that carried parasites? For Fig 5C: how are the ring stages of parasites distinguished from the later stages (trophozoites, gametocytes)? Did the methodology to define the percentage rings always type the same number of parasites (and if so, how many)? If not, the accuracy of the measurement of proportion would be affected by the number of parasites counted e.g. larger error bars with lower parasitaemia.

5. Figure 5H is too small to be useful - the images are very blurred, and it is not clear why the RBC are visible if these are thick film images? Better images are required, and it would also be helpful to non-specialists to mark the different parasite stages with different coloured arrows.

6. Duration of infection (Fig 6A): the labelling of the months is unclear - suggest to use a single letter over each column?

7. Discussion lines 529-569: while it is very likely that the immune response to PfEMP1 and other VSA affecting cytoadherence influences parasite growth and pathogenicity, this has not been examined in the current paper. This section is therefore over-long given that no conclusions can be drawn from the data presented (and indeed, the section emphasises this omission in the study design).

Point by point reply to reviewers' comments EMM-2023-18321-V2-Q "Infection length and host environment influence on Plasmodium falciparum dry season reservoir"

Referee #1 (Remarks for Author):

The manuscript by Andrade et al. describes analyses of sera and *P. falciparum* parasites from blood samples collected during the dry and wet season in Mali. The authors investigate a number of hypotheses that have been proposed to explain parasite persistence during the dry season. They used a comprehensive and well-designed series of experiments to test these hypotheses and, while the results are mostly negative, shed new lights on the mechanisms underlying parasite persistence.

I really appreciated the comprehensiveness of the analyses, the multi-pronged approaches to study both serum and parasites, and the objectivity in presenting the results. Overall, I find the study interesting and definitely worth publishing.

We thank the reviewer for the positive evaluation of our work.

I have a few minor comments that I describe below.

- the description of the serum selection for the first analysis is hard to follow (lines 151-154). The authors might want to clarify this section and maybe add a supplemental figure. The colors on Figure 1 are also very difficult to distinguish.

We agree with the reviewer that further explaining sample selection and clarifying Fig. 1 benefits the reader. We have edited the manuscript to better describe plasma selection and have changed Fig. 1 so that the colour code is more perceptible and colourblind-safe. We have also revised Fig. 1 to replace the graphic representation of study participants below the histogram of clinical malaria cases, by small square boxes displaying the number of individuals included in each serological analysis.

- the significance threshold used for the metabolite analysis (lines 161- 164) needs to be included in the text.

We have added the significance threshold used to define significance to the revised manuscript, and it can now be read in line 161 of the revised manuscript "(Pairwise student t-test, adjusted p-value < 0.05)".

Figure 2B also seems to show two distinct distributions (for the p-values under the cutoff). Why is that?

The reviewer correctly points an apparent double distribution of p-values below the significance cut-off in Fig. 2B. This arose from the inclusion of all the values for each metabolite across the different pairwise comparisons, where each metabolite has a value for each comparison. As we sorted metabolites by increasing $-\log_{10}$ of the adjusted p-values obtained from the pairwise t-tests across all sample groups, it originated the apparent distribution remarked by the reviewer. We have now edited the plot and show metabolites sorted alphabetically on the x-axis rather than by increasing p-value and such double distribution is now less striking.

- the analyses of plasma supplementation on parasite growth in vitro (lines 207+) are really exciting but the authors need to clarify when these plasma samples were collected and whether the individuals were infected at the time of collection (or shortly before).

We have edited the manuscript description of results shown in Fig. 3 to make it clearer, and added Sup fig. 3A showing no difference between supplementation with plasma from positive (50%) or negative individuals (50%). In short, in Fig. 3A all plasmas were from uninfected individuals, in Fig. 3B plasmas were from uninfected and infected individuals, and Sup fig. 3A refers to these data. In Fig. 3C-F all plasma samples were from uninfected individuals and pooled by timepoint of collection. This information is now also in the revised results section.

- it looks like *ama1* was genotyped twice independently (once by amplicon sequencing and once by 4CAST).

What was the concordance between the two approaches?

*Ama1 was genotyped by amplicon sequencing of *ama-1* alone, and by 4CAST. In the revised manuscript, we combined the genotyping results from *ama1* alone and from 4CAST sequencing based on the consensus sequences obtained. We calculated the overall frequency of each sequence across all samples with each method and present it in the new Sup fig. 5C. The data show high concordance between the two methods for highly abundant *ama1* haplotypes. We found, however, larger discrepancy when comparing shared haplotypes across methods depending on sequencing depth and overall frequency. The challenge of interpreting amplicon data and the ability to detect parasite clones is acknowledged in lines 317-321 of the revised manuscript.*

Also, the statement that "newly transmitted infections lead to clinical malarial" (line 296) could be moderated (or reinforced) by a discussion of the limit of detection of these approaches.

We have revised the manuscript to state, in lines 321-324, that our observations are constrained by the limit of detection of the techniques used.

- the numbering of the multi-panel figures (Fig 3 and 4) is really annoying (C before B)

We changed the order of the panels in both Fig. 3 and 4, so that C no longer appears before B in either case.

- the analyses presented in Fig 5 are fascinating. However, given the extensive variations observed in %Rings on

Figure 5D, I would have liked to see an analysis of the growth in vitro vs. %Rings performed at the individual parasite level rather than only by agglomerating the data by group (i.e., for the dry season samples, how is the correlation between %Rings and fold-change of iRBCs?)

In the result section we describe that the highest increase in parasitaemia between any two timepoints of the short-term culture in October was not significantly different from what we had observed in short-term culture following collections during the dry season or in malaria cases. In fact, it was the time at which the highest increase occurred that differed between the time of the year of blood collection, as highlighted in the table below:

	fold Change	95% (CI)	time of highest increase (h)	95% (CI)
DRY	2.671	(2.197, 3.005)	24	(24, 30)
OCT	2.985	(1.864, 4.119)	30	(24, 30)
MAL	2.229	(2.000,2.826)	48	(36, 48)

Hence, we do not anticipate that higher proportion of rings will necessarily promote higher parasitaemia increase, but just a later one, as shown in Fig. 5G.

We have now performed the analysis requested by the reviewer and the results are shown below. The variation observed in %Rings was minimal in the malaria cases samples, limited in the October samples, and broad in the samples from the dry season, and we found only low or non-significant correlations between the %Rings and the highest increase in parasitaemia detected in the short-term culture.

Furthermore, it is worth noting, the method used to distinguish rings from non-Rings does not allow to differentiate younger from older ring-stages of P. falciparum.

Overall, a very comprehensive and interesting study presented very objectively.

Thank you very much.

David Serre

Referee #2 (Comments on Novelty/Model System for Author):

There are some issues with justification of methods, and the MS would benefit from alternative explanations for the patterns observed. These patterns are novel and likely to be highly cited, and the study has considerable epidemiological relevance for malaria. The data was collected from humans on the most virulent malaria parasite species (*P. falciparum*) so is an entirely relevant system for human health.

We appreciate the positive appreciation of our work, and we have clarified the justification of the methods used, and we expanded our discussion to consider further alternative explanations for the patterns observed.

Referee #2 (Remarks for Author):

In this study, the authors seek to answer how *P. falciparum* persists throughout the dry seasons in asymptomatic infections when transmission isn't possible - specifically, by investigating whether parasites respond to seasonality through cues within plasma composition. Samples from infected individuals through wet and dry seasons. The authors found that i) there is no significant difference in the plasma on the basis of seasonality or presentation (asymptomatic vs symptomatic), ii) likewise, plasma acquired during wet versus dry seasons had similar effects on artificially cultured parasites, iii) through the use of sequencing to track clones, authors found that clinical malaria was caused by clones more recently transmitted during the wet season, and iv) clones transmitted later in the wet season were more likely to survive longer and thus make it to the ensuing dry season. The authors found little evidence that parasites respond to serological cues in the plasma of their hosts. Instead, the authors propose that changes in the dynamics of circulating iRBCs, specifically the timing with which iRBCs sequester, could lead to clinical differences in wet versus dry season infections.

The paper does well in introducing and motivating the questions and differentiating between wet vs dry season and asymptomatic vs clinical infections. The patterns the authors describe are extremely interesting and likely to spur a lot of useful research in this space, but a more in-depth discussion of alternative explanations for these patterns and justification of the methods would greatly enhance the study (see major comments below). In particular, the discussion of possible explanations for the dynamics of circulating iRBCs and developmental stages of infections requires more elaboration.

We appreciate the reviewer acknowledgment of the relevance of our work and we expanded the manuscript to discuss alternative explanations for the patterns observed.

Major comments

A fuller discussion of possible explanations for the observed patterns would be useful. First, the introduction suggests that the authors are considering the possibility of a parasite adaptation in the form of sensing pathways to enable infections to persist through the dry season, but it is not clear how their primary explanation for observed patterns—parasites extending circulation times and being killed at higher rates in the spleen—would help parasites persist.

We have now made clearer in the introduction that our previously published data supports that low parasitaemias in persistent infections at the end of the dry season are kept below the immunological and clinical radars through increased splenic clearance of iRBCs that circulate longer and become at risk of removal in the spleen.

Second, the positive association between ring stage parasites and iRBC burden is exactly what would be expected according to population dynamics. An expanding parasite population (i.e., one that generates a large percentage of iRBCs) would show a bias towards earlier developmental ages (a higher percentage of rings). This pattern has been known for populations of any organism for decades and noted for malaria parasites specifically (e.g., Khoury et al. 2014 *Infect Immun*; Greischar et al. 2019 *Trends Parasitol*). The authors need to provide more explanation and justification for why they favor the hypothesis of increased circulation times over the alternative, that some later stages circulate in both wet and dry season infections but that those stages are harder to find in wet season infections because those infections are expanding.

As mentioned in response to similar comment from reviewer 1, in the result section we describe that the highest increase in parasitaemia between any two timepoints of the short-term culture in October was not significantly different from that observed in short-term culture after collections during the dry season or in malaria cases. It was, in fact, the time at which the highest increase occurred that was different between the time of the year of blood collection. Our data suggests equal ability to produce iRBCs year-round, and we do not anticipate that higher proportion of rings will promote higher increases in parasitaemia, just later ones, as seen in Fig. 5 G. Additionally, we have now added a paragraph to the discussion of the revised manuscript to acknowledge alternative explanations to the patterns observed, as suggested by the reviewer.

Third, highlighting other possibilities in the discussion would make this study more impactful, including the possibility that dry season infections show more later stages because they persist longer and have had longer to desynchronize. Wet season infections will be in the early portion of infection, since these infections do not last long, and may not have had time to desynchronize.

We have now included this point to the discussion lines-543-560.

Changing investment into transmission stage production would also alter rates of replication in the blood and may explain the dynamics of chronic infection in other hosts (e.g., canaries, Cornet et al. 2014 PLoS Path). Do the authors have reason to believe altered transmission investment strategies could not explain the patterns they see? If the authors have on transmission stages, it could bolster their conclusions. In lieu of such data, it would be helpful for the authors to discuss the potential role of transmission investment strategies in persistent infections and transmission during wet seasons.

Only a small proportion of intra-erythrocytic asexual parasites differentiate into sexual development (Kafsak et al. 2014), so we believe this would have a negligible effect on rates of replication in the blood. Although we have not quantified gametocyte carriage in the samples used in this study, we acknowledge the critical importance of sexual stages in resuming transmission as the rainy season returns. Gametocytes have been shown not to last longer than ~20 days in circulation, hence maintaining asexual replication during the dry season may be the best way to ensure a proportion of parasites converting into gametocytes when mosquitoes return. We agree that it is of great interest to investigate parasite transmissibility over the course of the dry season, and indeed, in a separate set of longitudinal experiments, we compared sexual conversion and gametocytogenesis of parasites collected from subjects at several points during the dry season versus subjects with acute symptomatic malaria and asymptomatic infection during the wet season, which is available as a preprint (doi: <https://doi.org/10.1101/2021.11.05.467456>). These data from the same village show a continuous investment in sexual conversion, which will promote a higher proportion of gametocytes when asexual stages are decreasing then when parasitaemias are increasing, due to the almost two-week long lag between sexual conversion and fully formed gametocytes in circulation. We feel however that these data are out of the scope of our current manuscript.

Is the pattern necessarily driven by parasites or could host differences in the dry season (e.g., immune exhaustion/depletion following intense wet season transmission) be responsible? Discussing these key points would help guide future work in this space

In our 2020 publication, we have extensively compared immunity of dry season asymptomatic carriers vs PCR negative individuals and found that asymptomatic carriage of *P. falciparum* imposed little to no alterations of immune response. Specifically, it did not elicit detectable inflammation or affect immune cell function, and it did not prevent loss of antibody responses during the dry season. We have now made this statement clearer in the discussion of the revised version of the manuscript lines 574-578.

More context and in some cases justification of the methods is needed to bolster the results. Some rationale and further citations would be useful on possible cues in plasma that could trigger changes in parasite strategies (paragraph starting at line 102).

We describe in the introduction of the manuscript several reported sensing responses and mechanisms by *P. falciparum* and other malaria parasites throughout the lifecycle, and in the mentioned paragraph we present our unbiased approach to investigate potential seasonal differences in plasma composition of Malian individuals and its impact in *P. falciparum* in vitro culture. We reworded the referred paragraph for improved clarity.

Counts of iRBCs were made in comparison to 300 leukocytes (lines 601-602), but are the authors confident that there were no systematic differences in leukocyte counts between hosts in the wet and dry seasons (and if so, why)? Studies suggest other methods (e.g., counting to a particular number of fields) are more accurate than counting to a particular number of leukocytes (Kilian et al. 2000 Trop Med Hyg Int Health, O'Meara et al. 2005 Am J Trop Med Hyg), and that could be especially true if there are systematic differences in mean leukocyte counts across season. The authors' previous study (Andrade et al. 2020) seems to show no systematic differences in leukocyte composition but a case needs to be made that overall leukocyte abundance is also similar across seasons.

Microscopy count of iRBCs was made relative to 300 leukocytes in the context of the clinical study to define whether a study participant had or not clinical malaria, as this is the established and approved method of the clinical study protocol. Parasitaemias and parasitaemia fold change shown in Fig. 3 and Fig. 5 were determined by flow cytometry. We have now made it clearer in the manuscript and edited the methods section accordingly.

Minor

- Lines 58-60. This finding needs more explanation so that readers can get the key point without going back to read the previous paper. The phrasing leaves the time scale open to misinterpretation-circulation could refer to persistence in the blood rather than the very specific meaning of circulating as opposed to sequestered iRBCs. Adding a sentence to orient readers to malaria biology and terminology would be helpful.

We have rephrased to provide more detail and avoid confusion.

- Lines 66-67: The phrase "rigidity of substrate" is a bit mysterious in this context-rephrase or explain?

We have rephrased it to "contact with substrates of different rigidities"

- Line 218: Comparable in what way? Supplementary fig. 2 shows similar SYBR Green uptake (for which a brief explanation of the biological significance of that measure would be very helpful) and similar change in percent

iRBCs through time, but the phrasing in the main text could mean comparable intraerythrocytic developmental duration, or comparable multiplication rates.

We have rephrased it to “of the geometric mean fluorescence of nuclear staining (SybrGreen) revealed comparable parasite development in infected erythrocytes following in vitro culture.” The methods were also updated to better describe this analysis.

• Line 223: Could the authors add a few more words to explain the meaning of 'demonstrated sensing ability'?

We have rephrased it to “previously shown to respond to Lyso-PC levels in vitro”.

• Line 397: 'individual was found PCR' negative or positive presumably? *This has now been corrected to “PCR-”.*

• Line 477: late in the preceding wet season? *Yes, we have added preceding to that line for clarity.*

• In figure 1, the colored human shapes are a bit confusing as to what they represent - a brief explanation in the caption or text explaining what each person is (a set of participants, how many, etc.) should suffice.

We have revised Fig. 1 and its caption for improved clarity.

• Fig 2: Pathway analysis, and what pathway impact entails, requires more explanation.

We have moved the pathway analysis plot to Sup fig. 1 and rephrased the result and method section to clarify the reader. Pathway enrichment analysis is a computational biology method used to identify biological functions that may be overrepresented in a group of genes or metabolites more than would be expected by chance and ranks these functions by relevance. Identification of a given pathway as the most affected between two groups of samples entails a potential impact of that biological function differently between the two groups of samples. In our analysis the identification of amino acid- and glycerophospholipid- pathways highlights their depletion from plasma of children with clinical malaria due to high number of parasites. We now mention this in the discussion.

• The tracking of clones throughout the seasons and their survival time is a supporting pillar to one of the paper's conclusions (lines 434, 500, 513), and a supplemental graph that actually shows the timelines of the different clones - from when they were acquired (including wet and dry season) to how long they likely survived - would be useful in illustrating the finding that parasites acquired earlier in the wet season had shorter survival/those transmitted later in the wet season were more likely to last the dry season.

Following the reviewer's suggestion, we have now included as part of the Sup Fig. 6 two different visualisations of the whole set of ama1 haplotypes included in the infection length analysis (n=844 allowing 2 skips). The top panel shows haplotypes ordered by length, highlighting that clones appear and disappear throughout the wet season, and showing that short infections appear throughout the wet season, while long infections, particularly the ones bridging two wet seasons are transmitted late in the wet season. On the bottom, we included the distribution of clones and their infection length showing a skew towards clones appearing and disappearing during the wet season.

The authors offer to make their code available upon request, but the study would be more likely to be used (and cited) if the code were made publicly available (e.g., on Github).

We appreciate the reviewer's suggestion and have now included the code allowing interested users to determine haplotype length adopting the desired number of allowed skips between positive timepoints, as described in this work. The code can be found under the URL https://github.com/macarrasq/falciparum_infection_length.git in a personal github account, and this information is now added to the data availability section of our manuscript.

Referee #3 (Comments on Novelty/Model System for Author):

See main review below for full details.

Statistical analysis is poor - recommend engaging with a statistician.

Novelty and medical impact are low, mainly due to failure to find significant correlations.

The paper does not include work with a model organism - the actual parasite causing human malaria is used, and the samples are from naturally infected children.

Ethical issues are covered appropriately.

Referee #3 (Remarks for Author):

In this study the authors have investigated factors contributing to the persistence of human malaria parasites (*P. falciparum*) in the human body during the long dry season where there is no mosquito transmission. They focus on potential cues in the blood that the parasite could respond to in order to change growth characteristics, by analysing the metabolome and proteome within plasma at different time periods throughout the year. Alternatively they hypothesise that parasite persistence is related to the length of an infection.

Their results are largely negative - they failed to find any significant differences in the metabolites and proteins present in the plasma taken in the wet and dry seasons, nor in their impact on parasite growth in vitro. They then conclude that this means that there cannot be a sensing mechanism in parasites to adjust their growth/ behaviour seasonally. This conclusion is not convincing, since not all components of plasma have been analysed - significant omissions are lipids and immune effectors, and they did not investigate sensing mechanisms, but only potential cues. "Absence of Evidence is not Evidence of Absence" [Sagan]. A second analysis concludes that parasites transmitted later in the wet season had a higher chance of surviving the long dry season, and that persistence is linked to the length of infection. The statistical analyses to support this conclusion are not wholly convincing.

We agree that lack of evidence does not prove absence and we state in the discussion of our manuscript that "the absence of in vitro evidence of a sensing mechanism could be attributed to the inability to fully reproduce the seasonal environment in vitro".

We provide evidence of minimal differences in metabolites, proteins and now lipids, as suggested by the reviewer, between plasma samples collected along the year from children without and without infection. Furthermore, we have followed the reviewer suggestion to improve the statistical methods used throughout the paper and can now more confidently say that parasites transmitted later in the wet season have a higher chance of surviving the long dry season.

Parasite persistence over long periods in the absence of reinfection from mosquitoes is poorly understood, and this paper adds a little to what is known already, but generally reinforces previous work from the same group and from others. There is relatively little of novelty presented, largely because the study failed to find differences between wet and dry season host plasma. The paper is unlikely to be of strong interest to more than a specialised audience.

The main shortcomings of the manuscript are the small number of individuals that have been included in the metabolome and proteome study, the omission of analyses of lipid content and immune effectors in the plasma, and the weakness of the statistical approaches. Failure to find a difference does not mean that there is not a difference if the study is underpowered. It is possible that with more sophisticated statistical approaches, for example multivariate modelling approaches, stronger conclusions could be drawn, and I would recommend that the authors engage with a statistician/modeller if they have not done so already. Another weakness is the lack of detail provided in the methodology at various points (given below) - while it is possible that some of the methods have been published previously, this information is not given either, and so a reader wishing to understand at a deeper level what has been done is out of luck.

We have clarified the sample and controls included/presented in the metabolome and proteome analyses, and we have added a lipidomic analysis to the manuscript including 19 samples of asymptomatic carriers and 20 uninfected individuals during the dry season and 18 malaria cases in the wet season. And we have amended the description and justification of the methods used in the revised manuscript.

We disagree with the statement from the reviewer that failing to find differences equates to "little of novelty presented", and argue that presentation of negative findings from newly performed experiments and analyses are of value to the scientific community.

Major Points

1) Sample size: The number of individuals in the metabolomics study was surprisingly small - they appear to have included only 5 infected, and 6 parasite -free individuals during the dry season, with 5 age- and sex-matched individuals in the wet season (presenting with clinical malaria). What is the statistical power given these low numbers to see any significant difference in the presence, absence or levels of any of the 456 metabolites examined? The absence of a significant difference between groups, given these low numbers, does not indicate that a difference does not exist, but could simply be a lack of statistical power to observe a difference.

We agree that an investigation of statistical power may help with the interpretation of our results. In response, we have conducted a power analysis using the sample sizes for each of our experimental groups (asymptomatic, $n = 11$; negative, $n = 21$; dry season, $n = 32$; October, $n = 11$; MAL, $n = 6$) and with the statistical test used to detect differences in metabolite abundances (a two-sided t-test). Given that a total of 456 metabolites were investigated, we included a Benjamini-Hochberg (BH) multiple-tests correction in our original analysis, thereby fixing the false-discovery rate at 5%; however, as this method adjusts based on the rank-order distribution of observed p-values, incorporating it into a power analysis is complicated and would require making distributional assumptions about effect sizes. Instead, we provide a conservative lower bound on the power using a Bonferroni (BF) adjusted t-test, which controls family-wise error rate and is more stringent than a BH-correction. For comparison, we also provide a power curve without adjustment for multiple correction as a maximum upper bound. Note that for power analyses of t-tests it is customary to represent effect sizes with Cohen's d statistic, which is calculated as the difference in means of the groups divided by a pooled estimate of the standard deviation.

From this analysis can be seen that, indeed, we would be powered to consistently detect small effects, such as where the difference in means between the two groups was close to their standard deviation (e.g. Cohen's $d \sim 1$). However, for larger effects, we would be able to detect differences in abundances between metabolites with our sample sizes and approach. We have the most power comparing the dry season vs. October and asymptomatic vs. negative groups – where we would expect to detect effect sizes of in the $d \sim 1$ to 2 range 80% of the time. Interestingly, where we have the worst power (negative vs MAL; asymptomatic vs MAL) is where we actually detect significant differences in metabolite abundances; indicating that for our biological scenario large effects are occurring.

We agree with the reviewer that higher numbers of samples would give us more power to detect small differences in metabolite concentrations. However, for some experiments it is not always logistically or financially feasible to generate large enough data sets to completely satisfy power calculations.

2) Control samples: There appear to be no non-infected individuals included in the analysis from the wet season. This seems a major omission given that dietary differences exist in wet and dry season, which could influence some metabolites.

Our analyses include wet season (Oct) non-infected individuals in the metabolome and proteome data. We have added a scheme to Fig. 1, where samples in the different serological analyses are shown and the number of replicates is mentioned. Additionally, we added an annotation column to the heatmap presented in Fig. 2C which splits samples by month of collection and infectious status, showing that in the wet season there are negative and asymptomatic positive samples, which addresses the query about dietary differences across seasons.

3) Statistical analyses: There is a lack of detail on the statistical analyses generally - for example, PCA is not explained.

We have now further detailed the PCA, how it was performed from the filtered and normalized data set in the methods section. We applied a filtering threshold of missingness of 0.8 for metabolites and for 0.9 for individuals.

a) The statistical testing is rather simplistic, being mainly single variate analysis (t-tests, chi-sq), whereas a multivariate approach (such as GLMM) could be really beneficial, and may even reveal significant findings, for example a generalised linear modelling approach would allow multiple components to be analysed instead of

multiple pairwise comparisons, and could also allow for repeated measures (if done correctly). The data shown in supplementary figure 6C show an apparent bimodal distribution, which an analysis of medians would miss.

The reviewer's remark is important and we have taken steps to address it, given the bimodal distribution in Fig. 6C Sup fig 6A, B and D. We have endeavoured to address the concerns by two different methods. First, we calculated the mean response (infection length in days) per individual, followed by a multivariate ANOVA to see the fixed effects of haemoglobin type, sex and age group to the response variable (average clonal length). With this approach (plot below), we did not find any variable to be statistically significant with clonal infection length and we do not see the same effect of bimodality using this approach, and in most cases the data seems to be normally distributed (except perhaps for the impact of being male on the clonal length of infection). Even for the comparison between the two age groups we obtained a non-significant association ($p=0.085$), while we found a significant difference when comparing the median length all clonal ($p\text{-value} = 0.036$).

Furthermore, we have performed a linear mixed effects model (LMM), which allows the inclusion of random effects from the individuals contained in the analysis (differences between hosts that might impact the response variable). With this approach we determined statistically significant differences in age groups, with the 10-13 age group having a significantly lower response variable (clonal infection length), than the younger age group (7-9), even when accounting for fixed and random effects, supporting the result shown on the current version of the manuscript.

Lastly, we produced a Kaplan-Meier survival analysis as suggested by the reviewer and found also a statistically significant association between the probability of a clone surviving until the end of the dry season and the younger age group (7-9). This supports the current finding shown in figure 6D for which we did a contingency table analysis of individuals in the two age groups and the proportions of clones either persisting or being cleared.

b) In some cases, statistical testing is missing, but conclusions are drawn that a finding is significant, e.g. line 231-236 "parasites from asymptomatic children in the dry season...increased parasitaemia after a shorter time in culture than wet season cases" - where is the statistical analysis and P value?

We appreciate the reviewer's comment and we have attempted to add further statistical support to some of our analyses. To address the specific query, we have now combined the dry season data of all media supplementation conditions shown in Fig. 3C and compared it to clinical malaria samples combined, we observe significantly higher fold changes in the dry season samples at 24 and 36 h post-culture indicating earlier parasitaemia increase. Now shown in Fig. 3D.

This analysis need to consider the parasite density in the samples since this could affect parasite growth, and not necessarily in a linear fashion - for example, nutrients can be limiting for optimum growth at very high parasitaemia. Presumably the parasitaemia in the clinical samples was much higher than in the asymptomatic cases - or were the samples matched for parasite density?

The reviewer's concerns that ex-vivo parasite growth might be suboptimal due to the varying parasitaemias and/or limitation of media nutrients is valid. We have now made clearer in the methods that samples collected from malaria cases shown in Fig. 3C were diluted 100x to allow for optimal parasite development and replication. The data shown in Fig. 5E, malaria case samples were cultured undiluted and diluted 1:10, 1:25 and 1:50 with non-infected blood to ensure that initial parasitaemia was low (0.5–1%) and that all cultured parasites could grow to their maximum potential. Moreover, parasite growth (measured as highest fold change of %iRBCs in Fig. 5E) from clinical malaria samples is identical to asymptomatic dry or wet season ones (albeit at a different timepoint), highlighting that parasite growth was not compromised in these conditions.

Also fig 5D: "positive correlation between parasitaemia and percentage rings" - there is no correlation analysis and no P value to support this. The distribution of % rings is very broad/variable at lower parasitaemia and it is not obvious there is any correlation (at least linear).

The reviewer is correct that our choice of words was not the best and we have reworded the description of Fig. 5D. While we could find significant linear correlation between parasitaemia and % rings in the asymptomatic samples of the dry and the wet season, the malaria cases samples presented 100% rings in almost all samples, making it difficult to detect any correlation.

Sample_type
 a DrySeason
 x Oct
 ■ MAL

c) Non-independent samples (repeated samples from the same individual): The metabolomic analyses included repeated samples taken at different time points from the same individual. This can be an issue for interpretation of the results as these repeated samples cannot be treated as independent samples (a requirement for many statistical tests) - it seems intuitive that an individual will have genetically or environmentally-determined variation in the levels of some metabolites. Similarities between samples from the same individual should be picked up by the PCA (as a principal component), but only if the information on individual ID was included as a variable in the analyses. It is unclear from the description of the PCA if this was done, or what PC1 and PC2 (Figure 2A) represent. The problem of repeated samples from the same individual affects, for example, the interpretation of Figure 2B - are the differences between the 5 individual asymptomatic carriers and the 5 clinical cases due to repeated measurements of the same individual skewing the results - one individual could easily do this given the very small sample size, if a particular metabolite/class of metabolite was especially low or high in that individual. Has the correlation of metabolite presence/absence between repeated samples from the same individual been analysed, to assess if there is correlation over time? Statistical tests such as Student's t-test assume independence so the use of this test would be inappropriate e.g. Fig 2D

Regarding the comment on whether samples are non-independent, we have considered the possibility of across and within host variability that may suggest non-independence. First, we employed analysis by principal components and show no particular grouping of samples by individual, regardless of infectious status or time of year, suggesting that these samples can indeed be treated as independent and therefore support the subsequent statistical tests performed for the metabolomics analysis, and this is now included in Sup fig.1.

Furthermore, we have determined metabolite abundance over time the full set of 12 individuals and observe relatively consistent abundances regardless of individual or time of year, also included in Sup fig.1.

d) Survival analysis (Fig 6E) - a Kaplan Meier survival analysis would be appropriate here - it is unclear how this analysis has been done? An OR is reported but the Figure legend describes a Fisher's Exact test (which returns only a P value).

We appreciate the reviewer's comment, and have revisited the analysis on the infection length using different methods also suggested by the reviewer in an earlier point. We now use a GLMM contingency table analysis, since each infection exists on a timeline (as currently shown in fig 6E). Essentially, each clonal infection has a start (early or late in the wet season) and an end/outcome (persistent until the end of the dry season or cleared throughout the wet season). What we had described previously was a 2x2 contingency table adding up all the clonal infections regardless of number of repeated measurements per subject, so in order to handle repeated measurements (multiple timepoints and different "end" per subject), we generated a contingency table as the example below, where each subject will have multiple rows depending on the clones per timepoint:

ID	Start_of_infection	End_of_infection	Count
Kali0070	Early	Cleared	5
Kali0070	Late	Persistent	2
Kali0070	Late	Cleared	2
Kali0070	Early	Persistent	0
Kali0072	Early	Cleared	5
Kali0072	Late	Persistent	2
Kali0072	Late	Cleared	5
Kali0072	Early	Persistent	1

We used Count as a response variable (how many clones are persistent/cleared and were transmitted late or early) so the model will have 3 fixed effects: start of the infection, end of the infection or the interaction term which refers to the relationship between start and end of infection. We first attempted a Poisson distribution model and found a statistically significant association of the 2-way interaction between start of infection (early or late) and the end of it (persistent or cleared), with a p-value of 5.27e-9. To interpret the results, we then computed the odds ratio from the estimate using the exponential function which resulted in an odds ratio of 3.323378, therefore the odds of a persistent "end" of infection are 3.32 times higher for Lately transmitted clones than for earlier ones.

Furthermore, we performed a negative binomial GLMM model, for which we can include a random subject effect and obtained a very similar result as with the Poisson. We again obtained a statistically significant interaction

between start and end of infection (p-value 9.9e-10, odds-ratio 3.32486455541021). Similarly, because of the repeated observations, we also performed other types of GLMM models. Our concern was that there would be many zeroes in some observations, for example that within one timepoint one subject has zero clones that persist vs many clones that are cleared throughout the dry season. For this, we used a negative binomial with zero inflation parameter and obtained almost identical result as with the Poisson model (p-value 5.4e-19 and odds ratio of 3.34355259840936). We have modified the text accordingly and below are the results of the 3 models, showing that the model with the highest log-likelihood and lowest Akaike Information Criteria is the Poisson model, therefore is the one we have reported in the revised manuscript.

Model	Log-Likelihood	AIC
GLMM Poisson (lme4)	-416.7	843.4
GLMM Neg Binom (lme4)	-416.4370	844.8741
GLMM ZINB (glmmADMB)	-415.67	845.34

The conclusion that the time of first appearance of a "clone" (allele) is inversely related to its survival length (lines 438-441) has no statistical support (Supplementary Fig 6C) - what is the P value/stat for the correlation?

The reviewer is correct that do not show a correlation in Sup fig. 6C. We have reworded the statement at the end of the result section in the revised manuscript.

4) Genotyping/ COI: It is not clear why different approaches have been used e.g. ama1 amplicon sequencing, msp2 genotyping, and why these have been analysed separately rather than using all the information available to draw a conclusion about the number of genotypes present. The use of a single gene to define infection multiplicity and similarity of genotypes - although ama1 diversity seems reasonable within the study population - is risky when there are some alleles at higher frequency. Multiple genes are preferred for this purpose. Please also re-consider the use of the word "clone" - you describe an ama1 allele, and the parasites carrying that allele may differ at other loci, and so may not be the same genotype (or clone).

We conducted the ama1 analysis with samples available in the lab, and included data already collected from a previous analysis with mps2 that remained unpublished from which 7 samples fitted the criteria of this question. We could remove it, but it is unpublished data and the results are consistent with the larger ama1 dataset, and support our conclusions.

Highly frequent ama1 alleles is indeed something we cannot fully correct for without further loci and we acknowledge this limitation in the revised version of the manuscript

We have revised the allele wording.

5) Analysis of plasma: there are limitations in metabolomics, as partially acknowledged in the discussion, but another limitation is the absence of analysis of lipids - which standard metabolomics does poorly. Lipidomics is possible (but has not been applied here) - this should also be noted in the discussion as an omission, and is important given the reliance of parasites on lipids salvaged from the host for optimum growth, and their importance in signalling (for example for production of sexual stages). Another omission is seasonal differences in immune effectors in the plasma (e.g. cytokines, antibodies) which have not been considered at all. The conclusions drawn that there is no difference between dry and wet season plasma is therefore questionable - since lipids and immune effectors have not been examined?

We have added in new Fig. 2D and Supplementary table 4 a lipidomic analysis by Liquid chromatography-mass spectrometry (LC-MS) of 21 lipids present in the plasma of children infected during the dry season (March, n=19, May n=19), versus those of age- and sex-matched negative individuals at the end of the dry season (n=20) children, and children showing clinical malaria symptoms in the wet season (MAL, n=18). PCA of these data showed no separation of samples by either time of the year, infectious status or clinical manifestation, and the lipids found at significantly different concentrations support the observations in the metabolomic analysis, and also highlight differences not detected by the metabolomic.

Cytokines and immune effectors have been thoroughly examined in children with and without asymptomatic infections at the beginning and end of the dry season in our 2020 publication and no differences were detected, which we now mention in the introduction and discussion of the revised manuscript.

6) Duration of infection: What is the statistical support to conclude that short infections were seen more frequently than long infections (line 419-)? In Figure C, were the age-grouped samples also matched for sex or how was this included in the analysis? A multivariate analysis here would allow both age and sex to be considered (and could also include COI, any haemoglobinopathy, etc).

We conclude that most infection are short, because out of the 844 individual infections counted in the 92 children more than 400 had durations below 49 days (which we defined as short) and only 284 were long (which we defined as >129 days). We attempted to make the text clearer. We have now performed a multivariate analysis which we described earlier and will help address this point.

Minor points

1. Were the plasma samples heat inactivated before use in the in vitro growth experiments (methods, lines 592-3, 698-700)? Plasma samples used in sensing experiments were not heat-inactivated.

2. It is unclear if the antibodies were depleted from all plasma samples before their use in the in vitro growth work. Lines 698-700 (Methods) say they were, but the text around line 212 suggests otherwise - please clarify. Plasma samples in Fig. 3A and B were not antibody-depleted, and all other figures and sup figures related with sensing experiments were antibody-depleted. We have clarified it in the revised version of the manuscript.

3. For the data shown in Figure 4C, the p value given for the chi-sq test is $p=0.08$. This is non-significant at the 0.05 threshold, so the conclusion in the text (line 283-285) "clinical malaria cases...in presence of shared dry season clone...happened earlier in the transmission season than...newly transmitted clones" is not supported by the analysis - there is no significant difference between these two groups based on the P value. We have clarified it in the revised version of the manuscript that no statistical significance was observed.

4. Determination of parasitaemia (data, fig 5A): It is unclear how sybrgreen and mitotracker have been used to determine parasitaemia in FACS - the methodology does not explain this and there is no reference to a previous publication. For example, how are white blood cells (which would be positive for both mitotracker and Sybrgreen) distinguished from parasitised RBC? Were the parasitaemias calculated from the white blood cell counts or from the proportion of RBC that carried parasites? For Fig 5C: how are the ring stages of parasites distinguished from the later stages (trophozoites, gametocytes)? Did the methodology to define the percentage rings always type the same number of parasites (and if so, how many)? If not, the accuracy of the measurement of proportion would be affected by the number of parasites counted e.g. larger error bars with lower parasitaemia.

RBCs and leucocytes are easily distinguishable by flow cytometry regardless of fluorescence markers. Their size and granularity allow the forward scatter and side scatter of light to clearly separate them. Furthermore, RBC pellets from infected individuals used in our study were drawn by venepuncture and collection with Cellular Preparation Tubes (CPTs) that separate the majority of mononuclear cells from the RBCs at the bottom. Some neutrophils will make it into the RBC pellets, but are easily gated out because of their greater size and granularity. Every staining included RBCs from a negative individual confirming no contamination of leucocytes in the RBC gate, which could then not be evaluated for its sybrgreen and mitotracker content in the iRBC gate. Collection of RBC pellet from CPT tubes was now clarified in the method section, and Sup fig. 8 was added to exemplify the gating strategy.

5. Figure 5H is too small to be useful - the images are very blurred, and it is not clear why the RBC are visible if these are thick film images? Better images are required, and it would also be helpful to non-specialists to mark the different parasite stages with different coloured arrows.

The images have now been cropped to make the parasites bigger and more visible in each panel. A description in the figure legend now guides the reader to the different parasite stages observed. Of note, in the previous images (uncropped) only a white blood cell was present, and remnants of RBCs, which are both commonly visible in thick blood smears.

6. Duration of infection (Fig 6A): the labelling of the months is unclear - suggest to use a single letter over each column?

Because the sampling is performed twice per month (total of 12 timepoints), we opted for having the months evenly spaced over the columns. We have now added July so that each month has two columns of data, because the first timepoint is at the end of July and the last one is at the end of December.

7. Discussion lines 529-569: while it is very likely that the immune response to PfEMP1 and other VSA affecting cytoadherence influences parasite growth and pathogenicity, this has not been examined in the current paper. This section is therefore over-long given that no conclusions can be drawn from the data presented (and indeed, the section emphasises this omission in the study design).

We have attempted to shorten the discussion throughout and particularly on this topic, but consider the immune response to PfEMP1 and other VSA a relevant theme to the data presented, which may guide other readers to engage on future developments on this field.

9th Apr 2024

Dear Dr. Portugal,

Thank you for the resubmission of your work to EMBO Molecular Medicine. We have now heard back from two of the original referees who had evaluated your manuscript. As you will see below, the reviewers find that there are still important concerns that have not been fully addressed, and therefore we would ask you to further revise your manuscript. As Reviewer 3 was not able to re-review, we had a cross-commenting session in which Reviewers 1 and 2 were requested to look over your responses to Reviewer 3, and they continue to have concerns on the revisions and responses to Reviewer 3 that will also need to be addressed. These points have been copied below along with the reviews.

Addressing the reviewers' concerns in full in a point-by-point response will be necessary for further considering the manuscript in our journal, and acceptance of the manuscript will entail another round of review. As EMBO Molecular Medicine generally encourages only a single round of revision, we would advise you to address the remaining comments as fully as possible. If you would like to discuss further the points raised by the referees, I am available to do so via email or video. Let me know if you are interested in this option.

As requested in the previous decision letter, please ensure that your revised manuscript includes the following requested item; failure to include requested items will delay the evaluation of your revision.

We require:

4) A .docx formatted letter INCLUDING the reviewers' reports and your detailed point-by-point responses to their comments. As part of the EMBO Press transparent editorial process, the point-by-point response is part of the Review Process File (RPF), which will be published alongside your paper.

5) A complete author checklist, which you can download from our author guidelines (<https://www.embopress.org/page/journal/17574684/authorguide#submissionofrevisions>). Please insert information in the checklist that is also reflected in the manuscript. The completed author checklist will also be part of the RPF.

6) Please note that all corresponding authors are required to supply an ORCID ID for their name upon submission of a revised manuscript.

7) It is mandatory to include a 'Data Availability' section after the Materials and Methods. Before submitting your revision, primary datasets produced in this study need to be deposited in an appropriate public database, and the accession numbers and database listed under 'Data Availability'. Please remember to provide a reviewer password if the datasets are not yet public (see <https://www.embopress.org/page/journal/17574684/authorguide#dataavailability>).

In case you have no data that requires deposition in a public database, please state so in this section. Note that the Data Availability Section is restricted to new primary data that are part of this study. This study includes no data deposited in external repositories.

8) For data quantification: please specify the name of the statistical test used to generate error bars and P values, the number (n) of independent experiments (specify technical or biological replicates) underlying each data point and the test used to calculate p-values in each figure legend. The figure legends should contain a basic description of n, P and the test applied. Graphs must include a description of the bars and the error bars (s.d., s.e.m.). Please provide exact p values.

9) Our journal encourages inclusion of *data citations in the reference list* to directly cite datasets that were re-used and obtained from public databases. Data citations in the article text are distinct from normal bibliographical citations and should directly link to the database records from which the data can be accessed. In the main text, data citations are formatted as follows: "Data ref: Smith et al, 2001" or "Data ref: NCBI Sequence Read Archive PRJNA342805, 2017". In the Reference list, data citations must be labeled with "[DATASET]". A data reference must provide the database name, accession

number/identifiers and a resolvable link to the landing page from which the data can be accessed at the end of the reference. Further instructions are available at .

13) Author contributions: CRediT has replaced the traditional author contributions section because it offers a systematic machine readable author contributions format that allows for more effective research assessment. Please remove the Authors Contributions from the manuscript and use the free text boxes beneath each contributing author's name in our system to add specific details on the author's contribution. More information is available in our guide to authors.

Share synopsis text and image, as well as eTOC:

Please note that these would be the final versions and changes during proofing are usually not allowed

16) As part of the EMBO Publications transparent editorial process initiative (see our policy here:

https://www.embopress.org/transparent-process#Review_Process), EMBO Molecular Medicine will publish online a Peer Review File (PRF) to accompany accepted manuscripts.

In the event of acceptance, this file will be published in conjunction with your paper and will include the anonymous referee reports, your point-by-point response and all pertinent correspondence relating to the manuscript. Let us know whether you agree with the publication of the PRF and as here, if you want to remove or not any figures from it prior to publication.

I look forward to seeing a revised form of your manuscript as soon as possible.

Yours sincerely,

Poonam Bheda

Poonam Bheda, PhD
Scientific Editor
EMBO Molecular Medicine

Use this link to login to the manuscript system and submit your revision: <https://embomolmed.msubmit.net/cgi-bin/main.plex>

***** Reviewer's comments *****

Referee #1 (Remarks for Author):

The authors have addressed my main concerns and I have no further comments.

Referee #2 (Remarks for Author):

The authors have made important changes in this revision, but the central argument still needs further explanation and justification. One of the major points is that the parasites transmitted later in the season are more likely to survive to the end of the dry season and thus more likely to transmit again, but without further elaboration it is not clear whether this pattern reflects a change in parasite allocation/life history strategies. It seems like the authors are not arguing for a winnowing-down of the faster growing parasite genetic variants in infections over the course of the dry season, leaving only the variants with poor ability to sequester (and hence cause clinical disease). Instead, the authors seem to be alluding to differences in var gene expression and not in genetic background of different variants, such that late in the dry season strains are forced to use var genes that do not allow efficient sequestration. When transmitted at the beginning of the wet season, these strains can once again use their full var gene repertoire to enable fast growth within hosts. If that is the argument, it needs to be laid out clearly. (Note that all line numbers refer to the clean draft without track-changes.)

Generally, much of what's written in the point-by-point responses could benefit the paper if included. Point 6 in the response letter: "Only a small proportion of intra-erythrocytic asexual parasites differentiate into sexual development (Kafsak et al. 2014), so we believe this would have a negligible effect on rates of replication in the blood. Although we have not quantified gametocyte carriage in the samples used in this study, we acknowledge the critical importance of sexual stages in resuming transmission as the rainy season returns. Gametocytes have been shown not to last longer than ~20 days in circulation, hence maintaining asexual replication during the dry season may be the best way to ensure a proportion of parasites converting into gametocytes when mosquitoes return." This justification could be included and elaborated on in the main paper. Though the authors discount the role of transmission investment, that is another mechanism by which parasite strains could slow their growth (see theory from Koella & Antia 1995 Theor Pop Biol about the potential for such changes in investment to be adaptive in evading immune clearance), and depending on environmental conditions, transmission investment can be enormous depending on in vitro conditions (up to 70% see Bruce et al. 1990 Parasitology).

The discussion of results for lipids needs a lot more context to justify the section header "Seasonality and asymptomatic infections have little to no detectable effect on the plasma metabolites, lipids and proteins". What should readers make of the finding that one Lyso-PC species (Lyso-PC 20:4) was significantly increased in malaria cases (compared to what?) and another, Lyso-PC 18:0 was significantly decreased in malaria cases versus dry season infected or negative individuals? Are there dose response curves in the literature for these other Lyso-PC species and their impact on sexual differentiation (i.e., something like Fig. 2C in Brancucci et al. 2017 Cell for Lyso-PC 16:0)? Is there a reason Lyso-PC 16:0 wasn't included in this analysis (was it not present)?

It would be helpful for the authors to lay out their logic of their hypothesis explicitly and compare expected patterns to a clearly defined null hypothesis. For example, the authors put a lot of weight on the timing of the highest increase in parasitemia, but the logic for when the largest increase would be expected under their hypothesis and under the null expectation is not explained. Is the null hypothesis that the parasite population is asynchronous and multiplying at a constant rate? In that case, the largest increase in %iRBC would be expected to occur over the longest time interval (from 36 to 48 hours). An alternate hypothesis might be a synchronized population in which case the largest increase in %iRBC could be during the longest time interval or during a shorter interval depending on the initial median age of the sample introduced into culture. If the authors could lay out their logic and expectations in that level of detail, it would greatly help with comprehension.

Related to that issue, the y-axes in Fig. 3C,D, and Fig. 5E are difficult to interpret since the time steps on the x-axis are uneven and/or unstated. For example, the first set of points in Fig. 5E indicates the fold change in % iRBCs from what time point to 18 hours in culture? The x-axis in the left three panels for the dry season occur at 6 hour intervals, so possibly the first set of points indicates the fold change from 12 to 18 hours but that should be stated clearly in the caption. The right two panels skip over hour 42, so the rightmost set of points appears to be fold change over 12 hours, which is difficult to compare to the fold change over six hours for the other sets of points. The authors should consider plotting the %iRBC over time, either the mean or individual lines for each replicate, with the x-axis appropriately spaced (e.g., double the space between 36 and 48 as compared to the interval between 30 and 36). These time series data would be much easier to interpret plotted in that way. If the authors retain only the mean in Fig. 5E, it might be worth keeping present Fig. 5E as a supplemental figure.

The differences in percentage of rings could still be merely the result of population dynamics. To be clear, a bias towards earlier developmental stages is not the result of "varying abilities to expand parasitemia" (line 543-544) but rather the result of faster population expansion. This pattern can also be seen in human populations, where there is a bias towards younger individuals in countries with faster population growth, while countries with declining number show an aging population. If you randomly sampled individuals from a country with rapid population growth and from a country with a declining population, you would find a larger proportion of older individuals from the latter. If the individuals were instead red blood cells infected with malaria parasites, then samples from a declining population will show more older stages, which will be able to generate new infected RBCs more quickly in vitro. The statement in lines 545-548 is difficult to understand-don't Fig. 5C and D show larger %iRBCs in malaria cases both in vivo and in vitro (respectively)? The authors correctly point out that fevers are not required to obtain synchrony, but the study they cite from Farnert shows a mix of synchrony and asynchrony in asymptomatic infections, and there is not enough known about synchrony to determine what pattern of synchrony would be expected in clinical versus asymptomatic infections. Sampling infections at different times (e.g., at the first acute peak versus chronic infection) would result in very different age distributions even if the underlying populations showed identical asynchrony, so lines 555-559 require more elaboration or rephrasing.

Lines 221-224: Which lab-adapted strain was used here?

Lines 357-358: The authors state that the increase in parasitemia was similar but was the interval between samples similar? The interpretation is very different for a similar increase in parasitemia that takes place over a shorter time interval (faster proliferation) or a longer time interval (slower proliferation) versus if the time interval was similar.

More context is needed for the interesting findings with respect to persistence (or lack thereof) of different clones (lines 413-415). Does the lack of overlap in alleles between strains circulating in the wet versus dry seasons necessarily indicate that some clones failed to transmit or could it also mean that clones successfully transmitted and recombined within the mosquito to result in different alleles circulating the following wet season?

Lines 569-end: In the potential explanations of *P. falciparum* seasonal response to antibodies, more explanation is needed to justify the authors conclusions. Moreover, the conclusions need to be more clearly stated with more detail on the biological impact and what patterns should be expected.

Figure 2D: I appreciate the color map, but it's difficult to distinguish which p-values are significant (fall to the right of the red line indicated in the key). Could the authors add asterisks to indicate which boxes exhibit significant differences?

Cross-commenting comments:

As far as the authors' responses to Rev. 3, I believe they have done all they can do within the limits of the experimental design (which of course is constrained by practical and ethical considerations). I agree with the authors' argument that their negative results are still useful and important. However, there are some points that are critical and appear to be limited to their responses to Rev. 3 rather than presented in the main text. If included in the main text, I believe that will suffice to address the concerns raised by Rev. 3.

1. The power analysis performed in response to major point 1 should be added to the supplement and referenced in the main text. This is a critical component of the analysis when sample sizes are so limited.
2. The added analysis (esp. Kaplan-Meier) performed in response to point 3a should be included in the supplement.
3. Regarding rev. 3's argument in favor of quantifying the correlation between %rings and parasitemia, I believe there is a more fundamental problem regarding the authors' prediction regarding the expected pattern. Incorporating the correlation coefficients they find could be helpful if they can clarify their expectations regarding why there should ever fail to be a positive correlation between %rings and parasitemia in a growing infection.

4. The issue of non-independent samples is an important one, and I appreciate the authors additional analysis that showed a lack of clustering of samples by individual (though I'm not totally convinced that addresses the issue). The authors report relatively consistent metabolite abundance over time for individuals, so I believe they might be better off presenting the graph that shows that consistency and using a single (or mean) value per individual and thereby avoid the problem of repeated measures. If they wish to retain repeated measures for individuals then they should include individual ID as a variable as rev. 3 suggests.

5. The authors present GLMM results for 3 different models in a table in the text and elect to retain the Poisson model in the main text (response to rev. 3 point 3d). It's not essential, but I would urge the authors to include that table and the details of those models in the supplement.

Minor--worth mentioning that plasma samples used in sensing experiments were not heat-inactivated in the text (rev. 3 minor point 1)?

-

*REPLY to REV2 (and REV3)***Referee #2** (Remarks for Author):

The authors have made important changes in this revision, but the central argument still needs further explanation and justification. One of the major points is that the parasites transmitted later in the season are more likely to survive to the end of the dry season and thus more likely to transmit again, but without further elaboration it is not clear whether this pattern reflects a change in parasite allocation/life history strategies. It seems like the authors are not arguing for a winnowing-down of the faster growing parasite genetic variants in infections over the course of the dry season, leaving only the variants with poor ability to sequester (and hence cause clinical disease). Instead, the authors seem to be alluding to differences in var gene expression and not in genetic background of different variants, such that late in the dry season strains are forced to use var genes that do not allow efficient sequestration. When transmitted at the beginning of the wet season, these strains can once again use their full var gene repertoire to enable fast growth within hosts. If that is the argument, it needs to be laid out clearly. (Note that all line numbers refer to the clean draft without track-changes.)

We appreciate the positive comment on the alterations already made to the previous version of our manuscript. And we now also further clarify in lines 300-303 that at the population level, the majority of alleles detected are found both in malaria cases and dry season infections, and it is only rare haplotypes that may be seen in only one of the conditions. And in lines 457-460 we describe that most of the haplotypes observed at any point in the wet season could both persist until the end of the dry season or be cleared. Additionally, we show in lines 467-472 that older age of the host is linked with shorter infection and lower probability of maintaining persistent infections until the end of the dry season likely indicating that it is not the genetic background of the parasite determining its ability to survive the dry season, but the immune context imposing reduced sequestration in asymptomatic infections. (all line numbers refer to the clean version without track-changes.)

Generally, much of what's written in the point-by-point responses could benefit the paper if included. Point 6 in the response letter: "Only a small proportion of intra-erythrocytic asexual parasites differentiate into sexual development (Kafsak et al. 2014), so we believe this would have a negligible effect on rates of replication in the blood. Although we have not quantified gametocyte carriage in the samples used in this study, we acknowledge the critical importance of sexual stages in resuming transmission as the rainy season returns. Gametocytes have been shown not to last longer than ~20 days in circulation, hence maintaining asexual replication during the dry season may be the best way to ensure a proportion of parasites converting into gametocytes when mosquitoes return." This justification could be included and elaborated on in the main paper. Though the authors discount the role of transmission investment, that is another mechanism by which parasite strains could slow their growth (see theory from Koella & Antia 1995 Theor Pop Biol about the potential for such changes in investment to be adaptive in evading immune clearance), and depending on environmental conditions, transmission investment can be enormous depending on in vitro conditions (up to 70% see Bruce et al. 1990 Parasitology).

*We have now incorporated our response to the reviewers in the manuscript's revised discussion in lines 528-537, making mention to transmission investment as a possible mechanism by which parasite strains could slow their growth, and we cite both Bruce et al. and Cornet et al.. Bruce et al. show that single schizont cultured with fresh red blood cells originate either only gametocytes or only asexual parasites, and that the produced gametocytes were very low at low parasite densities in culture, while high parasite densities led to a greater proportion of gametocytes. This observation would in fact suggest lower gametocyte investment during the dry season, when parasitaemias are very low. Likewise, Cornet et al. show in the *P. relictum* infection model that transmission was potentiated through increased asexual replication but no direct effect on gametocyte production was described.*

The discussion of results for lipids needs a lot more context to justify the section header "Seasonality and asymptomatic infections have little to no detectable effect on the plasma metabolites, lipids and proteins". What should readers make of the finding that one Lyso-PC species (Lyso-PC 20:4) was significantly increased in malaria cases (compared to what?) and another, Lyso-PC 18:0 was significantly decreased in malaria cases versus dry season infected or negative individuals? Are there dose response curves in the literature for these other Lyso-PC species and their impact on sexual differentiation (i.e., something like Fig. 2C in Brancucci et al. 2017 Cell for Lyso-PC 16:0)? Is there a reason Lyso-PC 16:0 wasn't included in this analysis (was it not present)?

We thank the reviewer for highlighting this point. We have now changed the section header to "Seasonality and asymptomatic infections promote minimal differences on plasma metabolites, lipids and proteins", which we believe is valid given that the only significant differences were observed when comparisons include the clinical malaria cases' samples.

Regarding the LysoPC 16:0, it was not included in the lipidomic analysis performed by the colleagues in the University of Mainz. Brancucci et al. 2017 tested Lyso-PC 16:0, Lyso-PC 18:0 and LysoPC 18:1 and report that all three species inhibit gametocytogenesis at 20uM. We now make mention of this point in the revised result section in lines 239-240. Lyso-PC 20:4 effect on Plasmodium falciparum's gametocytogenesis has, to our knowledge, not been tested or described in the literature.

It would be helpful for the authors to lay out their logic of their hypothesis explicitly and compare expected patterns to a clearly defined null hypothesis. For example, the authors put a lot of weight on the timing of the

highest increase in parasitemia, but the logic for when the largest increase would be expected under their hypothesis and under the null expectation is not explained. Is the null hypothesis that the parasite population is asynchronous and multiplying at a constant rate? In that case, the largest increase in %iRBC would be expected to occur over the longest time interval (from 36 to 48 hours). An alternate hypothesis might be a synchronized population in which case the largest increase in %iRBC could be during the longest time interval or during a shorter interval depending on the initial median age of the sample introduced into culture. If the authors could lay out their logic and expectations in that level of detail, it would greatly help with comprehension.

We have now tried to make clearer in lines 244-245 and 251-253 that experiments shown in Fig 3 were meant to interrogate if para freshly isolated parasites from infected Malian individuals would be more sensible to seasonal cues than lab-adapted strains. We considered that the absence a sensing test on fresh parasites, it could not be excluded that lack of sensing responses would be due to lab-adaptation. With this experiment we also questioned if P. falciparum was more responsive to plasma supplementation in certain times of the year than others. Our null hypothesis was, on one hand, that if freshly collected parasites were in general more responsive to seasonal serological cues than lab-adapted ones, we would detect a difference in response to the dry vs wet plasma supplementation in all conditions (Jan, May and MAL) of the experiment; and on the other hand we considered that if a sensing ability would be occurring only during some part of the year, our test of parasites collected in the beginning (Jan), end (May) of the dry season, and during clinical cases of malaria (MAL) would allow defining the condition at which sensing was most present. We now explain these points in the result section and reordered the description of the results to emphasize the lack of difference in overall increase of parasitaemia, and we moved the section describing the differences in time to achieve the increase to later in the text. We believe this description is relevant, or readers may be left questioning why increases in parasitaemia occurs at different times.

In culture, parasites will increase parasitaemia whenever schizonts rupture and reinvade new RBCs; and this is linked to the initial developmental stages present of the sample when placed in in vitro culture. For the increase in parasitaemia to be delayed until the 36-46 hours in culture, the parasite population must be synchronous and composed of very young rings when the experiment starts, as shown to occur in the samples collected from malaria cases in the wet season. However, more asynchronous populations will, once in culture, originate earlier increases in parasitaemia; these will be as early and as large depending on how developed and how frequent the non-early ring stages are upon sample collection from the patient, as shown to occur in the samples collected from asymptomatic infections. Likewise, dry season parasitaemias with lower % of rings are shown to lead to earlier increases in parasitaemia than asymptomatic infections in the wet season.

Related to that issue, the y-axes in Fig. 3C,D, and Fig. 5E are difficult to interpret since the time steps on the x-axis are uneven and/or unstated. For example, the first set of points in Fig. 5E indicates the fold change in % iRBCs from what time point to 18 hours in culture? The x-axis in the left three panels for the dry season occur at 6 hour intervals, so possibly the first set of points indicates the fold change from 12 to 18 hours but that should be stated clearly in the caption. The right two panels skip over hour 42, so the rightmost set of points appears to be fold change over 12 hours, which is difficult to compare to the fold change over six hours for the other sets of points. The authors should consider plotting the %iRBC over time, either the mean or individual lines for each replicate, with the x-axis appropriately spaced (e.g., double the space between 36 and 48 as compared to the interval between 30 and 36). These time series data would be much easier to interpret plotted in that way. If the authors retain only the mean in Fig. 5E, it might be worth keeping present Fig. 5E as a supplemental figure.

The reviewer is correct that we could provide more information on figures Fig. 3 C,D, and Fig. 5E. We have now expanded the text and figure legend to make clear that measurements were done at 0, 18, 24, 36, an /or 46 h, and that the plots show the fold change between each timepoint and its preceding one, which in case of the 18h will be the start point of the culture at 0h. It can now be read in the legends "Fold change is defined as %iRBC $t(n)$ /%iRBC $t(n-1)$ ".

The reviewer is also correct that 42h is not shown, as no measurements were done at that specific timepoint. Nevertheless, we have now plotted the mean fold change per sample group (Dry season, October or MAL) over equally spaced time intervals (18, 24, 30, 36, 42 and 48), and these further support the conclusion drawn from Fig. 5E. This is now included in lines 378-379 and in the new Supplementary figure 6. Furthermore, by fitting a Loess regression model, the overall trend in the fold changes over time can also be captured, as seen in the plot below.

Additionally, although we agree with the reviewer that spacing the x-axis to the appropriate distances between the time of measurements may be helpful, we do not think that showing the data as %iRBC over time would afford clarity to the reader. As shown in Fig. 5A parasitaemias are very broad within and between the different groups, and averages of parasitaemias will not transmit the patterns as clearly as the fold changes do. For readers with keen curiosity about individual parasitaemias, the raw data is available on the source data on the EMBO Molecular Medicine website and the raw .fcs files are available at flow repository website under ID FR-FCM-Z76J.

The differences in percentage of rings could still be merely the result of population dynamics. To be clear, a bias towards earlier developmental stages is not the result of "varying abilities to expand parasitemia" (line 543-544) but rather the result of faster population expansion. This pattern can also be seen in human populations, where there is a bias towards younger individuals in countries with faster population growth, while countries with declining number show an aging population. If you randomly sampled individuals from a country with rapid population growth and from a country with a declining population, you would find a larger proportion of older individuals from the latter. If the individuals were instead red blood cells infected with malaria parasites, then samples from a declining population will show more older stages, which will be able to generate new infected RBCs more quickly in vitro. The statement in lines 545-548 is difficult to understand-don't Fig. 5C and D show larger %iRBCs in malaria cases both in vivo and in vitro (respectively)? The authors correctly point out that fevers are not required to obtain synchrony, but the study they cite from Farnert shows a mix of synchrony and asynchrony in asymptomatic infections, and there is not enough known about synchrony to determine what pattern of synchrony would be expected in clinical versus asymptomatic infections. Sampling infections at different times (e.g., at the first acute peak versus chronic infection) would result in very different age distributions even if the underlying populations showed identical asynchrony, so lines 555-559 require more elaboration or rephrasing.

We have corrected former lines 543-544, as suggested by the reviewer, which can be now found in line 559. And we also made clearer, in agreement with the reviewer's comment that faster expanding populations in vivo may also promote sampling of younger rather than older parasite populations.

Lines 221-224: Which lab-adapted strain was used here?

We have added to the text the information that was only available on the figure legend stating that the strain used was Pf3D7.

Lines 357-358: The authors state that the increase in parasitemia was similar but was the interval between samples similar? The interpretation is very different for a similar increase in parasitemia that takes place over a shorter time interval (faster proliferation) or a longer time interval (slower proliferation) versus if the time interval was similar.

We have edited the manuscript to better describe the length of intervals analysed 16h (0 -16), 8h (16h – 24), 6h (24 - 30 and 30 - 36) or 12h (36 - 48), and we state that increases of parasitaemia were measured between any two consecutive timepoints of the experiment. Our measurement detects the magnitude and the time after culture at which we can observe the highest increase of parasitaemia, the intervals of time are similar across conditions and we observe earlier increases in asymptomatic infections despite similar parasitaemia increase between samples collected in the dry or wet season, and regardless of symptoms.

This could indicate a faster than 48 h intraerythrocytic replicative cycle in asymptomatic infections, or alternatively, that these parasites circulate longer without adhering to the host vascular endothelium, for being more developed than circulating parasites in clinical malaria cases at the time of the blood draw, which is in fact what we see and have previously described.

The increase in parasitaemia observed in asymptomatic season samples is not necessarily occurring in a shorter interval. The asymptomatic samples have a higher proportion of later stages, and these have already progressed further in their developmental cycle towards an earlier increase in parasitaemia as we observe in vitro. Samples from malaria cases, that are almost exclusively composed of young rings, have a longer way to conclude the intra-erythrocytic cycle culminating in later parasitaemia increase.

More context is needed for the interesting findings with respect to persistence (or lack thereof) of different clones

(lines 413-415). Does the lack of overlap in alleles between strains circulating in the wet versus dry seasons necessarily indicate that some clones failed to transmit or could it also mean that clones successfully transmitted and recombined within the mosquito to result in different alleles circulating the following wet season?

In former lines 413-415 we state: "The broad range of parasitaemia, and differences of ring-stage proportion observed in asymptomatic individuals during the wet season (Oct) compared to the dry season (shown in Fig. 5) suggest that within an infection not all parasite clones will reach the persistent state required to bridge two wet seasons." And by this we aim to suggest that parasites showing, in October, characteristics more alike to parasites causing malaria (young rings and high parasitaemia) in the wet season are likely to disappear as we do not see those features at the end of the dry season, but the reviewer is correct that this does not need to be related to particular alleles and we agree that recombination during mosquito infection can indeed promote different alleles in the following wet season. In lines 426-429 we have removed the word "clones" from our statement and included a sentence to describe the decrease in parasitaemia seen during the dry season that supports loss of parasites since transmission until the end of the dry season.

We also would like to point out that the lack of overlap in alleles between strains circulating in the wet versus dry seasons is only observed at the individual level (shown in Fig. 4), where we demonstrate that it is unlikely that clones persisting from the preceding dry season will promote clinical malaria symptoms within the same host, once the wet season and transmission restart.

However, at the population level, as stated above we observed that the majority of the 47 ama1 haplotypes found persisting until the end of the dry season in some individuals, can also be found to be present and to clear prior to the end of the wet season in other individuals, indicating that no particular ama1 haplotype is more prone to persist than others.

Lines 569-end: In the potential explanations of *P. falciparum* seasonal response to antibodies, more explanation is needed to justify the authors conclusions. Moreover, the conclusions need to be more clearly stated with more detail on the biological impact and what patterns should be expected.

We have reworded this section of the discussion

Figure 2D: I appreciate the color map, but it's difficult to distinguish which p-values are significant (fall to the right of the red line indicated in the key). Could the authors add asterisks to indicate which boxes exhibit significant differences?

We agree with the reviewer that the colour code of the scale bar and the line defining the p-value threshold was not sufficiently clear. We now highlight with a red outline the boxes referring to significantly enriched lipids in either group comparison.

Cross-commenting comments:

As far as the authors' responses to Rev. 3, I believe they have done all they can do within the limits of the experimental design (which of course is constrained by practical and ethical considerations). I agree with the authors' argument that their negative results are still useful and important. However, there are some points that are critical and appear to be limited to their responses to Rev. 3 rather than presented in the main text. If included in the main text, I believe that will suffice to address the concerns raised by Rev. 3.

1. The power analysis performed in response to major point 1 should be added to the supplement and referenced in the main text. This is a critical component of the analysis when sample sizes are so limited.

The power analysis has now been added to the method section in lines 932-942 and the figure made in reply to Rev. 3 is now included in the manuscript as the new Supplementary fig. 8.

2. The added analysis (esp. Kaplan-Meier) performed in response to point 3a should be included in the supplement.

We have included the Kaplan-Meier analysis and the linear mixed effects model in the revised version of Supplementary fig. 6.

3. Regarding rev. 3's argument in favor of quantifying the correlation between %rings and parasitemia, I believe there is a more fundamental problem regarding the authors' prediction regarding the expected pattern.

Incorporating the correlation coefficients they find could be helpful if they can clarify their expectations regarding why there should ever fail to be a positive correlation between %rings and parasitemia in a growing infection.

We have now made clearer that we do find a positive correlation between the percentage of rings and parasitaemia across all samples, and we now incorporate the correlation coefficients and P values as suggested by the reviewer. We now state in lines 359.362 that significant positive correlations were found across all samples ($R = 0.56$, $p < 2.2e-16$), and in the dry ($R = 0.47$, $p = 1.2e-09$) and wet season ($R = 0.54$, $p = 2.8e-06$) asymptomatic samples, while the narrow range of ~100% of ring-stages observed in malaria cases samples did not allow the detection of a significant correlation ($R = -0.3$, $p = 0.065$) (Fig. 5D).

*To the reviewer's point regarding possible lack of a positive correlation between %rings and parasitaemia in a growing infection, Silamut and White have reported in 1993 variable relations between circulating early and late stages of *P. falciparum*, and these do not always accompany the parasite density; in fact, the presence of*

schizonts in the peripheral blood, particularly if accompanied by high parasitaemia, is known as a poor prognostic sign in falciparum malaria (Field, 1949) likely due to the saturated cytoadhesion to the host endothelium.

4. The issue of non-independent samples is an important one, and I appreciate the authors additional analysis that showed a lack of clustering of samples by individual (though I'm not totally convinced that addresses the issue). The authors report relatively consistent metabolite abundance over time for individuals, so I believe they might be better off presenting the graph that shows that consistency and using a single (or mean) value per individual and thereby avoid the problem of repeated measures. If they wish to retain repeated measures for individuals then they should include individual ID as a variable as rev. 3 suggests.

Although we acknowledge the reviewer's point about sample independence, we believe that we are not substantially violating the independence assumption of the pairwise t-test, and we describe below our reasons:

- we do not detect differences between asymptomatic infections and negative individuals despite the repeated measures.

- the only differences observed - and only in a few metabolites – are LARGE and driven by depletion of metabolites in the clinical malaria cases due to the high parasitaemias, as has been previously reported in Colvin et al 2020.

- correlation of metabolite profiles from the same individual are broadly overlapping with correlations across individuals; and clearly an individual's metabolite abundance profile varies considerably over time. (we determined all pairwise correlations (R-squared) across all metabolites and between all individuals and timepoints, and compared those correlations for timepoints from different individuals and for timepoints within the same individual. Then, we looked at how the between and within correlation of malaria cases vs not (asymptomatic or negative) and see that the distribution of the correlation values overlaps between the two groups, regardless of sample numbers (see plot below). What this implies is that regardless of repeated sampling of the same individual, the correlation is not more statistically significant than if we would take timepoints from different individuals, when comparing to the clinical malaria cases. Likewise, we see the correlations also overlapping when comparing all other infection status (asymptomatic or negative) for the between or the within individual timepoint comparison. This is now shown alongside the power analysis in new Supplementary figure 9.

Distribution of R-squared (R^2) value of a linear regression model of all possible pairwise combinations of metabolite abundance for all timepoints and individuals split by pairwise comparisons within the same individual (applies only to individuals with repeated measurements) and all timepoints between individuals. Individuals with no paired timepoints were excluded and since only the first malaria episode per person was included, the within comparison of MAL does not apply.

Jointly, these factors indicate that our results are not just an artefact of repeated sampling, as suggested by Reviewer #3

- Regarding the potential use the mean, we could do this, but it would further reduce our power which is already on the lower end of what is optimal. Had we seen that there was no or very little variation in metabolite abundances within individuals over time -- we would agree with the reviewer that taking the mean would be necessary despite the negative effect on power -- however, given the observed variation, we believe that, on balance, retaining the repeated sampling is the best way to extract as much real biology from the data as possible.

- Regarding including IDs in the PCA, we have done this in our first response to this comment; and we have now included both the PCA (Supplementary fig.1) and correlation distributions (Supplementary fig. 9) as a supplementary figure in the revised version of the manuscript.

5. The authors present GLMM results for 3 different models in a table in the text and elect to retain the Poisson model in the main text (response to rev. 3 point 3d). It's not essential, but I would urge the authors to include that table and the details of those models in the supplement.

We have now specified in lines 482-483 that the model retained for this was the Poisson GLMM. Additionally, as suggested by the reviewer, we have included the table with the comparison of the 3 models as a supplementary table included in Supplementary fig. 6.

Minor--worth mentioning that plasma samples used in sensing experiments were not heat-inactivated in the text (rev. 3 minor point 1)?

We have now updated the method section to indicate this.

8th Jul 2024

Dear Dr. Portugal,

Thank you for the submission of your revised manuscript to EMBO Molecular Medicine. Your manuscript has now been re-reviewed by one of the original reviewers. Based on their advice (included below), I am pleased to inform you that we will be able to accept your manuscript pending the following final amendments and appropriate response to the reviewer:

- 1) In the main manuscript file, please include keywords to max. 5.
- 2) Data Availability: Please move the Data Availability section to the end of the Materials and Methods section. Please also update the statement by removing reviewer access passwords and ensure that all datasets are publicly available. Please also ensure that a specific URL is included for each dataset.
- 3) Please include a "Disclosure and competing interests statement". We updated our journal's competing interests policy in January 2022 and request authors to consider both actual and perceived competing interests. Please review the policy <https://www.embopress.org/competing-interests>. If you are confident that neither you nor any of your co-authors have anything to declare that readers could interpret as influencing or affecting your work, please state that "The authors state they have no competing interests or disclosures."
- 4) Author contributions: Please remove it from the manuscript and specify author contributions in our submission system. CRediT has replaced the traditional author contributions section because it offers a systematic machine-readable author contributions format that allows for more effective research assessment. You are encouraged to use the free text boxes beneath each contributing author's name to add specific details on the author's contribution. More information is available in our guide to authors:
<https://www.embopress.org/page/journal/17574684/authorguide#authorshipguidelines>
- 5) In the Materials and Methods, please take care of the following:
 - Studies with human research participants: Please confirm, in the section where you declare that the study was authorized, that the experiments conformed to the principles set out in the WMA Declaration of Helsinki and the Department of Health and Human Services Belmont Report. Please note that this is a separate statement from the specific ethics committee approval and informed consent.
 - Primers: please ensure primers used are included in the Methods (or if included in table format, that the table is included in the Appendix)
 - Please ensure that a statement on whether or not blinding was done is included in the Methods even if no blinding was done.
- 6) All Materials and Methods need to be described in the main text using our 'Structured Methods' format, which is required for all research articles. According to this format, the Methods section includes a Reagents and Tools Table (listing key reagents, experimental models, software and relevant equipment and including their sources and relevant identifiers) followed by a Methods and Protocols section describing the methods using a step-by-step protocol format. The aim is to facilitate adoption of the methodologies across labs. More information on how to adhere to this format as well as a downloadable template (.docx) for the Reagents and Tools Table can be found in our author guidelines:
<https://www.embopress.org/page/journal/17574684/authorguide#structuredmethods>
An example of a Method paper with Structured Methods can be found here:
<https://www.embopress.org/doi/10.15252/msb.20178071>.
- 7) For the figures and figure legends, please take care of the following:
 - Please remove all figures from main manuscript file and leave only main figure legends placed after the references. The figures should be uploaded individually as high-resolution files.
 - There are 10 supplementary figures. Up to 5 can be made Expanded View figures, which should also be removed from the main manuscript and uploaded individually as high-resolution files. Each figure will need to fit onto one page. The legends should stay in the manuscript, with the heading Expanded View Figures Legends, and placed after the main figure legends. The remaining 5 can be compiled in an Appendix file, with the legends under each figure, and renamed Appendix Figure S1, etc. The appendix should be uploaded in PDF format and needs a table of contents with page numbers. Please make sure to update the callouts of all figures in the main manuscript text. Please check "Author Guidelines" for more information:
<https://www.embopress.org/page/journal/17574684/authorguide#figureformat>
 - Please indicate the statistical test used for data analysis in the legends of figure 6e, supplementary figure 1d.
 - Please note that the box plots need to be defined in terms of minima, maxima, centre, bounds of box and whiskers, and percentile in the legends of figures 2d; 4c; 5a, c, e; 6b-c, supplementary figures 1c; 2c; 3a-b; 5a-b; 6a-b; 7b.
 - Please note that the box plots need to be defined in terms of minima, maxima, bounds of box and whiskers in the legends of figures 3a-c, e-f; 6d.
 - Please note that information related to n is missing in the legends of figures 2d; 3a-b, e-f; 4c; 6c, supplementary figures 3a-c; 5a-b; 6a-b; 7b.
 - Although 'n' is provided, please describe the nature of entity for 'n' in the legends of figure 6b, supplementary figure 2c.
- 8) Tables: Please rename the tables to Dataset EV1, etc. Each dataset will need its legend removed from the manuscript and added to the corresponding file in a separate tab. Please update their callouts in main manuscript text.
- 9) Funding: Please ensure that all funding sources are entered into the manuscript submission system, and that the project numbers are included where applicable

10) Synopsis:

- Synopsis image: Please provide a graphic that summarises the main findings of the manuscript on a glance and upload it as a high-resolution jpeg file 550 pixels wide x (250-400) pixels high.

- Synopsis text: Please provide a short standfirst (maximum of 300 characters, including space), limit the bullet points to max. 5 and upload it as a separate .doc file. Please write the bullet points to summarise the key NEW findings. They should be designed to be complementary to the abstract - i.e. not repeat the same text. We encourage inclusion of key acronyms and quantitative information (maximum of 30 words / bullet point). Please use the passive voice.

11) Source Data: Please ensure that the Source Data are uploaded as a single source data file (zipped) per figure, with the panels clearly visible in the folder structure.

12) The Paper Explained: Please provide "The Paper Explained" and add it to the main manuscript text. Please check "Author Guidelines" for more information. <https://www.embopress.org/page/journal/17574684/authorguide#researcharticleguide>

13) For more information: This space should be used to list relevant web links for further consultation by our readers. Could you identify some relevant ones and provide such information as well? Some examples are patient associations, relevant databases, OMIM/proteins/genes links, author's websites, etc...

14) Please place individual sections of the manuscript in the following order: Title page - Abstract & Keywords - Introduction - Results - Discussion - Materials & Methods - Data Availability - Acknowledgements - Disclosure and Competing Interests Statement - The Paper Explained - For More Information - References - Figure Legends - Expanded View Figure Legends.

15) As part of the EMBO Publications transparent editorial process initiative (see our policy here:

https://www.embopress.org/transparent-process#Review_Process), EMBO Molecular Medicine will publish online a Peer Review File (PRF) to accompany accepted manuscripts. This file will be published in conjunction with your paper and will include the anonymous referee reports, your point-by-point response and all pertinent correspondence relating to the manuscript. Let us know whether you agree with the publication of the PRF and as here, if you want to remove or not any figures from it prior to publication. Please note that the Authors checklist will be published at the end of the PRF.

16) Please provide a point-by-point letter INCLUDING my comments as well as the reviewer's reports and your detailed responses (as Word file).

I look forward to reading a new revised version of your manuscript as soon as possible.

Yours sincerely,

Poonam Bheda

Poonam Bheda, PhD
Scientific Editor
EMBO Molecular Medicine

*** Instructions to submit your revised manuscript ***

***** Reviewer's comments *****

Referee #2 (Remarks for Author):

The authors have addressed my concerns, and this revised MS is much clearer. I have a few minor suggestions:

I'd suggest that the authors clarify in the abstract their conclusion, that parasite transmission investment and capacity for growth does not change but rather parasites are forced to suboptimal PfEMP1 proteins as lengthy infections progress, leading to increased clearance and slower population growth. The wording "decreased virulence...is...a consequence of infection length" is vague enough that it is likely to cause confusion. This point is explained much more fully in the discussion, and it would be helpful to explain in similar terms (albeit more succinctly) here.

Specify how the duration of infection underpins the seasonality of clinical symptoms (lines 478-479). These lines need something as clear as the explanation in the discussion lines 658-665.

The wording in lines 516-518 does not seem to line up with the wording of the caption in supp. fig. 7G, which seems to suggest that older children have shorter times until they exhibit malaria. Clarify wording?

***** Reviewer's comments *****

Referee #2 (Remarks for Author):

The authors have addressed my concerns, and this revised MS is much clearer. I have a few minor suggestions:

I'd suggest that the authors clarify in the abstract their conclusion, that parasite transmission investment and capacity for growth does not change but rather parasites are forced to suboptimal PfEMP1 proteins as lengthy infections progress, leading to increased clearance and slower population growth. The wording "decreased virulence...is...a consequence of infection length" is vague enough that it is likely to cause confusion. This point is explained much more fully in the discussion, and it would be helpful to explain in similar terms (albeit more succinctly) here.

We revised the abstract to more clearly mention that capacity for growth is maintained in the dry season. As we present no data on transmission investment or PfEMP1 expression in this study we feel more comfortable maintaining those statements in the discussion only. Our last sentence of the abstract now reads: "We propose that the decreased virulence observed in persisting parasites during the dry season is not due to the parasites sensing ability, nor is it linked to a decreased capacity for parasite replication but rather a consequence decreased cytoadhesion associated with infection length".

Specify how the duration of infection underpins the seasonality of clinical symptoms (lines 478-479). These lines need something as clear as the explanation in the discussion lines 658-665.

We revised the manuscript accordingly, and now added to the sentence in question "where all parasites have the potential to cause malaria shortly after transmission but progressively transition to a persistent, low parasitaemia state"

The wording in lines 516-518 does not seem to line up with the wording of the caption in supp. fig. 7G, which seems to suggest that older children have shorter times until they exhibit malaria. Clarify wording?

*The reviewer is correct, we have by mistake copied the wrong caption to the legend of Sup Fig. 7G. we have now corrected it to: "G. Kaplan-Meier analysis showing the probability of survival of *P. falciparum* clones (y-axis) over time, from the start of the wet season (July 2012) to the end of the dry season (May 2013) (x-axis) in individuals of 7-9 (aqua) and 10-13 (coral) years old, with the log-rank test used to determine statistical differences."*

7th Aug 2024

Dear Dr. Portugal,

Thank you for the submission of your revised manuscript to EMBO Molecular Medicine. In checking our formatting requests, we found the following issues that still need to be resolved:

- 1) The Paper Explained: Please proofread and correct this section as there are numerous spelling errors.
- 2) Synopsis text: Please shorten the standfirst to a maximum of 300 characters, including spaces. There are also a few spelling errors in the text, please check and correct.
- 3) Track changes in the manuscript can now be turned off/changes accepted.

Please address these issues and resubmit your manuscript as soon as possible.

Yours sincerely,

Poonam Bheda

Poonam Bheda, PhD
Scientific Editor
EMBO Molecular Medicine

The authors addressed the minor editorial issues.

9th Aug 2024

Dear Dr. Portugal,

We are pleased to inform you that your manuscript is accepted for publication and is now being sent to our publisher to be included in the next available issue of EMBO Molecular Medicine.

Yours sincerely,

Poonam Bheda, PhD
Scientific Editor
EMBO Molecular Medicine
